# ACHIEVING $\tilde{O}(1)$ STRONG CONSTRAINT VIOLATION AND SUBLINEAR STRONG REGRET IN ONLINE CMDPS

## ABSTRACT

We study safe online reinforcement learning in Constrained Markov Decision Processes (CMDPs) under strong regret and violation metrics. Existing methods that achieve sublinear strong reward regret inevitably incur cumulative strong constraint violation that grows with the number of episodes $T$. To address this limitation, we propose **Flex**ible safety **D**omain **O**ptimization via **M**argin-regularized **E**xploration (**FlexDOME**), the first algorithm in the literature that provably achieves near-constant $\tilde{O}(1)$ strong constraint violation and ensures a sublinear $\tilde{O}(T^{7/8})$ strong reward regret. FlexDOME, built on the regularized primal-dual framework, introduces a decaying safety margin to the constraint threshold. This margin tightens the feasible region to avoid constraint violation, which relaxes in order $\tilde{O}(t^{-1/8})$ to guarantee feasibility, offering a proper safety-performance trade-off. We then propose a policy-dual divergence potential function that helps establish a non-asymptotic last-iterate convergence guarantee. Experiments demonstrate that Flex-DOME significantly enhances safety with negligible reward sacrifice, in full agreement with the theory.[1]

## 1 INTRODUCTION

Reinforcement Learning (RL) has achieved remarkable successes in recent years (Liu et al., 2024; Ruiz et al., 2025; Milani et al., 2024). It formulates sequential decision making as a Markov Decision Process (MDP), where an agent learns a policy to maximize cumulative reward (Sutton et al., 1998). However, classical MDPs lack sufficient mechanisms to ensure safety, hindering deployment in safety-critical environments (Garcıa & Fernández, 2015; Brunke et al., 2022). Constrained Markov Decision Processes (CMDPs) address this limitation by incorporating constraints on cumulative costs (Altman, 1999).

In online CMDPs, an agent needs to learn in an unknown environment while satisfying safety constraints per episode, which makes safety particularly challenging. Classic reward regret and constraint violation allow cancellations over time (Ding et al., 2020), obscuring prolonged unsafe behavior, which is unacceptable in safety-critical settings (Fisac et al., 2019). This motivates strong metrics, i.e., strong reward regret (sum of positive per-episode suboptimality) and strong constraint violation (sum of positive per-episode violations), with no cancellation (Efroni et al., 2020). Such strong safety guarantees naturally arise in settings like power-grid regulation, where cumulative violations induce mechanical or thermal stress, and in clinical control (e.g., automated anesthesia), where even a few severe threshold breaches may trigger irreversible harm (Su et al., 2025; Cai et al., 2023). In these cases, harms cannot be 'averaged out,' making strong metrics more suitable than classic ones. In this setting, a fundamental trilemma emerges among (i) stringent safety, (ii) optimal performance, and (iii) last-iterate convergence. Existing approaches are forced to compromise: on the one

---

[1] The code has been submitted and will be released.

hand, primal-dual methods achieve last-iterate convergence but their strong violations grow with the number of episodes $T$ (Müller et al., 2024; Kitamura et al., 2024); on the other hand, methods with tighter regrets often sacrifice last-iterate convergence by applying only to averaged policies (Stradi et al., 2024; 2025; Zhu et al., 2025). While achieving stringent safety (e.g., near-constant or even zero violation) is well-studied under the classic regret paradigm (Liu et al., 2021; Bai et al., 2022; Ma et al., 2024), these assurances vanish under the more demanding strong metrics. This naturally raises the pivotal question:

> *Can we design an efficient CMDP algorithm that achieves (i) $\tilde{O}(1)$ strong constraint violation, (ii) sublinear strong regret, and (iii) last-iterate convergence?*

We answer this question affirmatively. To address this challenge, we propose **Flex**ible safety **D**omain **O**ptimization via **M**argin-regularised **E**xploration (**FlexDOME**), a regularized primal-dual algorithm that achieves robust safety by tightening the feasible set with a decaying margin. The core intuition is to mimic human strategy for risk management: maintaining a 'margin for safety' when operating under uncertainty. This safety margin creates a proactive buffer against violations. At an early stage of learning, when the uncertainty is high, we take a large margin, steering the agent away from high-risk regions; as information accrues, the margin decays at a rate of $\tilde{O}(t^{-1/8})$ ($t$ is the training time), progressively relaxing conservatism and enabling the pursuit of (possibly) higher-reward policies near the boundary. The success of this dynamic margin hinges on a stable learning process. Standard primal-dual methods are often plagued by oscillatory dynamics (Efroni et al., 2020). We tame these oscillations by introducing entropy and $L_2$ regularization to ensure a strongly convex-concave optimization landscape.

We prove that FlexDOME attains $\tilde{O}(1)$ strong constraint violation and $\tilde{O}(T^{7/8})$ strong reward regret, with non-asymptotic last-iterate convergence. To the best of our knowledge, this is the first primal-dual algorithm to achieve all three guarantees; see Table 1. Our theoretical analysis unfolds in three stages. We first introduce a policy-dual divergence potential function, which establishes a linear convergence rate of our iterates towards the optimum of the margin-regularized problem. Next, we derive per-episode performance bounds that bridge the gap to the true CMDP optimum and characterize the inherent safety-performance trade-off. Finally, we integrate these components with a decaying safety margin that is slow enough to neutralize per-episode violation terms yet fast enough for the induced bias to vanish.

We conduct experiments on a randomly generated tabular CMDP, which fully corroborate our theoretical claims. Across both stochastic-threshold and standard fixed-threshold settings, FlexDOME consistently outperforms the vanilla primal-dual baseline (Efroni et al., 2020) and the state-of-the-art UOpt-RPGPD algorithm (Kitamura et al., 2024), maintaining near-zero instantaneous violations and achieving markedly lower cumulative strong violations. An ablation study further validates our design choices, confirming that the regularization framework is critical for preventing the oscillatory dynamics. We anticipate that our framework can pave the way for provably safe reinforcement learning in high-stakes domains.

**Related Work.** Under weak regret metrics, primal-dual methods establish $\tilde{O}(\sqrt{T})$ regret and constraint violation guarantees (Efroni et al., 2020). To enhance safety, subsequent works introduce a safety margin to primal-dual methods, attaining $\tilde{O}(\sqrt{T})$ weak regret and $\tilde{O}(1)$ weak constraint violation guarantees (Liu et al., 2021; Kalagarla et al., 2025). However, their analysis relies on using the cumulative safety margin to offset the cumulative constraint violation; consequently, the underlying primal-dual dynamics are still prone to the oscillations that preclude guarantees for strong regret or last-iterate convergence. The allowance for error cancellation makes weak regret an inadequate metric for safety-critical tasks. To address this, Efroni et al. (2020) introduce the more stringent strong regret metric, which accumulates only positive deviations of reward and constraint. Müller et al. (2023) propose an augmented Lagrangian method which attains sublinear strong regret/violation with a strictly known safe policy. Relaxing the requirement of a strictly safe policy, Müller et al. (2024) and Kitamura et al. (2024) propose a regularized primal-dual framework to achieve the last-iterate convergence guarantee with strong constraint violation, achieving rates of $\tilde{O}(T^{0.93})$

| Algorithm | Strong Regret | Strong Violation | Last-iterate Convergence | Unknown Safe Policy | Stochastic Threshold |
|---|---|---|---|---|---|
| (Müller et al., 2023) | $\tilde{O}(\sqrt{T})$ | $\tilde{O}(\sqrt{T})$ | ✓ | × | No |
| (Müller et al., 2024) | $\tilde{O}(T^{0.93})$ | $\tilde{O}(T^{0.93})$ | ✓ | ✓ | No |
| (Kitamura et al., 2024) | $\tilde{O}(T^{6/7})$ | $\tilde{O}(T^{6/7})$ | ✓ | ✓ | No |
| (Stradi et al., 2025) | $\tilde{O}(\sqrt{T})$ | $\tilde{O}(\sqrt{T})$ | × | ✓ | No |
| (Zhu et al., 2025) | $\tilde{O}(\sqrt{T})$ | $\tilde{O}(\sqrt{T})$ | × | ✓ | No |
| **FlexDOME** | $\tilde{O}(T^{7/8})$ | $\tilde{O}(1)$ | ✓ | ✓ | Yes |

Table 1: Comparison between FlexDOME and related work under strong regret and violation metrics. For clarity, dependencies on the state space ($S$), action space ($A$), and horizon ($H$) are omitted here.

and $\tilde{O}(T^{6/7})$, respectively. In parallel, Stradi et al. (2024) study adversarial loss with stochastic hard constraints that achieves $\tilde{O}(\sqrt{T})$ weak regret and near-constant strong violation. Stradi et al. (2025) and Zhu et al. (2025) attain a tighter $\tilde{O}(\sqrt{T})$ strong regret and violation only for averaged policies, which limit practical use. Table 1 summarizes the theoretical results from our work and the most relevant existing methods under strong regret and violation metrics.

## 2 PRELIMINARIES

**Constrained Markov decision process (CMDP).** We consider a finite-horizon Markov decision process (MDP), where the state space is denoted by $\mathcal{S}$ (with finite cardinality $S$), the action space by $\mathcal{A}$ (with finite cardinality $A$), and the horizon by $H$. At step $h \in [H]$, the agent occupies state $s_h \in \mathcal{S}$, takes action $a_h \in \mathcal{A}$, and the subsequent state $s_{h+1}$ is sampled from the transition probability $p : \mathcal{S} \times \mathcal{A} \times \mathcal{S} \to [0,1]$. $r_h : \mathcal{S} \times \mathcal{A} \to [0,1]$ represents the reward function at each step $h$. A policy $\pi = (\pi_1, \ldots, \pi_H) \in \Pi$ specifies a distribution $\pi_h(\cdot \mid s) \in \Delta(\mathcal{A})$ for every state–step pair, where $\Pi := \{(\pi_1, \cdots, \pi_H) \mid \forall h, s : \pi_h(\cdot \mid s) \in \Delta(\mathcal{A})\}$. A Constrained MDP augments this setting with $m$ constraints. For constraint $i \in [m]$ and step $h$, a constraint $d_{i,h}(s,a) \in [0,1]$ is incurred; the cumulative expectation must not fall below a given threshold $\alpha_i \in [0,H]$. Thus, a CMDP can be fully characterized by $\mathcal{M} = (\mathcal{S}, \mathcal{A}, H, p, r, d, \alpha)$.

In this work, we address a more challenging online learning setting where the model parameters are stochastic, including the safety thresholds. The agent has to collect samples for reward, constraints and thresholds to optimize the policy. Specifically, at each interaction $(s,a)$ for step $h$ and episode $t$, the agent observes stochastic samples: a reward $\tilde{r}_h^t(s,a)$, constraints $\{\tilde{d}_{i,h}^t(s,a)\}_{i=1}^m$, and thresholds $\{\tilde{\alpha}_{i,h}^t\}_{i=1}^m$ which are not state-action dependent. These are drawn from stationary but hidden distributions $\mathcal{R}$, $\{\mathcal{G}_i\}_{i=1}^m$ and $\{\mathcal{L}_i\}_{i=1}^m$, respectively. A detailed comparison between this setting and standard CMDPs is provided in Appendix A.

**Remark 1.** *It is worth noting that this setting covers the standard CMDPs under known thresholds. The fixed-threshold scenario can be viewed as a special instance of our framework, where the underlying threshold distribution is a Dirac delta function centered at the known constant value. Thus, all theoretical analyzes and results presented in this paper apply directly to the fixed-threshold setting without loss of generality.*

**Value and objective functions.** For any vector $v \in [0,1]^{\mathcal{S} \times \mathcal{A}}$ and policy $\pi \in \Pi$, consider the value functions

$$V_{v,h}^\pi(s) = \mathbb{E}_{\pi,p}\Big[\sum_{h'=h}^H v_{h'}(s_{h'}, a_{h'}) \mid s_h = s\Big], \quad Q_{v,h}^\pi(s,a) = \mathbb{E}_{\pi,p}\Big[\sum_{h'=h}^H v_{h'}(s_{h'}, a_{h'}) \mid s_h = s, a_h = a\Big],$$

where $V_{v,h}^{\pi}(s)$ denotes the expected sum of $v$ from step $h$ onward given $s_h = s$, and $Q_{v,h}^{\pi}(s, a)$ denotes the same expectation further conditioned on $a_h = a$. For notational brevity, set $V_v^{\pi} := V_{v,1}^{\pi}(s_1)$. The objective is to find a policy solution $\pi^{\star}$ to the following policy optimization problem,

$$\max_{\pi \in \Pi} \ V_r^{\pi} \quad \text{subject to} \quad V_{d_i}^{\pi} \geq \alpha_i \quad (\forall i \in [m]), \tag{1}$$

which identifies a policy that maximizes the expected cumulative reward while ensuring that the expected cumulative value of each constraint signal satisfies its threshold.

**Training protocol.** Across $T$ episodes, a policy $\pi_t$ is selected at the beginning of episode $t$ and executed for $H$ steps. The goal is to simultaneously minimize its strong reward regret and strong constraint violation,

$$\mathcal{R}_T(r) := \sum_{t=1}^{T} \left[ V_r^{\pi^{\star}} - V_r^{\pi_t} \right]_+, \quad \mathcal{R}_T(d) := \max_{i \in [m]} \sum_{t=1}^{T} \left[ \alpha_i - V_{d_i}^{\pi_t} \right]_+.$$

These expressions measure the cumulative sum of only the positive deviations, capturing how much the reward underperforms the optimal or how much the constraints are violated in each episode. Each positive error contributes its full amount to the total, and no future episode can offset it. Throughout, we assume the following Slater condition, which is mild as it holds when there exists some (unknown) strictly feasible policy (Efroni et al., 2020; Ying et al., 2022; Ding et al., 2023; Kitamura et al., 2024).

**Assumption 1.** *There exists an unknown policy $\pi^0 \in \Pi$ such that $V_{d_i}^{\pi^0} \geq d_i^0$, where $d_i^0 > \alpha_i$ for all $i \in [m]$. Set the Slater gap $\Xi := \min_{i \in [m]} \{ d_i^0 - \alpha_i \}$.*

**Notation.** For any $x \in \mathbb{R}$, we define the operation $[x]_+ := \max\{0, x\}$ to be the positive truncation of $x$. We use $O(\cdot)$ and $\Omega(\cdot)$ to denote asymptotic upper and lower bounds, respectively, and $\Theta(\cdot)$ when a bound is asymptotically tight. The symbol $\tilde{O}(\cdot)$ hides polylogarithmic factors, and $\lesssim$ denotes inequality up to constants and polylogarithmic factors.

# 3 FLEXDOME

This section introduces our algorithm, Flexible safety Domain Optimization via Margin-regularized Exploration (FlexDOME).

## 3.1 THE PRIMAL-DUAL SCHEME IN FLEXDOME

**Safety margin.** Our core idea is to proactively establish a 'margin of safety' to mitigate the effects of uncertainty in guaranteeing safety. We translate this idea into a formal mechanism by first introducing a time-varying safety margin $\epsilon_{i,t}$ for each episode $t$ and constraint $i$ into the original optimization problem (1):

$$\max_{\pi \in \Pi} \ V_r^{\pi} \quad \text{s.t.} \quad V_{d_i}^{\pi} \geq \alpha_i + \epsilon_{i,t} \quad (\forall i \in [m]), \tag{2}$$

where the constraints are tightened by the safety margins to enhance safety during learning. Correspondingly, the Lagrangian function is defined as follows:

$$\mathcal{L}_t(\pi, \lambda) := V_r^{\pi} + \sum_{i=1}^{m} \lambda_i \left( V_{d_i}^{\pi} - \epsilon_{i,t} - \alpha_i \right),$$

where $\lambda = [\lambda_1, \ldots, \lambda_m]^\top \in \mathbb{R}_+^m$ is the vector of non-negative dual variables (or Lagrange multipliers), with each $\lambda_i$ corresponding to the $i$-th constraint.

**Regularizations.** Standard primal-dual CMDP formulations lack strong convexity-concavity, which can cause oscillatory dynamics (Stooke et al., 2020). These oscillations can breach a simple safety buffer, and thus fail to achieve stringent safety guarantees (Moskovitz et al., 2023; Müller et al., 2024). To overcome this limitation, we introduce a *time-varying regularization framework* that provides the geometric stability necessary for the safety margin to be effective. By augmenting the Lagrangian with dynamically scaled entropy and $\ell_2$-norm penalties, we reshape the optimization landscape. Entropy regularization, $\mathcal{H}(\pi)$, smooths the policy space and ensures the primal objective is strongly concave, preventing extreme policy updates. The $\ell_2$ penalty, $\frac{1}{2}\|\lambda\|^2$, acts as a contraction mapping that suppresses excessively large dual variables and guarantees the dual objective is strongly convex, reducing gradient oscillations. Together, these components create a strongly convex-concave structure. The resulting regularized Lagrangian for regularization parameter $\tau_t > 0$ at episode $t$ is formulated as:

$$\mathcal{L}_{\tau_t,t}(\pi,\lambda) := V_r^\pi + \lambda^\top\left(V_d^\pi - \epsilon_t - \alpha\right) + \tau_t\left(\mathcal{H}(\pi) + \frac{1}{2}\|\lambda\|^2\right), \tag{3}$$

where $\mathcal{H}(\pi) := -\mathbb{E}_\pi\left[\sum_{h=1}^H \log(\pi_h(a_h|s_h))\right]$ is the policy entropy and $\epsilon_t$ denotes the vector of safety margins. The objective is to find the saddle point of this regularized problem over the policy space $\Pi$ and a compact dual domain $\mathcal{C} := [0, 4H/\Xi]^m$:

$$\max_{\pi\in\Pi}\min_{\lambda\in\mathcal{C}}\mathcal{L}_{\tau_t,t}(\pi,\lambda). \tag{4}$$

The strongly convex-concave structure guarantees that this problem has a unique saddle point, $\left(\pi_{\tau_t,\epsilon}^\star, \lambda_{\tau_t,\epsilon}^\star\right)$, which we define as the regularized optimizer for episode $t$.

## 3.2 ESTIMATIONS

FlexDOME employs a hybrid estimation strategy to navigate the unknown environment. It constructs optimistic estimates for rewards, constraints, and the entropy term to encourage exploration, while the transition model and thresholds are unbiasedly estimated directly from empirical data. Let $(s_h^l, a_h^l)$ denote the state-action pair visited in episode $l$ at step $h$. The term $\mathbf{1}_{\{\cdot\}}$ is the indicator function; thus, $N_h^{t-1}(s,a) = \sum_{l=1}^{t-1}\mathbf{1}_{\{s_h^l=s,a_h^l=a\}}$ is the total number of visits to $(s,a)$ at step $h$ before episode $t$. Then, the empirical averages for rewards, constraints, thresholds and transition probabilities can be calculated as follows:

$$\hat{r}_h^{t-1}(s,a) := \frac{\sum_{l=1}^{t-1}\tilde{r}_h^l(s,a)\,\mathbf{1}_{\{s_h^l=s,a_h^l=a\}}}{\max\{1, N_h^{t-1}(s,a)\}}, \qquad \hat{d}_{i,h}^{t-1}(s,a) := \frac{\sum_{l=1}^{t-1}\tilde{d}_{i,h}^l(s,a)\,\mathbf{1}_{\{s_h^l=s,a_h^l=a\}}}{\max\{1, N_h^{t-1}(s,a)\}},$$
$$\hat{\alpha}_i^{t-1} := \frac{\sum_{l=1}^{t-1}\sum_{h=1}^H\tilde{\alpha}_{i,h}^l}{(t-1)H}, \qquad \hat{p}_h^{t-1}(s' \mid s,a) := \frac{\sum_{l=1}^{t-1}\mathbf{1}_{\{s_h^l=s,a_h^l=a,s_{h+1}^l=s'\}}}{\max\{1, N_h^{t-1}(s,a)\}}. \tag{5}$$

We then construct the estimators for use in episode $t$. The safety threshold is estimated as the global empirical average of all historical observations: $\overline{\alpha}_i^t := \hat{\alpha}_i^{t-1}$. As each true threshold is constant, this method is data-efficient and yields an estimate that is independent of any specific state-action pair. The remaining state-action dependent estimators are constructed as follows:

$$\overline{r}_h^t(s,a) := \hat{r}_h^{t-1}(s,a) + \phi_h^{t-1}(s,a), \qquad \overline{d}_{i,h}^t(s,a) := \hat{d}_{i,h}^{t-1}(s,a) + \phi_h^{t-1}(s,a),$$
$$\overline{\psi}_h^t(s,a) := -\log(\pi_h^t(a\,|\,s)) + \phi_h^{p,t-1}(s,a)\log(A), \qquad \overline{p}_h^t(s'|s,a) := \hat{p}_h^{t-1}(s'|s,a). \tag{6}$$

The bonus term $\phi_h^t(s,a) := \phi_h^{r,t}(s,a) + \phi_h^{p,t}(s,a)$ combines the uncertainties from both rewards and transition estimations, where for any confidence parameter $\delta \in (0,1)$, the reward bonus is $\phi_h^{r,t}(s,a) = O\left(\sqrt{\frac{\log(mSAHT/\delta)}{\max\{1,N_h^t(s,a)\}}}\right)$ and the transition bonus is $\phi_h^{p,t}(s,a) = O\left(H\sqrt{\frac{S+\log(SAHT/\delta)}{\max\{1,N_h^t(s,a)\}}}\right)$.

### 3.3 LEARNING ALGORITHM

We now present **FlexDOME**, detailed in Algorithm 1. In each episode $t$, the algorithm first constructs an optimistic empirical CMDP $\mathcal{M}_t := (\mathcal{S}, \mathcal{A}, H, \overline{p}_t, \overline{r}_t, \overline{d}_t, \overline{\alpha}_t)$, using the estimators from Section 3.2. It then performs policy evaluation. To prevent optimistic bonuses from inflating value estimates unboundedly, we use a Truncated Policy Evaluation (TPE) routine (Efroni et al., 2020). TPE computes $V$-values for the constraint estimates and $Q$-values for the composite objective $\overline{y}_t := \overline{r}_t + \lambda_t^\top \overline{d}_t + \tau_t \overline{\psi}_t$, which aggregates the optimistic estimates of the reward, constraints, and entropy. See Algorithm 3 in Appendix D for details. Based on this, FlexDOME executes a single primal-dual update: the policy (primal variable) is updated via mirror ascent, and the dual variables are updated via projected gradient descent. The resulting policy is then deployed to collect new data for the next iteration.

---

**Algorithm 1** Flexible safety Domain Optimization via Margin-regularized Exploration (FlexDOME)

---

1: **Input:** $\mathcal{C} = [0, \frac{4H}{\Xi}]^m$, stepsize $\eta_t$, regularization $\tau_t$, number of episodes $T$, safety margin $\epsilon_{i,t}$ $(\forall i)$
2: **Initialize:** policy $\pi_{1,h}(a \mid s) = \frac{1}{A}$ $(\forall s, a, h)$, $\lambda_1 = \mathbf{0} \in \mathbb{R}^m$
3: **for** $t = 1$ **to** $T$ **do**
4:      Update estimators $\overline{r}_t, \overline{d}_t, \overline{\alpha}_t, \overline{\psi}_t$, and $\overline{p}_t$
5:      Truncated policy evaluation (Algorithm 3) for $\overline{y}_t$ and $\overline{d}_t$:
6:          $\left( \hat{Q}_{\overline{y}_t}^t(\cdot), \hat{V}_{\overline{d}_t}^t \right) \leftarrow \text{TPE}\left( \pi_t, \lambda_t, \overline{r}_t, \overline{d}_t, \overline{\psi}_t, \overline{p}_t \right)$
7:      Policy Update $(\forall h, s, a)$: $\pi_{t+1,h}(a \mid s) \propto \pi_{t,h}(a \mid s) \exp\left( \eta_t \hat{Q}_{h, \overline{y}_t}^t(s, a) \right)$
8:      Dual Update: $\lambda_{t+1} \leftarrow \text{Proj}_{\mathcal{C}}\left( (1 - \eta_t \tau_t)\lambda_t - \eta_t \left( \hat{V}_{\overline{d}_t}^t - \epsilon_t - \overline{\alpha}_t \right) \right)$
9:      Rollout $\pi_t$ and update counters and empirical model (i.e., $\hat{r}_t, \hat{d}_t, \hat{\alpha}_t, \hat{p}_t, N_t$)
10: **end for**

---

## 4 THEORETICAL ANALYSIS

This section establishes the theoretical guarantees for FlexDOME. We first present our main results on strong regret and violation bounds, followed by our practical guarantee of last-iterate convergence. We then detail the key technical lemmas that underpin these results. The full proofs for this section are deferred to Appendix E and Appendix F.

### 4.1 STRONG REGRET BOUNDS

We first provide the main theoretical results for FlexDOME.

**Theorem 1** (Strong regret bounds for reward and violation). *Let $\eta_t = t^{-3/4}$, $\tau_t = t^{-1/8}$ for $t \geq 1$, and $\epsilon_{i,t} = 6\sqrt{H^3 C_B} \left( t^{-1/8} \cdot \log(SAHt/\delta')^{1/4} \right)$ for any constraint $i$. For any confidence parameter $\delta \in (0, 1)$, with probability at least $1 - \delta$, Algorithm 1 achieves the following bounds:*

$$\mathcal{R}_T(r) \leq \tilde{O}(T^{7/8}) \quad and \quad \mathcal{R}_T(d) = \tilde{O}(1),$$

*where $T$ denotes the number of episodes, $C_B = \left( 1 + \frac{8mH}{\Xi} \right) \left( 4H\sqrt{2SA} \left( H\sqrt{S} + H + 1 \right) \right) + \frac{4mH}{\Xi}\sqrt{2H}$ is a $T$-independent constant and $\tilde{O}$ hides polylogarithmic factors in $(S, A, H, m, \log(T), \log(\frac{1}{\delta}), \Xi)$.*

Theorem 1 establishes $\tilde{O}(1)$ strong constraint violation and sublinear $\tilde{O}(T^{7/8})$ strong regret. This is the first result achieving this guarantee for online CMDPs. Our results are improved upon the state-of-the-art strong

constraint violation $\tilde{O}(\sqrt{T})$ proven by Stradi et al. (2025) and Zhu et al. (2025) to $\tilde{O}(1)$, and do not rely on prior knowledge of a strictly safe policy. Although they achieve tighter $\tilde{O}(\sqrt{T})$ strong reward regret, the algorithms were only established on the convergence of the averaged iterates and and can only achieve $\tilde{O}(\sqrt{T})$ strong constraint violation.

The core mechanism is the calibrated decay of the safety margin. Its role here is fundamentally different from its use in weak-regret settings (Liu et al., 2021; Kalagarla et al., 2025). Rather than using the sum of margins to cancel the total violation, our margin acts as a slowly decaying function designed to neutralize the per-episode violation term within the cumulative sum. The decay rate $\tilde{O}(t^{-1/8})$ is critical: a faster decay would be insufficient to absorb per-episode violations, causing the cumulative violation to grow, while a slower decay would persistently over-constrain the problem, inflating the reward regret. This precise calibration is what enables FlexDOME to achieve constant violation while maintaining sublinear regret. Concurrently, the diminishing learning rate $\eta_t$, regularization term $\tau_t$, and safety margin $\epsilon_{i,t}$ jointly ensure the algorithm's iterates converge towards the solution of the original optimization problem (1).

## 4.2 LAST-ITERATE CONVERGENCE

Beyond the regret and violation bounds, we prove that FlexDOME achieves last-iterate convergence, which is crucial for practical deployment, as it ensures the final policy is verifiably safe and near-optimal (Ding et al., 2023). The detailed proof is provided in Appendix F. For clarity, we first formally present its definition.

**Definition 1** (Last-iterate convergence). *A method that produces iterates $\{\pi_t\}_{t\in\mathbb{N}^+} \subset \Pi$ is called last-iterate convergent if for any constraint $i$*

$$V_r^{\pi^\star} - V_r^{\pi_t} \to 0 \quad and \quad \left[\alpha_i - V_{d_i}^{\pi_t}\right]_+ \to 0 \quad (t \to \infty).$$

Based on the definition, we present the following theorem.

**Theorem 2** (Last-iterate convergence). *Conditioned on Assumption 1, for small $\varepsilon > 0$ and $t = \Omega(\varepsilon^{-7})$, if $\eta_t = \Theta(\varepsilon^4)$, $\tau_t = \Theta(\varepsilon^2)$ and $\epsilon_{t,i} = \Theta(\varepsilon)$ for any constraint $i$, then we have*

$$\left[V_r^{\pi^\star} - V_r^{\pi_t}\right]_+ \le \Theta(\varepsilon), \quad \left[\alpha_i - V_{d_i}^{\pi_t}\right]_+ = 0 \quad (\forall\, i \in [m]).$$

Theorem 2 demonstrates that the final policy is guaranteed to be both $\varepsilon$-optimal and strictly constraint-satisfying. The core of this proof lies in the selection of the safety margin, which is set to be proportional to $\varepsilon$. Our analysis shows that the per-episode error terms are also of order $\Theta(\varepsilon)$. By choosing a sufficiently large constant of proportionality for the safety margin, we guarantee that after $\Omega(\varepsilon^{-7})$ iterations, the margin will absorb these error terms, driving the final violation to zero.

## 4.3 ANALYSIS SKETCH

This section outlines the core technical arguments underpinning our main theorems. Our analysis hinges on two key techniques: first, leveraging the convergence properties of a novel policy-dual potential function, which serves as a Lyapunov measure to track the learning dynamics, and second, rigorously characterizing the per-episode safety-performance trade-off enabled by the safety margin within our regularized framework.

We begin by introducing the policy-dual divergence potential function as follows:

$$\Phi_t = \sum_{s,h} \mathbb{P}_{\pi^\star_{\tau_t,\epsilon}}[s_h = s] \, \mathrm{KL}\!\left(\pi^\star_{\tau_t,\epsilon,h}(\cdot\,|\,s), \pi_{t,h}(\cdot\,|\,s)\right) + \frac{1}{2}\left\|\lambda^\star_{\tau_t,\epsilon} - \lambda_t\right\|^2.$$

It quantifies how closely the current policy-dual iterate $(\pi_t, \lambda_t)$ approximates the optimal margin-regularized policy-dual pair $(\pi^\star_{\tau_t,\epsilon}, \lambda^\star_{\tau_t,\epsilon})$. We prove that this function contracts at each step.

**Lemma 1** (Convergence to margin-regularized saddle points). *Let $\eta_t$, $\tau_t \leq 1$ and a confidence parameter $\delta \in (0, 1)$. With probability at least $1 - \delta$, the policy-dual divergence potential of Algorithm 1 holds*

$$\Phi_{t+1} \leq \exp\left(-\sum_{j=1}^{t} \eta_j \tau_j\right) \Phi_1 + \frac{HC + D}{2} \sum_{j=1}^{t} \eta_j^2 \exp\left(-\sum_{k=j+1}^{t} \eta_k \tau_k\right) + \sum_{j=1}^{t} \eta_j \delta_j \exp\left(-\sum_{k=j+1}^{t} \eta_k \tau_k\right).$$

*where $C = \exp\left(\eta_t H \left(1 + \frac{4mH}{\Xi} + \tau_t \log(A)\right)\right) \left(2A^{\eta_t \tau_t} H^2 \left(1 + \frac{4mH}{\Xi} + \tau_t \log(A)\right)^2 + \frac{128\tau_t^2 \sqrt{A}}{e^2}\right)$, $D = m\left(H + \tau_t \left(\frac{4H}{\Xi}\right)\right)^2$ and $\delta_j = \hat{V}_{\bar{y}_j}^j - V_{y_j}^{\pi_j} + \sum_i \frac{4H}{\Xi}\left(\hat{V}_{\bar{d}_{i,j}}^j - V_{d_i}^{\pi_j}\right)$.*

Lemma 1 shows that the iterates of FlexDOME contract towards a neighborhood of the margin-regularized saddle point. The upper bound consists of three primary components: (i) a decaying term dependent on the initial potential $\Phi_1$; (ii) the accumulated optimization error from the primal-dual updates; and (iii) the statistical error from estimating the unknown CMDP model.

**Remark 2.** *Our analysis extends the framework of Müller et al. (2024) in two critical aspects. First, our framework accommodates decaying learning rates $\eta_t$ and regularization $\tau_t$, which are essential for achieving our final regret bounds. Second, the statistical error term, $\delta_j$, in our analysis explicitly incorporates the uncertainty from estimating the stochastic safety thresholds.*

To bridge the gap to the original CMDP, our analysis divides the episodes into two parts, divided by $C'' = O\left((H^3 C_B)^4 \log^2(H^3 C_B)\right)$. For episodes $t < C''$, the margin may be large, so we bound the per-episode regret by $H$. For episodes $t \geq C''$, the decaying margin is guaranteed to be sufficiently small such that $\epsilon_{i,t} \leq \Xi/2$. Consequently, for this regime, the optimization problem (4) has at least one feasible solution by Assumption 1 and exhibits strong duality. Our main technical lemmas are therefore derived for these episodes. We then introduce error bounds linking the convergence metric with performance guarantees.

**Lemma 2** (Per-episode trade-off). *For any $t \geq C''$, any constraint $i$ and any sequence $\{\pi_t\}_{t \in [T]}$, it holds*

$$\left[V_r^{\pi^\star} - V_r^{\pi_t}\right]_+ \leq H^{3/2}\left(2\Phi_t\right)^{1/2} + H\log(A)\tau_t + \frac{H}{\Xi}\epsilon_{i,t},$$

$$\max_{i \in [m]} \left[\alpha_i - V_{d_i}^{\pi_t}\right]_+ \leq \left[H^{3/2}\left(2\Phi_t\right)^{1/2} + \frac{4H}{\Xi}\tau_t - \epsilon_{i,t}\right]_+.$$

Lemma 2 is the crux of our analysis, as it mathematically crystallizes the safety-performance trade-off. The first inequality shows that the reward sub-optimality is upper-bounded by three terms: learning error, regularization bias and safety margin bias. The second inequality reveals that the constraint violation is bounded by the same learning error and regularization bias, but is directly counteracted by the safety margin. This formalizes the *trade-off*: a larger margin $\epsilon_{i,t}$ provides a stronger buffer against violation but simultaneously increases the potential reward sub-optimality. The main theorems are then established by integrating these lemmas and meticulously calibrating the decay schedules of all parameters to navigate this trade-off.

## 5 EXPERIMENTS

We conduct experiments comparing our FlexDOME algorithm with the vanilla primal-dual baseline (Efroni et al., 2020) and the state-of-the-art (SOTA) UOpt-RPGPD algorithm (Kitamura et al., 2024). Vanilla PD provides a standard primal-dual update that allows us to isolate the effect of the time-varying regularization and safety margin, while UOpt-RPGPD represents the strongest existing method with a proven last-iterate convergence guarantee. Our comparison therefore focuses on algorithms that are most relevant to the last-iterate regime studied in this work. Targeted ablation studies are performed to dissect the contributions of

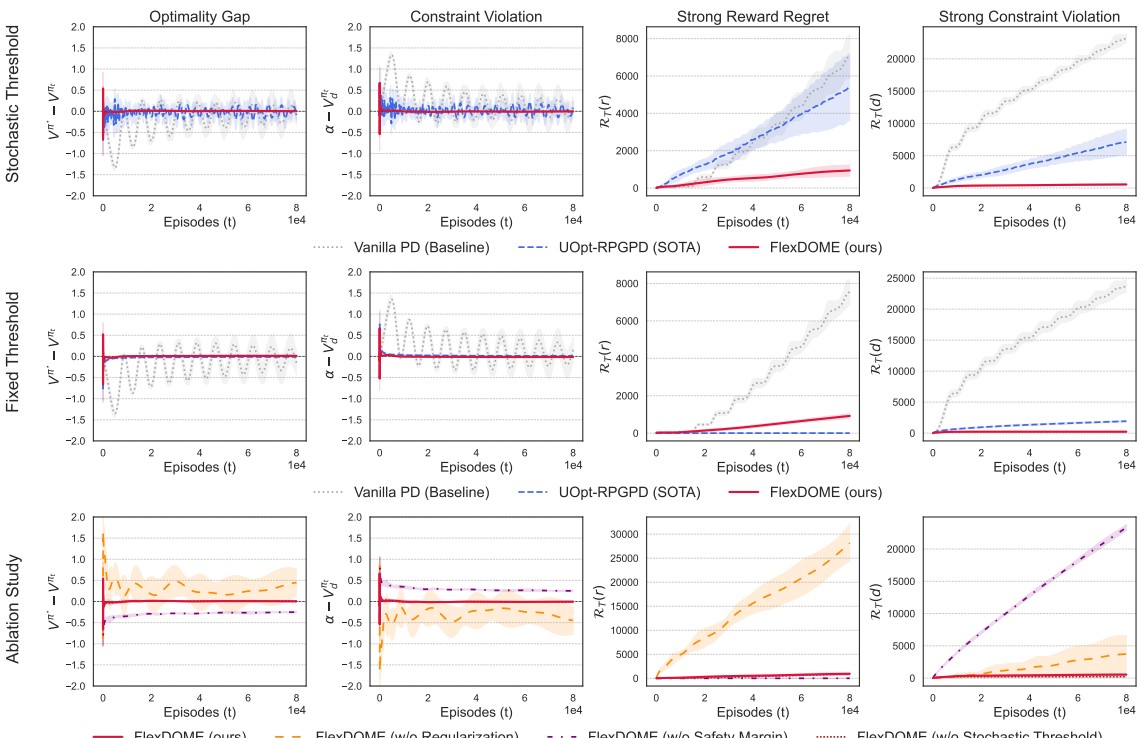

Figure 1: Performance comparison of **FlexDOME** (ours) against **UOpt-RPGPD** and **Vanilla PD** baselines under both stochastic-threshold (**top row**) and fixed-threshold (**middle row**) settings. The **bottom row** presents an ablation study on key components of our method: the safety margin, regularization, and the stochastic threshold mechanism. All plots show the mean and standard error over 5 seeds.

our algorithm's key components. Performance is measured by the instantaneous optimality gap and constraint violation, alongside their corresponding strong regrets. The evaluations are performed on randomly generated tabular CMDPs. Following the setup in Kitamura et al. (2024), we construct environments where the objective and the constraint are in conflict to create a non-trivial trade-off between reward maximization and violation minimization. We evaluate in two threshold settings: a stochastic environment where the per-episode threshold is drawn from a Gaussian distribution, and a standard fixed-threshold case. We set $S = 20$, $A = H = 5$ and focus on a single constraint for clear visualization. All results are averaged over 5 independent runs with different random seeds. Further experimental details are provided in Appendix G.

Our empirical results fully corroborate our theoretical findings. Figure 1 shows that, in the stochastic-threshold environment, FlexDOME is the only algorithm that maintains near-zero instantaneous violation, leading to a flat, near-constant cumulative strong violation curve. In contrast, both the baseline and the SOTA method exhibit oscillatory behavior and incur growing strong constraint violation. The middle row of Figure 1 shows that FlexDOME retains its safety advantage in standard fixed-threshold environments; however, this robust constraint satisfaction comes at the cost of a slight trade-off in reward regret compared to UOpt-RPGPD. The ablation studies (bottom row) confirm that removing the regularization framework reintroduces the severe oscillations characteristic of standard primal-dual methods, underscoring its necessity for stable learning. FlexDOME closely tracks an oracle (with access to the true threshold), confirming that our estimation mechanism is efficient and does not compromise safety or performance.

## 6 CONCLUSION

This paper addresses the challenge of achieving stringent, provable safety in online CMDPs under strong-regret metrics. We propose FlexDOME, a novel regularized primal-dual algorithm that incorporates a decaying safety margin to navigate the safety-performance trade-off. We prove that FlexDOME can simultaneously achieve a near-constant $\tilde{O}(1)$ strong constraint violation, sublinear $\tilde{O}(T^{7/8})$ strong reward regret, and a non-asymptotic last-iterate convergence guarantee. To our best knowledge, FlexDOME is the first algorithm in the literature to achieve these three guarantees concurrently. Our experiments corroborate these theoretical findings. Our work provides an affirmative answer to the open question of whether an efficient primal-dual method can achieve constant strong constraint violation with sublinear strong regret. However, a gap remains to the optimal $\tilde{O}(\sqrt{T})$ strong regret. We hope our analysis inspires further research on no-regret learning in CMDPs, including extensions to settings with function approximation and infinite-horizon problems.

## ETHICS STATEMENT

We declare no potential conflict of interest. We are not aware of any issues related to legal compliance, research integrity, or other ethical considerations.

## REPRODUCIBILITY STATEMENT

We have taken several steps to ensure reproducibility. All assumptions underlying our theoretical results are explicitly stated in Section 2. For the empirical results, we use only publicly available environments, described in Section 5, with training details, hyperparameters, and evaluation metrics reported in Appendix G. To further support reproducibility, we have submitted and will publicly release our code.

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

## A CLARIFICATIONS ON THE DISTINCTION BETWEEN STANDARD CMDPs AND CMDPs WITH STOCHASTIC THRESHOLDS

This section provides a rigorous analysis of the fundamental differences between the standard Constrained Markov Decision Process (CMDP) and the CMDPs with stochastic thresholds, as introduced in this work. We first formally define each setting, contrast their optimization objectives by highlighting the informational disparity, analyze for the non-degenerate nature of our problem formulation, and finally discuss the generality of our results.

### A.1 FORMAL DEFINITIONS OF CMDPs WITH STOCHASTIC THRESHOLDS

We begin by formally defining the two problem settings.

**Definition 2** (Standard CMDPs). *A standard episodic CMDP is defined by the tuple $\mathcal{M} = (\mathcal{S}, \mathcal{A}, H, p, r, d, \boldsymbol{\alpha})$, where $\mathcal{S}, \mathcal{A}, H, p, r, d$ are the state space, action space, horizon, transition dynamics, reward function, and constraint functions, respectively. The threshold $\boldsymbol{\alpha} = (\alpha_1, \ldots, \alpha_m)$ is a vector of scalars, where each $\alpha_i \in \mathbb{R}$ is a **pre-specified and known constant** given as part of the problem definition.*

**Definition 3** (CMDPs with stochastic thresholds). *An episodic CMDP with stochastic thresholds is defined by the tuple $\mathcal{M}' = (\mathcal{S}, \mathcal{A}, H, p, r, d, \{\mathcal{L}_i\}_{i=1}^m)$, where the first six components are as defined above. For each constraint $i$, $\mathcal{L}_i$ is an unknown probability distribution from which the agent observes stochastic samples $\tilde{\alpha}_{i,h}^t \sim \mathcal{L}_i$ at each step $h$ and episode $t$. The expectation of these samples defines a threshold $\alpha_i = \mathbb{E}_{\mathcal{L}_i}[\tilde{\alpha}_{i,h}^t]$, where $\alpha_i$ is a scalar constant that is **unknown** to the agent.*

Algorithm 2 depicts this interaction, where at each episode $t$, the agent executes a policy $\pi_t$ and observes not only rewards and constraints, but also the thresholds themselves.

---

**Algorithm 2** Agent-Environment Interaction for $t \in [T]$

---

**Require:** Policy $\pi_t \in \Pi$
1: Environment initializes state $s_1 \in \mathcal{S}$
2: **for** $h = 1, \ldots, H$ **do**
3:      Agent takes action $a_h \sim \pi_t(\cdot \mid s_h)$
4:      Agent observes reward $\tilde{r}_h^t(s_h, a_h)$, constraint $\tilde{d}_{i,h}^t(s_h, a_h)$, and threshold $\tilde{\alpha}_{i,h}^t$ for $i \in [m]$
5:      Environment evolves to $s_{h+1} \sim p(\cdot \mid s_h, a_h)$
6: **end for**

---

## A.2    CONCENTRATION OF THE EMPIRICAL THRESHOLD ESTIMATOR

We analyze the concentration properties of the empirical threshold estimator defined in Equation 5. The following theorem establishes a high-probability bound on the deviation of this estimator from the true mean threshold $\alpha_i$.

**Lemma 3** (Concentration of empirical thresholds). *Assume the stochastic thresholds $\tilde{\alpha}_{i,h}^l$ are independently drawn for each episode $l \in [t]$ and step $h \in [H]$. Further, assume that each sample is bounded, such that $\tilde{\alpha}_{i,h}^l \in [0, H]$. Let the empirical estimator for the threshold of constraint $i$ at the beginning of episode $t + 1$ be defined as*

$$\hat{\alpha}_i^{t+1} := \frac{1}{tH} \sum_{l=1}^{t} \sum_{h=1}^{H} \tilde{\alpha}_{i,h}^l,$$

*and let the true mean be $\alpha_i = \mathbb{E}[\tilde{\alpha}_{i,h}^l]$. Then, for any $\delta \in (0, 1)$, with probability at least $1 - \delta$, the following bound holds:*

$$\left| \hat{\alpha}_i^{t+1} - \alpha_i \right| \leq \sqrt{\frac{H \log(2/\delta)}{2t}} := \zeta_i^{t+1}.$$

*Proof.* Let $\{X_j\}_{j=1}^n$ be a set of $n = tH$ independent random variables, where each $X_j$ corresponds to one of the observed stochastic thresholds $\tilde{\alpha}_{i,h}^l$ for $l \in [t], h \in [H]$. By assumption, each random variable is bounded within the interval $[0, H]$, thus for all $j$, the range $(b_j - a_j)$ is $H$.

The empirical estimator $\hat{\alpha}_i^{t+1}$ is the sample mean $\bar{X} = \frac{1}{n} \sum_{j=1}^n X_j$. The true mean $\alpha_i$ is the expected value of this sample mean, $\mathbb{E}[\bar{X}]$. By Hoeffding's inequality and Substituting our parameters ($n = tH$ and $b_j - a_j = H$), we have:

$$\mathbb{P}\left( \left| \hat{\alpha}_i^{t+1} - \alpha_i \right| \geq c \right) \leq 2 \exp\left( -\frac{2(tH)^2 c^2}{\sum_{j=1}^{tH} H^2} \right)$$

$$= 2 \exp\left( -\frac{2tc^2}{H} \right)$$

We set the right-hand side of the probability bound: $\delta = 2 \exp\left( -\frac{2tc^2}{H} \right)$. Solving for the deviation $c$, we obtain

$$c = \sqrt{\frac{H \log(2/\delta)}{2t}}.$$

Thus, with probability at least $1 - \delta$, the error $|\hat{\alpha}_i^{t+1} - \alpha_i|$ is bounded by $c$. This completes the proof. $\qquad\square$

**Lemma 4** (Union bound for empirical thresholds). *Given $\delta \in (0,1)$, with probability at least $1 - \delta$, the following holds uniformly for each constraint $i \in [m]$ and episode $t \in [T]$:*

$$\left|\hat{\alpha}_i^{t+1} - \alpha_i\right| \leq \zeta^{t+1},$$

*where $\zeta^{t+1} = \sqrt{\frac{H \log(2mT/\delta)}{2t}}$.*

*Proof.* By Lemma 3, for any given confidence level $\delta'$ and given constraint $i$, we have:

$$\mathbb{P}\left[\left|\hat{\alpha}_i^{t+1} - \alpha_i\right| \leq \sqrt{\frac{H \log(2/\delta')}{2t}}\right] \geq 1 - \delta'.$$

Taking a union bound over all possible choices of $i \in [m]$ and $t \in [T]$, we have:

$$\mathbb{P}\left[\bigcap_{i,t}\left\{\left|\hat{\alpha}_i^{t+1} - \alpha_i\right| \leq \zeta_i^{t+1}\right\}\right] \geq 1 - mT\delta'.$$

Letting $\delta = mT\delta'$ and substituting into $\zeta_i^{t+1}$, we derive the stated uniform bound with probability at least $1 - \delta$. This completes the proof. $\qquad\square$

The theorem demonstrates that the empirical estimator $\hat{\alpha}_i^{t+1}$ converges to the true mean $\alpha_i$ at a rate of $\mathcal{O}(1/\sqrt{t})$.

## B    PREPARATION LEMMAS

**Lemma 5** (Müller et al. (2024)). *Let $V := \Delta([d])$, and $g \in \mathbb{R}_{\geq 0}^d =: X$. Then $\tilde{x} := \arg\max_{x \in X} g^\top x - \frac{1}{\eta_t}\mathrm{KL}(\tilde{x}, x)$ and $\arg\max_{x \in V} g^\top x - \frac{1}{\eta_t}\mathrm{KL}(\tilde{x}, x)$ exist and are unique. Moreover, if $g$ only has non-negative entries, then for all $x^\star \in V$ we have*

$$g^\top(x^\star - x) \leq \frac{\mathrm{KL}(x^\star, x) - \mathrm{KL}(x^\star, x')}{\eta_t} + \frac{\eta_t}{2}\sum_{i=1}^d \tilde{x}_i g_i^2.$$

**Lemma 6** (Müller et al. (2024)). *The performance gap admits the decomposition:*

$$
\begin{aligned}
&V_{y_t}^{\pi^\star_{\tau_t,\epsilon}} - V_{y_t}^{\pi_t} \\
&= \hat{V}_{\bar{y}_t}^t - V_{y_t}^{\pi_t} \\
&\quad + \sum_{h \in [H]} \mathbb{E}\left[\langle \hat{Q}_{\bar{y}_t,h}^t(s_h,\cdot), \pi^\star_{\tau_t,h}(\cdot \mid s_h) - \pi_{t,h}(\cdot \mid s_h)\rangle \mid s_1, \pi^\star_{\tau_t,\epsilon}, p\right] \\
&\quad + \sum_{h \in [H]} \mathbb{E}\left[-\hat{Q}_{\bar{y}_t,h}^t(s_h,\cdot) + y_{t,h}(s_h,a_h) + \langle p_h(\cdot \mid s_h, a_h), \hat{V}_{\bar{y}_t,h+1}^t(\cdot)\rangle \mid s_1, \pi^\star_{\tau_t,\epsilon}, p\right].
\end{aligned}
$$

**Lemma 7** (Altman (1999)). *Suppose the transition function is $P$. For any mixed policy $\pi^{mix} = B_\gamma \pi^1 + (1 - B_\gamma)\pi^2$, where $B_\gamma$ is a Bernoulli distributed random variable with mean $\gamma$. Then there exists a Markov policy $\hat{\pi}$ that*

$$V_{r,h}^{\hat{\pi}}(p) = V_{r,h}^{\pi^{mix}}(p), \qquad \forall r, s, h.$$

**Lemma 8.** *Let $\{A_t\}_{t=1}^{\infty}$ and $\{B_t\}_{t=1}^{\infty}$ be two sequences of positive real numbers. Assume that the limit of their ratio exists and is a constant $L$ strictly less than 1:*

$$\lim_{t \to \infty} \frac{A_t}{B_t} = L < 1.$$

*Then the partial sum $S_T = \sum_{t=1}^{T} [A_t - B_t]_+$ is bounded by a constant that is independent of $T$, i.e., $S_T = O(1)$, where $[x]_+ := \max(0, x)$.*

*Proof.* To prove that the sum is $O(1)$, it must be shown that the corresponding infinite series $\sum_{t=1}^{\infty}[A_t - B_t]_+$ converges. For a series of non-negative terms, it is sufficient to show that the summand is identically zero for all terms beyond a finite threshold $t_0$. A non-zero term in the sum occurs only if $A_t > B_t$.

The given condition is $\lim_{t \to \infty} \frac{A_t}{B_t} = L$, where $L < 1$. By the formal definition of a limit, for every $\varepsilon > 0$, there exists a positive integer $t_0$ such that for all $t > t_0$, the inequality $\left| \frac{A_t}{B_t} - L \right| < \varepsilon$ holds. This is equivalent to:

$$L - \varepsilon < \frac{A_t}{B_t} < L + \varepsilon.$$

The objective is to prove that $\frac{A_t}{B_t} < 1$ for sufficiently large $t$. To achieve this from the inequality above, it is sufficient to ensure that the right-hand side, $L + \varepsilon$, is strictly less than 1. Since $L < 1$, the distance $1 - L$ is a fixed positive number. A valid and convenient choice for $\varepsilon$ is therefore:

$$\varepsilon = \frac{1 - L}{2}.$$

This choice of $\varepsilon$ is guaranteed to be positive.

For this choice of $\varepsilon$, the right-hand side of the limit inequality becomes:

$$L + \varepsilon = L + \frac{1 - L}{2} = \frac{2L + 1 - L}{2} = \frac{1 + L}{2}.$$

Since $L < 1$, it follows that $1 + L < 2$, and therefore $\frac{1+L}{2} < 1$.

By the definition of the limit, for the chosen $\varepsilon$, there must exist a threshold $t_0$ such that for all $t > t_0$:

$$\frac{A_t}{B_t} < L + \varepsilon = \frac{1 + L}{2}.$$

As it has been shown that $\frac{1+L}{2} < 1$, it follows that for all $t > t_0$:

$$\frac{A_t}{B_t} < 1.$$

Since $B_t$ is a positive sequence, the inequality $\frac{A_t}{B_t} < 1$ implies $A_t < B_t$, which in turn means $A_t - B_t < 0$ for all $t > t_0$. Therefore, the summand of the series becomes:

$$[A_t - B_t]_+ = \max(0, A_t - B_t) = 0, \quad \forall t > t_0.$$

The total sum can then be split into a finite part and a tail of zeros:

$$\sum_{t=1}^{T}[A_t - B_t]_+ = \sum_{t=1}^{t_0}[A_t - B_t]_+ + \sum_{t=t_0+1}^{T} 0 = \sum_{t=1}^{t_0}[A_t - B_t]_+.$$

This is a finite sum of finite numbers, which evaluates to a constant value that is independent of the upper limit $T$ (for $T > t_0$). Therefore, the sum is bounded by a constant, and the conclusion is that:

$$\sum_{t=1}^{T}[A_t - B_t]_+ = O(1). \qquad \square$$

## C  FEASIBILITY AND STRONG DUALITY FOR THE MARGIN-REGULARIZED CMDP

Recall $\epsilon_{i,t} = 6\sqrt{H^3 C_B}\left(t^{-1/8} \cdot \log(SAHt/\delta')^{1/4}\right)$ for all constraint $i$, where $\delta' = \delta/4$ and $C_B = \left(1 + \frac{8mH}{\Xi}\right)\left(4H\sqrt{2SA}\left(H\sqrt{S} + H + 1\right)\right) + \frac{4mH}{\Xi}\sqrt{2H}$ . The existence of a feasible solution to (4) can be guaranteed if $\epsilon_{i,t} \leq \Xi$. Let $C''$ be the smallest value such that for any $t \geq C''$, $\epsilon_{i,t} \leq \Xi/2$. Then this optimization problem (4) has at least one feasible solution for any $t \geq C''$. By calculation, we can obtain $C'' = O\left(K^4 \cdot \log^2(K)\right)$, where $K = H^3 C_B$ and then $C''$ is a constant and $T$-independent.

We establish the fundamental theoretical properties of the regularized Lagrangian formulation presented in Section 3. Our analysis proceeds by reformulating the problem in the space of occupancy measures. For clarity, we restate the regularized Lagrangian for a fixed episode $t$ and regularization parameter $\tau_t > 0$:

$$\mathcal{L}_{\tau_t,t}(\pi, \lambda) := V_r^\pi(p) + \lambda^\top \left(V_d^\pi(p) - \epsilon_t - \alpha\right) + \tau_t \mathcal{H}(\pi) + \frac{\tau_t}{2}\|\lambda\|^2,$$

where the optimization problem is $\max_{\pi \in \Pi} \min_{\lambda \in \mathcal{C}} \mathcal{L}_{\tau_t,t}(\pi, \lambda)$ over the policy space $\Pi$ and the compact dual domain $\mathcal{C} := [0, \frac{4H}{\Xi}]^m$.

**Lemma 9** (Strong duality of the margin-regularized problem). *For any fixed episode $t \geq C''$ and regularization parameter $\tau_t > 0$, the regularized CMDP problem exhibits strong duality. That is,*

$$\max_{\pi \in \Pi} \min_{\lambda \in \mathcal{C}} \mathcal{L}_{\tau_t,t}(\pi, \lambda) = \min_{\lambda \in \mathcal{C}} \max_{\pi \in \Pi} \mathcal{L}_{\tau_t,t}(\pi, \lambda),$$

*and both optima are attained.*

*Proof.* Let $q^\pi \in \mathbb{R}^{HSA}$ be the occupancy measure corresponding to a policy $\pi \in \Pi$, defined as $q_h^\pi(s,a) := \mathbb{P}[s_h = s, a_h = a|s_1; p, \pi]$. The set of all valid occupancy measures forms a convex polytope, denoted by $Q(p)$. By definition, the police entropy is $\mathcal{H}(\pi) = -\mathbb{E}_\pi[\sum_h \log \pi_h(a_h|s_h)]$. The expectation can be rewritten as a sum over the state-action space: $\mathcal{H}(\pi) = -\sum_{h,s,a} q_h(s,a) \log\left(\frac{q_h(s,a)}{\sum_{a'} q_h(s,a')}\right) := \mathcal{H}(q)$. The value functions and policy entropy can be expressed as linear and strictly concave functions of $q^\pi$, respectively. We can thus define an equivalent Lagrangian over the domain $Q(p) \times \mathcal{C}$:

$$\bar{\mathcal{L}}_{\tau_t,t}(q, \lambda) := r^\top q + \lambda(d^\top q - \epsilon_t - \alpha) + \tau_t \mathcal{H}(q) + \frac{\tau_t}{2}\|\lambda\|^2,$$

where $r, d \in \mathbb{R}^{HSA}$ are the vectors.

The optimization problem is equivalent to $\max_{q \in Q(p)} \min_{\lambda \in \mathcal{C}} \bar{\mathcal{L}}_{\tau_t,t}(q, \lambda)$. We verify the conditions for Sion's Minimax Theorem ((Sion, 1958)):

1. The domain $Q(p) \times \mathcal{C}$ is the product of a polytope and a hyperrectangle, and is therefore a non-empty, compact, and convex set.

2. The function $\bar{\mathcal{L}}_{\tau_t,t}(q, \lambda)$ is continuous over its domain.

3. For any fixed $\lambda \in \mathcal{C}$, $\bar{\mathcal{L}}_{\tau_t,t}(q, \lambda)$ is strictly concave in $q$. This is because $r^\top q + \lambda(d^\top q - \epsilon_t - \alpha)$ is linear in $q$, and the entropy term $\tau_t \mathcal{H}(q)$ is strictly concave for $\tau_t > 0$.

4. For any fixed $q \in Q(p)$, $\bar{\mathcal{L}}_{\tau_t,t}(q, \lambda)$ is strictly convex in $\lambda$. This is because $\lambda(d^\top q - \epsilon_t - \alpha)$ is linear in $\lambda$, and the term $\frac{\tau_t}{2}\|\lambda\|^2$ is strictly convex for $\tau_t > 0$.

Since all conditions are met, it guarantees that the max-min and min-max values are equal and that optimizers exist. $\qquad\square$

**Lemma 10** (Saddle point inequalities). *Let $(\pi^\star_{\tau_t,\epsilon}, \lambda^\star_{\tau_t,\epsilon})$ be the saddle point of $\mathcal{L}_{\tau_t,t}$. Then for any episode $t \geq C''$, any policy $\pi \in \Pi$ and any dual variable $\lambda \in \mathcal{C}$, the following two inequalities hold:*

*(i)* $\quad V_r^\pi + \lambda^{\star\top}_{\tau_t,\epsilon}(V_d^\pi - \epsilon_t - \alpha) + \tau_t \mathcal{H}(\pi) \leq V_r^{\pi^\star_{\tau_t,\epsilon}} + \lambda^{\star\top}_{\tau_t,\epsilon}(V_d^{\pi^\star_{\tau_t,\epsilon}} - \epsilon_t - \alpha) + \tau_t \mathcal{H}(\pi^\star_{\tau_t,\epsilon})$

*(ii)* $\quad \lambda^{\star\top}_{\tau_t,\epsilon}(V_d^{\pi^\star_{\tau_t,\epsilon}} - \epsilon_t - \alpha) \leq \lambda^\top(V_d^{\pi^\star_{\tau_t,\epsilon}} - \epsilon_t - \alpha) + \frac{\tau_t}{2}(\|\lambda\|^2 - \|\lambda^\star_{\tau_t,\epsilon}\|^2)$

*Proof.* According to Lemma 9, we immediately obtain $\mathcal{L}_{\tau_t,t}(\pi, \lambda^\star_{\tau_t,\epsilon}) \leq \mathcal{L}_{\tau_t,t}(\pi^\star_{\tau_t,\epsilon}, \lambda^\star_{\tau_t,\epsilon}) \leq \mathcal{L}_{\tau_t,t}(\pi^\star_{\tau_t,\epsilon}, \lambda)$. The inequalities are derived by expanding the saddle point definition from $\mathcal{L}_{\tau_t,t}(\pi, \lambda^\star_{\tau_t,\epsilon}) \leq \mathcal{L}_{\tau_t,t}(\pi^\star_{\tau_t,\epsilon}, \lambda^\star_{\tau_t,\epsilon})$ and $\mathcal{L}_{\tau_t,t}(\pi^\star_{\tau_t,\epsilon}, \lambda^\star_{\tau_t,\epsilon}) \leq \mathcal{L}_{\tau_t,t}(\pi^\star_{\tau_t,\epsilon}, \lambda)$, respectively. $\qquad\square$

# D  PROPERTIES OF THE MODEL

**Estimators**  For each constraint $i \in [m]$, state $s$, action $a$, episode $t \in [T]$ and step $h \in [H]$, define $(s_h^l, a_h^l)$ as the state-action pair visited in episode $l$ at step $h$, and let $(s_h^l, a_h^l, s_{h+1}^l)$ denote the state-action pair $(s_h^l, a_h^l)$ is visited and the environment evolves to next state $s_{h+1}^l$ at step $h$ in episode $l$, let $\mathbf{1}_X$ represent the indicator function of $X$ and $N_h^t(s,a) = \sum_{l=1}^t \mathbf{1}_{\{s_h^l = s, a_h^l = a\}}$ is the total number of visits to the pair $(s,a) \in \mathcal{S} \times \mathcal{A}$ at step $h$ up to episode $t \in [T]$. We first give the empirical averages of the thresholds, rewards, constraints and transition probabilities as follows:

$$\hat{\alpha}_i^t := \frac{1}{tH}\sum_{l=1}^t \sum_{h=1}^H \tilde{\alpha}_{i,h}^l, \qquad\qquad (\forall i \in [m])$$

$$\hat{r}_h^t(s,a) := \frac{\sum_{l=1}^t \tilde{r}_h^l(s,a)\,\mathbf{1}_{\{s_h^l = s, a_h^l = a\}}}{\max\{1, N_h^t(s,a)\}},$$

$$\hat{d}_{i,h}^t(s,a) := \frac{\sum_{l=1}^t \tilde{d}_{i,h}^l(s,a)\,\mathbf{1}_{\{s_h^l = s, a_h^l = a\}}}{\max\{1, N_h^t(s,a)\}}, \qquad (\forall i \in [m])$$

$$\hat{p}_h^t(s' \mid s,a) := \frac{\sum_{l=1}^t \mathbf{1}_{\{s_h^l = s, a_h^l = a, s_{h+1}^l = s'\}}}{\max\{1, N_h^t(s,a)\}}.$$

Next, we define optimistic estimators for the reward, constraints and entropy bonus, and unbiased estimators for transition probabilities and thresholds as follows:

$$\overline{\alpha}_i^t := \hat{\alpha}_i^{t-1}, \tag{7a}$$

$$\overline{r}_h^t(s,a) := \hat{r}_h^{t-1}(s,a) + \phi_h^{t-1}(s,a), \tag{7b}$$

$$\overline{d}_{i,h}^t(s,a) := \hat{d}_{i,h}^{t-1}(s,a) + \phi_h^{t-1}(s,a), \tag{7c}$$

$$\overline{p}_h^t(s'|s,a) := \hat{p}_h^{t-1}(s'|s,a), \tag{7d}$$

$$\overline{\psi}_h^t(s,a) := \psi_h^t(s,a) + \phi_h^{p,t-1}(s,a)\log(A). \tag{7e}$$

The bonus term $\phi_h^t$ combines the uncertainties arising from both reward and transition estimations at step $h$ in episode $t$: $\phi_h^t(s,a) = \phi_h^{r,t}(s,a) + \phi_h^{p,t}(s,a)$, where the reward bonus $\phi_h^{r,t}(s,a) = \mathcal{O}\left(\sqrt{\frac{\ln(mSAHT/\delta')}{\max\{1, N_h^t(s,a)\}}}\right)$ and the transition bonus $\phi_h^{p,t}(s,a) = \mathcal{O}\left(H\sqrt{\frac{S + \ln(SAHT/\delta')}{\max\{1, N_h^t(s,a)\}}}\right)$ for any confidence parameter $\delta' \in (0,1)$.

For convenience, we deonte

$$y_t := r + \lambda_t^\top d + \tau_t \psi_t,$$
$$\bar{y}_t := \bar{r}_t + \lambda_t^\top \bar{d}_t + \tau_t \bar{\psi}_t.$$

**Success event** Fixing a confidence parameter $\delta > 0$ and defining $\delta' := \delta/4$, we first introduce the following *failure events*:

$$F_t^\alpha := \left\{ \exists i : \left| \hat{\alpha}_i^{t-1} - \alpha_i \right| \geq \zeta^{t-1} \right\},$$

$$F_t^r := \left\{ \exists s, a, h : \left| \hat{r}_h^{t-1}(s,a) - r_h(s,a) \right| \geq \phi_h^{r,t-1}(s,a) \right\},$$

$$F_t^d := \left\{ \exists s, a, h, i : \left| \hat{d}_{i,h}^{t-1}(s,a) - d_{i,h}(s,a) \right| \geq \phi_{i,h}^{r,t-1}(s,a) \right\},$$

$$F_t^p := \left\{ \exists s, a, h : \left\| p_h(\cdot \mid s,a) - \hat{p}_h^{t-1}(\cdot \mid s,a) \right\|_1 H \geq \phi_h^{p,t-1}(s,a) \right\},$$

$$F_t^N := \left\{ \exists s, a, h : N_h^{t-1}(s,a) \leq \frac{1}{2} \sum_{j<t} q_h^{\pi_j}(s,a) - H \log\left( \frac{SAH}{\delta'} \right) \right\}.$$

Then, we define the union of these events over all episodes,

$$F^\alpha := \bigcup_{t \in [T]} F_t^\alpha, \quad F^r := \left( \bigcup_{t \in [T]} F_t^r \right) \bigcup \left( \bigcup_{t \in [T]} F_t^d \right),$$

$$F^p := \bigcup_{t \in [T]} F_t^p, \quad F^N := \bigcup_{t \in [T]} F_t^N.$$

Furthermore, the success event $\mathcal{E}$ is defined as the complement of those failure events:

$$\mathcal{E} = \overline{F^\alpha \cup F^r \cup F^p \cup F^N}.$$

We have the following lemma.

**Lemma 11** (Success event). *Setting* $\delta' = \frac{\delta}{4}$, *we have* $\mathbb{P}[\mathcal{E}] \geq 1 - \delta$.

*Proof.* We apply the union bound to each event separately. By Lemma 4, we have $\mathbb{P}[F^\alpha] \leq \delta'$. Using Hoeffding's inequality and union bound arguments over all state-action-step combinations, similarly, we obtain $\mathbb{P}[F^r] \leq \delta'$. Using concentration inequalities for multinomial distributions ((Maurer & Pontil, 2009)) and the union bound, we derive $\mathbb{P}[F^p] \leq \delta'$. Employing similar techniques as in ((Dann et al., 2017)), by bounding occupancy measure deviations, we obtain $\mathbb{P}[F^N] \leq \delta'$.

Combining these results with the union bound, we have

$$\mathbb{P}[F^\alpha \cup F^r \cup F^p \cup F^N] \leq \mathbb{P}[F^\alpha] + \mathbb{P}[F^r] + \mathbb{P}[F^p] + \mathbb{P}[F^N] \leq 4\delta' = \delta.$$

Thus, $\mathbb{P}[\mathcal{E}] = 1 - \mathbb{P}[F^\alpha \cup F^r \cup F^p \cup F^N] \geq 1 - \delta$.

This completes the proof. $\qquad\square$

**Truncated policy evaluation** Truncated policy evaluation is essential in CMDPs under stochastic threshold settings. Given the presence of stochastic constraints and additional exploration bonuses, unbounded value estimates can lead to instability and hinder theoretical analysis. We employ truncation to maintain boundedness and numerical stability of value functions.

Formally, for given estimates of reward $\bar{r}_h(s,a)$, constraint functions $\bar{d}_{i,h}(s,a)$, transition probabilities $\bar{p}_h(\cdot \mid s,a)$, we iteratively compute truncated $Q$ and $V$ value estimates. The truncated Q-value update at each timestep $h$ is expressed as below,

$$\hat{Q}_h^t(s,a;\bar{l},\bar{p}) = \min\left\{\bar{l}_h(s,a) + \sum_{s'}\bar{p}_h(s' \mid s,a)\hat{V}_{\bar{l},h+1}^t(s'),\ H-h+1\right\},$$

where $\bar{l}_h(s,a)$ denotes the generalized immediate payoff (reward or cost with bonus), and $\hat{V}_h^\pi(s;\bar{l},\bar{p})$ denotes the truncated value function,

$$\hat{V}_{\bar{l},h}^t(s) = \left\langle \hat{Q}_h^t(s,a;\bar{l},\bar{p}), \pi_h^t(a \mid s)\right\rangle.$$

For composite variable $\bar{y}_t$, we define its truncated value function as follows:

$$\hat{Q}_{\bar{y}_t,h}^t(s,a) := \hat{Q}_{\bar{r}_t,h}^t(s,a) + \sum_{i=1}^m \lambda_{t,i}\hat{Q}_{\bar{d}_{i,t},h}^t(s,a) + \tau_t\hat{Q}_{\bar{\psi}_t,h}^t(s,a),$$

$$\hat{V}_{\bar{r},h}^t(s) = \langle \hat{Q}_h^t(s,\cdot;\bar{r},\bar{p}), \pi_h^t(\cdot \mid s)\rangle.$$

The detailed truncated policy evaluation algorithm is shown in Algorithm 1.

**Estimation error**  We next show bounds on the estimation error of empirical estimator $\bar{r}$, $\bar{d}$ and $\bar{\psi}$.

**Lemma 12** (Bound on the estimation error). *Let $T' \in [T]$ be a number of episodes. The total estimation error for the reward function and constraint function, conditioned on the success event $\mathcal{E}$, satisfies the following upper bound:*

$$\sum_{t=1}^{T'}(\hat{V}_{\bar{r}_t}^t - V_r^{\pi_t}) \le \left(2\sqrt{L_r} + 2H\sqrt{L_p}\right) \cdot \left(6HSA + 2H\sqrt{SAT'} + 2HSA\log(T') + 5\log\frac{2HT'}{\delta}\right),$$

$$\sum_{t=1}^{T'}(\hat{V}_{\bar{d}_{t,i}}^t - V_{d_i}^{\pi_t}) \le \left(2\sqrt{L_r} + 2H\sqrt{L_p}\right) \cdot \left(6HSA + 2H\sqrt{SAT'} + 2HSA\log(T') + 5\log\frac{2HT'}{\delta}\right),$$

$$\sum_{t=1}^{T'}(\hat{V}_{\bar{\psi}_t}^t - V_{\psi_t}^{\pi_t}) \le 2H\log(A)\sqrt{L_p}\left(6HSA + 2H\sqrt{SAT'} + 2HSA\log(T') + 5\log\frac{2HT'}{\delta}\right).$$

*where $L_r = \frac{1}{2}\log\left(\frac{2SAH(m+1)T}{\delta'}\right)$ and $L_p = 2S + 2\log\left(\frac{SAHT}{\delta'}\right)$.*

*Proof.* We first bound the total estimation error by the sum of the expectations of the bonus terms from Müller et al. (2024).

$$\sum_{t=1}^{T'}(\hat{V}_{\bar{r}_t}^t - V_r^{\pi_t}) \le 2\sum_{t=1}^{T'}\sum_{h=1}^H \mathbb{E}[\phi_h^{r,t-1}(s_h^t,a_h^t)] + 2\sum_{t=1}^{T'}\sum_{h=1}^H \mathbb{E}[\phi_h^{t-1,p}(s_h^t,a_h^t)].$$

By substituting the definitions of the bonus terms $b^r$ and $b^p$ and factoring out the shared summation structure, the total error is bounded by:

$$\sum_{t=1}^{T'}(\hat{V}_{\bar{r}_t}^t - V_r^{\pi_t}) \le \left(2\sqrt{L_r} + 2H\sqrt{L_p}\right)\sum_{t=1}^{T'}\sum_{h=1}^H \mathbb{E}\left[\frac{1}{\sqrt{n_{t-1,h}(s_h^t,a_h^t) \vee 1}}\right],$$

---

**Algorithm 3** TPE (Truncated Policy Evaluation)

---

**Require:** estimates $\bar{r}_h^t$, $\bar{d}_{i,h}^t$, $\bar{p}_h^t$, policy $\pi_h^t$.

1: Initial $\hat{V}_{\bar{r},H+1}^t(s) = \hat{V}_{\bar{d}_i,H+1}^t(s) = \hat{V}_{\bar{\psi},H+1}^t(s) = 0$ for all $s$, $i$.

2: **for** $h = H, H-1, \ldots, 1$ **do**

3:     **for** $(s,a) \in \mathcal{S} \times \mathcal{A}$ **do**

4:         **Compute truncated Q-function:**

5:         $\hat{Q}_{\bar{r}_t,h}^t(s,a) = \min\left\{\bar{r}_h(s,a) + \langle \bar{p}_h(\cdot \mid s,a)\hat{V}_{\bar{r}_t,h+1}^t(\cdot)\rangle, H-h+1\right\}$

6:         $\hat{Q}_{\bar{\psi}_t,h}^t(s,a) = \min\left\{\bar{\psi}_h^t(s,a) + \langle \bar{p}_h(\cdot \mid s,a)\hat{V}_{\bar{\psi}_t,h+1}^t(\cdot)\rangle, \psi_h^t(s,a) + (H-h+1)\log(A)\right\}$

7:         **for** $i = 1, \ldots, m$ **do**

8:             $\hat{Q}_{\bar{d}_{i,t},h}^t(s,a) = \min\left\{\bar{d}_{i,h}(s,a) + \langle \bar{p}_h(\cdot \mid s,a)\hat{V}_{\bar{d}_{i,t},h+1}^t(\cdot)\rangle, H-h+1\right\}$

9:         **end for**

10:     **end for**

11:     **for** all $s \in \mathcal{S}$ **do**

12:         **Compute truncated V-function:**

13:         $\hat{V}_{\bar{r},h}^t(s) = \langle\hat{Q}_{\bar{r}_t,h}^t(s,\cdot), \pi_h^t(\cdot \mid s)\rangle$

14:         $\hat{V}_{\bar{\psi},h}^t(s) = \langle\hat{Q}_{\bar{\psi}_t,h}^t(s,\cdot), \pi_h^t(\cdot \mid s)\rangle$

15:         **for** $i = 1, \ldots, m$ **do**

16:             $\hat{V}_{\bar{d}_i,h}^t(s) = \langle\hat{Q}_{\bar{d}_{i,t},h}^t(s,\cdot), \pi_h^t(\cdot \mid s)\rangle$

17:         **end for**

18:     **end for**

19: **end for**

20: **for** $h = 1, \ldots, H$ and all $(s,a)$ **do**

21:     $\hat{Q}_{\bar{y}_t,h}^t(s,a) := \hat{Q}_{\bar{r}_t,h}^t(s,a) + \sum_{i=1}^m \lambda_{t,i}\hat{Q}_{\bar{d}_{i,t},h}^t(s,a) + \tau_t\hat{Q}_{\bar{\psi}_t,h}^t(s,a)$

22: **end for**

23: **return** $\left\{\hat{Q}_{\bar{y}_t,h}^t(s,a)\right\}_{h,s,a}$ and $\left\{\hat{V}_{\bar{d}_{i,t},h}^t(s)\right\}_{s,h,i}$

---

where $L_r = \frac{1}{2}\log\left(\frac{2SAH(m+1)T}{\delta'}\right)$ and $L_p = 2S + 2\log\left(\frac{SAHT}{\delta'}\right)$. The core summation term involves the inverse square root of visitation counts. Using the high-probability bound for this term from Liu et al. (2021), it can be shown that:

$$\sum_{t=1}^{T'}\sum_{h=1}^{H}\mathbb{E}\left[\frac{1}{\sqrt{n_{t-1,h}(s_h^t,a_h^t) \vee 1}}\right] \leq 6HSA + 2H\sqrt{SAT'} + 2HSA\log(T') + 5\log\frac{2HT'}{\delta}.$$

To put all term together, we get our final result:

$$\sum_{t=1}^{T'}(\hat{V}_{\bar{r}_t}^t - V_r^{\pi_t}) \leq \left(2\sqrt{L_r} + 2H\sqrt{L_p}\right) \cdot \left(6HSA + 2H\sqrt{SAT'} + 2HSA\log(T') + 5\log\frac{2HT'}{\delta}\right).$$

The proof for $d_i$ $(\forall i \in [m])$ is identical. For entropy bonus, we have

$$\sum_{t=1}^{T'}(\hat{V}_{\bar{\psi}_t}^t - V_{\psi_t}^{\pi_t}) \leq 2\sum_{t=1}^{T'}\sum_{h=1}^{H}\mathbb{E}\left[\phi_h^{t-1,p}(s_h^t,a_h^t)\log(A)\right].$$

and the rest of the proof follows as in the proof of the case of reward function. $\square$

**Lemma 13** (Bound on cumulative estimation discrepancies). *Conditioned on the good event $\mathcal{E}$, for any episode $T' \in [T]$, the cumulative sum of the per-episode estimation discrepancies $\delta_t$ is bounded as follows:*

$$\sum_{i=1}^{T'} \delta_i \leq C_B \sqrt{T' \log \frac{SAHT}{\delta'}} + \tilde{O}(S^{3/2}AH^2).$$

*where $C_B = \left(1 + \frac{8mH}{\Xi}\right)\left(4H\sqrt{2SA}\left(H\sqrt{S} + H + 1\right)\right) + \frac{4mH}{\Xi}\sqrt{2H}$ and $\delta_t$ is defined as the composite error at episode $t$:*

$$\delta_t := \left(\hat{V}_{\bar{r}_t}^t - V_r^{\pi_t}\right) + \sum_{i=1}^{m} \lambda_{t,i}\left(\hat{V}_{\tilde{d}_{t,i}}^t - V_{d_i}^{\pi_t}\right) + \tau_t\left(\hat{V}_{\tilde{\psi}_t}^t - V_{\psi_t}^{\pi_t}\right) + \sum_{i=1}^{m} \lambda_{t,i}\left|\hat{\alpha}_i^t - \alpha_i\right| + \sum_{i=1}^{m} \frac{4H}{\Xi}\left(\hat{V}_{\tilde{d}_{t,i}}^t - V_{d_i}^{\pi_t}\right).$$

*Proof.* The proof proceeds by decomposing the total sum $\sum_{t=1}^{T'} \delta_t$ and bounding each component term individually. Conditioned on the good event $\mathcal{E}$, we have:

$$\sum_{t=1}^{T'} \delta_t \leq \underbrace{\sum_{t=1}^{T'}\left(\hat{V}_{\bar{r}_t}^t - V_r^{\pi_t}\right)}_{(A)} + \underbrace{\sum_{t=1}^{T'}\sum_{i=1}^{m} \frac{8H}{\Xi}\left(\hat{V}_{\tilde{d}_{t,i}}^t - V_{d_i}^{\pi_t}\right)}_{(B)} + \underbrace{\sum_{t=1}^{T'} \tau_t\left(\hat{V}_{\tilde{\psi}_t}^t - V_{\psi_t}^{\pi_t}\right)}_{(C)} + \underbrace{\sum_{t=1}^{T'}\sum_{i=1}^{m} \lambda_{t,i}\left|\hat{\alpha}_i^t - \alpha_i\right|}_{(D)}$$

$$(8)$$

We bound each of the four terms on the right-hand side of Equation equation 8.

**Bounding terms (A), (B), and (C):** These terms represent the cumulative estimation errors for the value functions of the reward, constraints, and the policy entropy proxy $\psi_t$, respectively. We can bound them by leveraging Lemma 12.

For term (A), using Lemma 12 and inequality $\sqrt{a+b} \leq \sqrt{a} + \sqrt{b}$, it yields:

$$(A) \leq \left(\sqrt{2\log\frac{2SAH(m+1)T}{\delta'}} + 2H\sqrt{2S} + 2H\sqrt{2\log\frac{SAHT}{\delta'}}\right) \cdot \left(2H\sqrt{SAT'}\right) + \tilde{O}(S^{3/2}AH^2)$$

$$\leq \left(\sqrt{2\log(2(m+1))} + 2H\sqrt{2S} + (1+2H)\sqrt{2\log\frac{SAHT}{\delta'}}\right) \cdot \left(2H\sqrt{SAT'}\right) + \tilde{O}(S^{3/2}AH^2)$$

$$\leq \left(4H(H+1)\sqrt{2SA} + 4H^2S\sqrt{2A}\right)\sqrt{T'\log\frac{SAHT}{\delta'}} + \tilde{O}(S^{3/2}AH^2)$$

For term (B), we use the fact that the dual variables are bounded, i.e., $\lambda_{t,i} \leq \left(\frac{4H}{\Xi}\right)$ for all $i \in [m]$. This allows us to write:

$$(B) \leq \frac{8H}{\Xi}\sum_{i=1}^{m}\left(\sum_{t=1}^{T'}\left(\hat{V}_{\tilde{d}_{t,i}}^t - V_{d_i}^{\pi_t}\right)\right)$$

$$\leq \frac{8mH}{\Xi}\left(4H(H+1)\sqrt{2SA} + 4H^2S\sqrt{2A}\right)\sqrt{T'\log\frac{SAHT}{\delta'}} + \tilde{O}(S^{3/2}AH^2).$$

For term (C), we apply the bound from Lemma 12:

$$(C) = \sum_{t=1}^{T'} \tau_t \left( \hat{V}_{\hat{\psi}_t}^t - V_{\psi_t}^{\pi_t} \right)$$

$$\leq \max_t \{\tau_t\} \cdot \left( 4H(H+1)\sqrt{2SA} + 4H^2 S\sqrt{2A} \right) \sqrt{T' \log \frac{SAHT}{\delta'}} + \tilde{O}(S^{3/2}AH^2)$$

$$\leq \left( 4H(H+1)\sqrt{2SA} + 4H^2 S\sqrt{2A} \right) \sqrt{T' \log \frac{SAHT}{\delta'}} + \tilde{O}(S^{3/2}AH^2)$$

**Bounding term (D):** This term represents the cumulative error from the online estimation of the stochastic thresholds. Based on our derivation from Theorem 2, we have established a high-probability bound for this sum:

$$(D) = \sum_{t=1}^{T'} \sum_{i=1}^{m} \lambda_{t,i} \left| \hat{\alpha}_i^t - \alpha_i \right| \leq m \left( \frac{4H}{\Xi} \right) \sqrt{2H \log(2mT/\delta)T'}.$$

**Combining all terms:** By substituting the bounds for (A), (B), (C), and (D) back into Equation equation 8, we obtain the final upper bound for the cumulative discrepancy:

$$\sum_{i=1}^{T'} \delta_i \leq C_B \sqrt{T' \log \frac{SAHT}{\delta'}} + \tilde{O}(S^{3/2}AH^2)$$

where $C_B = \left(1 + \frac{8mH}{\Xi}\right) \left( 4H\sqrt{2SA} \left( H\sqrt{S} + H + 1 \right) \right) + \frac{4mH}{\Xi} \sqrt{2H}$. This completes the proof. $\qquad\square$

$Q$**-Value function bounds** We present the bounds for $Q$-value.

**Lemma 14** (Müller et al. (2024)). *For every state $s$, action $a$, step $h$, horizon $H$, and policy $\pi_t$ at $t$-th episode, it holds that*

$$\mathbb{E}\left[ \sum_{h'=h}^{H} -\log\big(\pi_{t,h'}(a_{h'} \,|\, s_{h'})\big) \mid s_h = s, a_h = a \right] \leq H \log(A) - \log(\pi_{t,h}(a \,|\, s)).$$

**Lemma 15** ($Q$-Value function bounds). *For any state $s$, action $a$, step $h$, we get*

$$0 \leq Q_{r+\lambda_t^\top d + \tau_t \psi_t, h}^{\pi_t}(s, a) \leq -\tau_t \log(\pi_{t,h}(a \mid s)) + H(1 + \frac{4mH}{\Xi} + \tau_t \log(A))$$

*Moreover, we have*

$$\sum_a \pi_{t,h}(a \,|\, s) \exp\left( \eta_t Q_{r+\lambda_t^\top d + \tau_t \psi_t, h}^{\pi_t}(s, a) \right) Q_{r+\lambda_t^\top d + \tau_t \psi_t, h}^{\pi_t}(s, a)^2$$

$$\leq \exp\left( \eta_t H \left(1 + \frac{4mH}{\Xi} + \tau_t \log(A)\right) \right) \left( 2A^{\eta_t \tau_t} H^2 \left(1 + \frac{4mH}{\Xi} + \tau_t \log(A)\right)^2 + \frac{128\tau_t^2 \sqrt{A}}{e^2} \right).$$

*Proof.* We first prove the bound for $Q_{y_t}^{\pi_t}$, where $y_t = r + \lambda_t^\top d + \tau_t \psi_t$. For all $s, a, h$, we have

$$0 \leq Q_{r + \lambda_t^\top d + \tau_t \psi_t, h}^{\pi_t}(s, a) \leq \left| Q_{r,h}^{\pi_t}(s, a) \right| + \sum_i \lambda_{i,t} \left| Q_{d_{i,t},h}^{\pi_t} \right| + \tau_t \left| Q_{\psi_t, h}^{\pi_t} \right|$$

$$\leq H + m \left( \frac{4H}{\Xi} \right) H + \tau_t \mathbb{E} \left[ \sum_{h'=h}^{H} - \log\left( \pi_{t,h'}(a_{h'} \mid s_{h'}) \right) \mid s_h = s, a_h = a \right]$$

$$\leq \underbrace{H \left( 1 + m \left( \frac{4H}{\Xi} \right) + \tau_t \log(A) \right)}_{C_0} - \tau_t \log(\pi_{t,h}(a \mid s)).$$

The last inequality holds by Lemma 14. According to the definitions, we have $y_{t,h}(s, a) = r_{t,h}(s, a) + \lambda_t^\top d_{t,h}(s, a) + \tau_t \psi_{t,h}(s, a)$, where $\psi_{t,h}(s, a) = - \log(\pi_{t,h}(a \mid s))$. Moreover, according to Euclidean triangle inequality $(a + b)^2 \leq 2a^2 + 2b^2$, we can obtain

$$Q_y^{\pi_t}(s, a)^2 \leq \underbrace{2H^2(1 + m \left( \frac{4H}{\Xi} \right) + \tau_t \log(A))^2}_{C_1} + 2\tau_t^2 \log^2 \left( \frac{1}{\pi_{t,h}(a \mid s)} \right).$$

We then get the following inequality:

$$\sum_a \pi_{t,h}(a \mid s) \exp\left( \eta_t Q_{y_t}^{\pi_t}(s, a) \right) Q_{y_t}^{\pi_t}(s, a)^2 \leq \underbrace{\sum_a \pi_{t,h}(a \mid s) \exp\left( \eta_t Q_{y_t}^{\pi_t}(s, a) \right) C_1}_{(1)}$$

$$+ \underbrace{\sum_a \pi_{t,h}(a \mid s) \exp\left( \eta_t Q_{y_t}^{\pi_t}(s, a) \right) 2\tau_t^2 \log^2 \left( \frac{1}{\pi_{t,h}(a \mid s)} \right)}_{(2)}.$$

For term (1) on the right-side, we first show

$$\pi_{t,h}(a \mid s) \exp\left( \eta_t Q_{y_t}^{\pi_t}(s, a) \right) \leq \pi_{t,h}(a \mid s) \exp\left( \eta_t C_0 - \eta_t \tau_t \log(\pi_{t,h}(a \mid s)) \right)$$

$$= \pi_{t,h}(a \mid s)^{1 - \eta_t \tau_t} \exp\left( \eta_t C_0 \right).$$

Thus, we obatin

$$(1) \leq \sum_a C_1 \pi_{t,h}(a \mid s)^{1 - \eta_t \tau_t} \exp\left( \eta_t C_0 \right)$$

$$\leq A^{\eta_t \tau_t} \exp(\eta_t C_0) C_1$$

$$= A^{\eta_t \tau_t} \exp\left( \eta_t H \left( 1 + \frac{4mH}{\Xi} + \tau_t \log(A) \right) \right) \left( 2H^2 \left( 1 + \frac{4mH}{\Xi} + \tau_t \log(A) \right)^2 \right). \quad (9)$$

The second inequality holds because $\sum_a \pi_{t,h}(a \mid s)^{1 - \eta_t \tau_t} \leq \max_\pi \sum_a \pi_{t,h}(a \mid s)^{1 - \eta_t \tau_t} \leq A^{\eta_t \tau_t}$. This is because the extreme case is the uniform distribution. Furthermore, for term (2), we follow the analysis in Müller et al. (2024). Assuming $\eta_t \tau_t \leq 1/2$, we have $\pi_{t,h}(a \mid s)^{1 - \eta_t \tau_t} \leq \pi_{t,h}(a \mid s)^{1/2}$. We then use the property that $q^{1/4} \log^2(1/q)$ is universally bounded by $64/e^2$ for any $q$ and apply the Cauchy-Schwarz

inequality to the remaining sum. This yields:

$$(2) \leq \sum_a \pi_{t,h}(a \,|\, s)^{1-\eta_t \tau_t} \exp(\eta_t C_0) 2\tau_t^2 \log^2\left(\frac{1}{\pi_{t,h}(a \,|\, s)}\right)$$

$$= \exp(\eta_t C_0) 2\tau_t^2 \sum_a \pi_{t,h}(a \,|\, s)^{1-\eta_t \tau_t} \log^2\left(\frac{1}{\pi_{t,h}(a \,|\, s)}\right)$$

$$\leq \exp(\eta_t C_0) 2\tau_t^2 \left(\frac{64\sqrt{A}}{e^2}\right)$$

$$= \frac{128\tau_t^2 \sqrt{A}}{e^2} \exp\left(\eta_t C_0\right). \tag{10}$$

Combining equation 9 and equation 10, we have

$$\sum_a \pi_{t,h}(a \,|\, s) \exp\left(\eta_t Q_{y_t}^{\pi_t}(s,a)\right) Q_{y_t}^{\pi_t}(s,a)^2$$

$$\leq \exp\left(\eta_t H \left(1 + \frac{4mH}{\Xi} + \tau_t \log(A)\right)\right) \left(2A^{\eta_t \tau_t} H^2 \left(1 + \frac{4mH}{\Xi} + \tau_t \log(A)\right)^2 + \frac{128\tau_t^2 \sqrt{A}}{e^2}\right).$$

$$\square$$

The bounds established for $Q$ also apply to the truncated $\hat{Q}$. This is because $\hat{Q}_{y_t,h} \leq H - h + 1$ by definition, and is less than or equal to the initial bound $C_0 - \tau_t \log(\pi_{t,h}(a|s))$ used in the proof above. We express the result as follows.

**Lemma 16** ($Q$-Value function bounds). *For any state $s$, action $a$, step $h$, we get*

$$0 \leq \hat{Q}_{\bar{y}_t,h}^t(s,a) \leq -\tau_t \log(\pi_{t,h}(a \,|\, s)) + H(1 + \frac{4mH}{\Xi} + \tau_t \log(A)).$$

*Moreover, we have*

$$\sum_a \pi_{t,h}(a \,|\, s) \exp\left(\eta_t \hat{Q}_{\bar{y}_t,h}^t(s,a)\right) \hat{Q}_{\bar{y}_t,h}^t(s,a)^2$$

$$\leq \exp\left(\eta_t H \left(1 + \frac{4mH}{\Xi} + \tau_t \log(A)\right)\right) \left(2A^{\eta_t \tau_t} H^2 \left(1 + \frac{4mH}{\Xi} + \tau_t \log(A)\right)^2 + \frac{128\tau_t^2 \sqrt{A}}{e^2}\right).$$

**Error summation bounds**    To establish our main regret bounds, we need carefully analyze the cumulative effect of two primary sources of error that arise from our learning algorithm: the optimization error stemming from the primal-dual updates, and the statistical error resulting from estimating the unknown CMDP model. The following two lemmas provide crucial bounds on summations that capture the behavior of these error terms over time.

**Lemma 17** (Bound on the optimization error). *Let $\eta_t = t^{-3/4}$ and $\tau_t = t^{-1/8}$ for $t \geq 1$. Let $C_0 = \sqrt{HC + D}$. Then for any $T \geq C''$, the following inequality holds:*

$$\sum_{t=C''}^{T} H^{3/2} \sqrt{HC + D} \left(\sum_{j=1}^{t} \eta_j^2 \exp\left(-\sum_{k=j+1}^{t} \eta_k \tau_k\right)\right)^{1/2} \leq K \cdot T^{11/16},$$

*where $K = \frac{16}{11} H^{3/2} \sqrt{HC + D} \sqrt{\zeta(3/2) + 4\sqrt{2}}$.*

*Proof.* Let $S$ denote the sum. We factor out the constants and define the inner term $X_t$

$$S = H^{3/2} C_0 \sum_{t=C''}^{T} X_t^{1/2}, \quad \text{where} \quad X_t = \sum_{j=1}^{t} j^{-3/2} \exp\left(-\sum_{k=j+1}^{t} k^{-7/8}\right).$$

To bound $X_t$, we split the sum over $j$ into two parts: $j \in [1, \lfloor t/2 \rfloor]$ and $j \in [\lfloor t/2 \rfloor + 1, t]$.

For $j \in [1, \lfloor t/2 \rfloor]$, the sum in the exponent is large. We can bound it from below by integrating over the second half of the range: $\sum_{k=j+1}^{t} k^{-7/8} \geq \int_{t/2+1}^{t+1} x^{-7/8} dx = 8((t+1)^{1/8} - (t/2+1)^{1/8}) \geq c_1 t^{1/8}$ for a constant $c_1 = 8(1 - 2^{-1/8})$. This is thus bounded by:

$$\sum_{j=1}^{\lfloor t/2 \rfloor} j^{-3/2} e^{-c_1 t^{1/8}} \leq e^{-c_1 t^{1/8}} \sum_{j=1}^{\infty} j^{-3/2} = \zeta(3/2) e^{-c_1 t^{1/8}}.$$

This term decays exponentially with $t$.

For $j \in [\lfloor t/2 \rfloor + 1, t]$, we have $j^{-3/2} \leq (t/2)^{-3/2} = 2^{3/2} t^{-3/2}$. We find a lower bound for the exponent's sum: $\sum_{k=j+1}^{t} k^{-7/8} \geq (t-j) t^{-7/8}$. This gives the bound on the sum for this part:

$$\sum_{j=\lfloor t/2 \rfloor + 1}^{t} 2^{3/2} t^{-3/2} \exp\left(-(t-j) t^{-7/8}\right).$$

Letting $l = t - j$, this becomes $2^{3/2} t^{-3/2} \sum_{l=0}^{\lceil t/2 \rceil - 1} (e^{-t^{-7/8}})^l$. For large $t$ (guaranteed by the assumption on $C''$), $t^{-7/8}$ is small. Using the inequality $1 - e^{-x} \geq x/2$ for sufficiently small $x > 0$, we bound the sum of the series by $\frac{1}{1 - e^{-t^{-7/8}}} \leq \frac{1}{(1/2) t^{-7/8}} = 2t^{7/8}$. The bound for this second part is therefore:

$$\sum_{j=\lfloor t/2 \rfloor + 1}^{t} j^{-3/2} \exp\left(-\sum_{k=j+1}^{t} k^{-7/8}\right) \leq 2^{3/2} t^{-3/2} \cdot 2 t^{7/8} = 4\sqrt{2} \cdot t^{-5/8}.$$

Combining the two parts, $X_t \leq \zeta(3/2) e^{-c_1 t^{1/8}} + 4\sqrt{2} t^{-5/8}$. Since the exponential term decays faster than any power law, for $t \geq C''$ we can define a constant $K_X = \zeta(3/2) + 4\sqrt{2}$ such that $X_t \leq K_X t^{-5/8}$. Consequently, $X_t^{1/2} \leq \sqrt{K_X} t^{-5/16}$.

Substituting this back into the expression for $S$:

$$S \leq H^{3/2} C_0 \sqrt{K_X} \sum_{t=C''}^{T} t^{-5/16}.$$

The sum is for a p-series with $p = 5/16$, which we bound with an integral:

$$\sum_{t=C''}^{T} t^{-5/16} \leq \int_{C''-1}^{T} x^{-5/16} dx \leq \frac{16}{11} T^{11/16}.$$

Combining all terms yields the final bound:

$$S \leq H^{3/2} C_0 \sqrt{K_X} \left(\frac{16}{11} T^{11/16}\right) = \frac{16}{11} H^{3/2} \sqrt{HC + D} \sqrt{\zeta(3/2) + 4\sqrt{2}} \cdot T^{11/16}.$$

$\square$

**Lemma 18** (Bound on the statistical error). *Let $\eta_t = t^{-3/4}$ and $\tau_t = t^{-1/8}$ for $t \geq 1$. Let $\{\delta_t\}_{t=1}^T$ be a sequence whose partial sums are bounded by $\sum_{i=1}^{T'} \delta_i \leq C_B \sqrt{T' \log \frac{SAHT}{\delta'}} + \tilde{O}(S^{3/2}AH^2) := B(T')$, where $C_B = \left(1 + \frac{8mH}{\Xi}\right)\left(4H\sqrt{2SA}\left(H\sqrt{S} + H + 1\right)\right) + \frac{4mH}{\Xi}\sqrt{2H}$. Then for any $t \geq C''$, the following inequality holds:*

$$\sum_{t=C''}^T \sqrt{2H^3} \left(\sum_{j=1}^t \eta_j \delta_j \exp\left(-\sum_{k=j+1}^t \eta_k \tau_k\right)\right)^{1/2} \leq \tilde{O}(T^{7/8}),$$

*Proof.* Let $S_1$ denote the sum. We can write it as $S_1 = \sqrt{2H^3} \sum_{t=C''}^T \sqrt{|Y_t|}$, where the inner term $Y_t$ is defined as:

$$Y_t = \sum_{j=1}^t \eta_j \delta_j \exp\left(-\sum_{k=j+1}^t \eta_k \tau_k\right).$$

The presence of the sequence $\delta_j$, for which we only have a bound on its cumulative sum, necessitates the use of summation by parts. Let $f_{t,j} = \eta_j \exp\left(-\sum_{k=j+1}^t \eta_k \tau_k\right)$ and $\Delta_j = \sum_{i=1}^j \delta_i$. A critical property for this method is the monotonicity of $f_{t,j}$ with respect to $j$. We examine the ratio:

$$\frac{f_{t,j+1}}{f_{t,j}} = \frac{\eta_{j+1}}{\eta_j} \exp(\eta_{j+1}\tau_{j+1}).$$

Based on the choice of $\eta_j$ and $\tau_j$, this ratio is greater than 1 for all $j \geq 1$. Thus, $f_{t,j}$ is strictly increasing in $j$. Applying the summation by parts formula to $Y_t = \sum_{j=1}^t f_{t,j}(\Delta_j - \Delta_{j-1})$ (with $\Delta_0 = 0$):

$$Y_t = f_{t,t}\Delta_t - \sum_{j=1}^{t-1} \Delta_j(f_{t,j+1} - f_{t,j}).$$

Using the triangle inequality and the established monotonicity ($f_{t,j+1} - f_{t,j} \geq 0$):

$$|Y_t| \leq |f_{t,t}||\Delta_t| + \sum_{j=1}^{t-1} |\Delta_j|(f_{t,j+1} - f_{t,j}).$$

We use the given bound $|\Delta_j| \leq B(j)$. Since $B(j)$ is an increasing function of $j$, we can bound $|\Delta_j| \leq B(t-1)$ for all terms inside the summation. The sum is a telescoping series equal to $f_{t,t} - f_{t,1}$. Since $B(t-1) < B(t)$ and $f_{t,1} > 0$:

$$|Y_t| \leq f_{t,t}B(t) + B(t-1)(f_{t,t} - f_{t,1}) \leq f_{t,t}B(t) + B(t)f_{t,t} = 2f_{t,t}B(t).$$

For $t \geq C''$, we have $f_{t,t} = \eta_t = t^{-3/4}$. Substituting the form for the bound $B(t)$:

$$|Y_t| \leq 2t^{-3/4}B(t) = 2C_B t^{-1/4}\sqrt{\log \frac{SAHT}{\delta'}} + \tilde{O}(S^{3/2}AH^2) \cdot t^{-3/4}.$$

Taking the square root and factoring out the dominant term and using inequality $\sqrt{a+b} \leq \sqrt{a} + \sqrt{b}$, we obtain a bound for $\sqrt{|Y_t|}$:

$$
\sqrt{|Y_t|} \leq \sqrt{2C_B t^{-1/4} \sqrt{\log \frac{SAHT}{\delta'}} + \tilde{O}(S^{3/2}AH^2) \cdot t^{-3/4}}
$$

$$
\leq \sqrt{2C_B t^{-1/4} \sqrt{\log \frac{SAHT}{\delta'}}} + \sqrt{\tilde{O}(S^{3/2}AH^2)t^{-3/4}}
$$

$$
= \sqrt{2C_B} \cdot t^{-1/8} \left(\log \frac{SAHT}{\delta'}\right)^{1/4} + \tilde{O}(S^{3/4}A^{1/2}H) \cdot t^{-3/8}.
$$

The main sum is thus bounded by:

$$
S_1 \leq \sqrt{8H^3 C_B} \left(\log \frac{SAHT}{\delta'}\right)^{1/4} \cdot \sum_{t=C''}^{T} t^{-1/8} + \tilde{O}(S^{3/4}A^{1/2}H) \cdot \sum_{t=C''}^{T} t^{-3/8},
$$

$$
\leq \frac{16}{7} \sqrt{2H^3 C_B} \left(\log \frac{SAHT}{\delta'}\right)^{1/4} \cdot T^{7/8} + \tilde{O}(S^{3/4}A^{1/2}H) \cdot T^{5/8},
$$

$$
\leq \tilde{O}(T^{7/8}).
$$

$\square$

# E    REGRETS OF REWARD AND CONSTRAINT VIOLATIONS

In this section, we prove the bounds for the strong reward regret and constraint violation of Algorithm 1. We first establish the linear convergence of the primal-dual divergence potential functions.

**Lemma 1** (Margin-regularized convergence). *Let $\eta_t$, $\tau_t < 1$ and a confidence parameter $\delta \in (0,1)$. With probability at least $1 - \delta$, the policy-dual divergence potential of Algorithm 1 holds*

$$
\Phi_{t+1} \leq \exp\left(-\sum_{j=1}^{t} \eta_j \tau_j\right) \Phi_1 + \frac{HC+D}{2} \sum_{j=1}^{t} \eta_j^2 \exp\left(-\sum_{k=j+1}^{t} \eta_k \tau_k\right) + \sum_{j=1}^{t} \eta_j \delta_j \exp\left(-\sum_{k=j+1}^{t} \eta_k \tau_k\right).
$$

*where $C = \exp\left(\eta_t H\left(1 + \frac{4mH}{\Xi} + \tau_t \log(A)\right)\right)\left(2A^{\eta_t \tau_t}H^2\left(1 + \frac{4mH}{\Xi} + \tau_t \log(A)\right)^2 + \frac{128\tau_t^2 \sqrt{A}}{e^2}\right)$,*
*$D = m\left(H + \tau_t\left(\frac{4H}{\Xi}\right)\right)^2$ and $\delta_j = \hat{V}_{\bar{y}_j}^j - V_{y_j}^{\pi_j} + \sum_i \frac{4H}{\Xi}\left(\hat{V}_{\bar{d}_{j,i}}^j - V_{d_i}^{\pi_j}\right)$.*

*Proof.* Conditioned on success event $\mathcal{E}$, we first decompose the primal dual gap for every episode $t$:

$$
\mathcal{L}_{\tau_t,t}(\pi_{\tau_t,\epsilon}^\star, \lambda_t) - \mathcal{L}_{\tau_t,t}(\pi_t, \lambda_{\tau_t,\epsilon}^\star) = \underbrace{\mathcal{L}_{\tau_t,t}(\pi_{\tau_t,\epsilon}^\star, \lambda_t) - \mathcal{L}_{\tau_t,t}(\pi_t, \lambda_t)}_{(1)} + \underbrace{\mathcal{L}_{\tau_t,t}(\pi_t, \lambda_t) - \mathcal{L}_{\tau_t,t}(\pi_t, \lambda_{\tau_t,\epsilon}^\star)}_{(2)}.
$$

**Bounding term (1)**  For term (1), by Lemmas 5 and 15, we have

$$
\begin{aligned}
(1) &= \mathcal{L}_{\tau_t,t}\big(\pi^\star_{\tau_{t,\epsilon}}, \lambda_t\big) - \mathcal{L}_{\tau_t,t}\big(\pi_t, \lambda_t\big) \\
&= V^{\pi^\star_{\tau_{t,\epsilon}}}_{r+\lambda_t^T g} - V^{\pi_t}_{r+\lambda_t^T g} \qquad\qquad\qquad\qquad\qquad\qquad (g = d - \tfrac{1}{H}\alpha) \\
&\quad + \tau_t \sum_{s,a,h} q^{\pi_t}_h(s)\pi_{t,h}(a\,|\,s) \log\big(\pi_{t,h}(a\,|\,s)\big) - \tau_t \sum_{s,a,h} q^{\pi^\star_{\tau_{t,\epsilon}}}_h(s)\pi^\star_{\tau_{t,\epsilon},h}(a\,|\,s) \log\big(\pi^\star_{\tau_{t,\epsilon},h}(a\,|\,s)\big) \\
&= V^{\pi^\star_{\tau_{t,\epsilon}}}_{r+\lambda_t^T g + \tau_t\psi_t} - V^{\pi_t}_{r+\lambda_t^T g + \tau_t\psi_t} \\
&\quad + \tau_t \sum_{s,a,h} q^{\pi^\star_{\tau_{t,\epsilon}}}_h(s)\pi^\star_{\tau_{t,\epsilon},h}(a\,|\,s) \log\big(\pi_{t,h}(a\,|\,s)\big) - \tau_t \sum_{s,a,h} q^{\pi^\star_{\tau_{t,\epsilon}}}_h(s)\pi^\star_{\tau_{t,\epsilon},h}(a\,|\,s) \log\big(\pi^\star_{\tau_{t,\epsilon},h}(a\,|\,s)\big) \\
&= V^{\pi^\star_{\tau_{t,\epsilon}}}_{r+\lambda_t^T g + \tau_t\psi_t} - V^{\pi_t}_{r+\lambda_t^T g + \tau_t\psi_t} - \tau_t \mathrm{KL}_t \quad\; (\mathrm{KL}_t = \sum_{s,a,h} q^{\pi^\star_{\tau_{t,\epsilon}}}_h(s)\pi^\star_{\tau_{t,\epsilon},h}(a\,|\,s) \log\big(\tfrac{\pi^\star_{\tau_{t,\epsilon},h}(a|s)}{\pi_{t,h}(a|s)}\big)) \\
&= V^{\pi^\star_{\tau_{t,\epsilon}}}_{y_t} - V^{\pi_t}_{y_t} - \tau_t \mathrm{KL}_t \\
&\le (\hat V^t_{\bar y_t} - V^{\pi_t}_{y_t}) + \frac{\mathrm{KL}_t - \mathrm{KL}_{t+1}}{\eta_t} + \frac{\eta_t H}{2} C - \tau_t \mathrm{KL}_t \\
&\le (\hat V^t_{\bar y_t} - V^{\pi_t}_{y_t}) + \frac{(1-\eta_t\tau_t)\mathrm{KL}_t - \mathrm{KL}_{t+1}}{\eta_t} + \frac{\eta_t H}{2} C, \qquad\qquad\qquad (11)
\end{aligned}
$$

where $C = \exp\big(\eta_t H\big(1 + \frac{4mH}{\Xi} + \tau_t \log(A)\big)\big)\Big(2A^{\eta_t\tau_t}H^2\big(1 + \frac{4mH}{\Xi} + \tau_t \log(A)\big)^2 + \frac{128\tau_t^2\sqrt{A}}{e^2}\Big)$.

**Bounding term (2)**

$$
\begin{aligned}
(2) &= \sum_i (\lambda_{t,i} - \lambda^\star_{\tau_{t,\epsilon},i})(\hat V^t_{\bar d_{t,i}} - \alpha_i - \epsilon_t + \tau_t\lambda_{t,i}) + \sum_i (\lambda_{t,i} - \lambda^\star_{\tau_{t,\epsilon},i})(V^{\pi_t}_{d_i} - \hat V^t_{\bar d_{t,i}}) - \frac{\tau_t}{2}\|\lambda_t - \lambda^\star_{\tau_{t,\epsilon}}\|^2 \\
&\le \frac{\|\lambda^\star_{\tau_{t,\epsilon}} - \lambda_t\|^2 - \|\lambda^\star_{\tau_{t,\epsilon}} - \lambda_{t+1}\|^2}{2\eta_t} + \frac{\eta_t}{2}\|\hat V^t_{\bar d} - \alpha - \epsilon_t + \tau_t\lambda_t\|^2 \\
&\quad + \sum_i \left(\frac{4H}{\Xi}\right)\left|\hat V^t_{\bar d_{t,i}} - V^{\pi_t}_{d_i}\right| - \frac{\tau_t}{2}\|\lambda_t - \lambda^\star_{\tau_{t,\epsilon}}\|^2 \\
&= \frac{(1-\eta_t\tau_t)\|\lambda^\star_{\tau_{t,\epsilon}} - \lambda_t\|^2 - \|\lambda^\star_{\tau_{t,\epsilon}} - \lambda_{t+1}\|^2}{2\eta_t} + \frac{\eta_t}{2} D + \sum_i \left(\frac{4H}{\Xi}\right)\left|\hat V^t_{\bar d_{t,i}} - V^{\pi_t}_{d_i}\right|, \qquad (12)
\end{aligned}
$$

where $D \le m(H + \tau_t\big(\frac{4H}{\Xi}\big))^2$. Because $\Phi_t = \mathrm{KL}_t + \frac{1}{2}\|\lambda_t - \lambda^\star_{\tau_{t,\epsilon}}\|^2$, combining equation 11 and equation 12, it holds

$$
\Phi_{t+1} \le (1-\eta_t\tau_t)\Phi_t + \frac{\eta_t^2}{2}(HC + D) + \underbrace{\eta_t\left(\hat V^t_{\bar y_t} - V^{\pi_t}_{y_t} + \sum_i \left(\frac{4H}{\Xi}\right)\left|\hat V^t_{\bar d_{t,i}} - V^{\pi_t}_{d_i}\right|\right)}_{:=\eta_t\delta_t}.
$$

Applying the recursion inductively, we get

$$
\Phi_{t+1} \leq \left( \prod_{j=1}^{t} (1 - \eta_j \tau_j) \right) \Phi_1 + \sum_{j=1}^{t} \left[ \left( \frac{\eta_j^2}{2} (HC + D) + \eta_j \delta_j \right) \left( \prod_{k=j+1}^{t} (1 - \eta_k \tau_k) \right) \right]
$$

$$
\leq \left( \prod_{j=1}^{t} (1 - \eta_j \tau_j) \right) \Phi_1 + \sum_{j=1}^{t} \left[ \frac{\eta_j^2}{2} (HC + D) \left( \prod_{k=j+1}^{t} (1 - \eta_k \tau_k) \right) \right]
$$

$$
+ \sum_{j=1}^{t} \left[ \eta_j \delta_j \left( \prod_{k=j+1}^{t} (1 - \eta_k \tau_k) \right) \right]
$$

$$
\leq \exp \left( - \sum_{j=1}^{t} \eta_j \tau_j \right) \Phi_1 + \frac{HC + D}{2} \sum_{j=1}^{t} \left[ \eta_j^2 \exp \left( - \sum_{k=j+1}^{t} \eta_k \tau_k \right) \right]
$$

$$
+ \sum_{j=1}^{t} \eta_j \delta_j \exp \left( - \sum_{k=j+1}^{t} \eta_k \tau_k \right). \tag{13}
$$

where the last inequality holds according to $(1 - x) \leq \exp(-x)$ for $x < 1$. This completes the result. $\qquad \square$

While we have established its theoretical convergence, it doesn't tell us how close our solutions truly are to the optimal policy. Therefore, we prove the error bounds to bridge this gap, linking our theoretical convergence directly to practical performance guarantees.

**Lemma 2** (Per-episode trade-off). *For any $t \geq C''$, any constraint $i$ and any sequence $(\pi_t)_{t \in [T]}$, it holds*

$$
\left[ V_r^{\pi^\star} - V_r^{\pi_t} \right]_+ \leq H^{3/2} \left( 2\Phi_t \right)^{1/2} + \tau_t H \log(A) + \frac{H}{\Xi} \epsilon_{i,t},
$$

$$
\max_{i \in [m]} \left[ \alpha_i - V_{d_i}^{\pi_t} \right]_+ \leq \left[ H^{3/2} \left( 2\Phi_t \right)^{1/2} + \tau_t \left( \frac{4H}{\Xi} \right) - \epsilon_{i,t} \right]_+ .
$$

*Proof.* We first bound the reward distance between the optimal policy and the actual policy. We present the decomposition as

$$
V_r^{\pi^\star} - V_r^{\pi_t} = \underbrace{V_r^{\pi^\star} - V_r^{\pi_{\tau_t, \epsilon}^\star}}_{(1)} + \underbrace{V_r^{\pi_{\tau_t, \epsilon}^\star} - V_r^{\pi_t}}_{(2)}.
$$

We bound terms (1) and (2) respectively. For term (1), to bound the difference between $V_r^{\pi^\star}$ and $V_r^{\pi_{\tau_t, \epsilon}^\star}$, we first construct a feasible policy for the more constrained problem, i.e., $V_d^{\pi}(p) \geq \alpha + \epsilon_t$. We define a probabilistic mixed policy for any $t \geq C''$

$$
\pi^{\text{mix}} = (1 - B_t)\pi^\star + B_t \pi^0,
$$

where $B_t$ is a Bernoulli distributed random variable with $\frac{\epsilon_{i,t}}{\Xi}$ and $\pi^0$ is the feasible policy under the original optimization problem (as shown in Assumption 1). Then, we have that for any constraint $i$

$$
V_{d_i}^{\pi^{\text{mix}}} = \left( 1 - \frac{\epsilon_{i,t}}{\Xi} \right) V_{d_i}^{\pi^\star} + \frac{\epsilon_{i,t}}{\Xi} V_{d_i}^{\pi^0}
$$

$$
\geq \left( 1 - \frac{\epsilon_{i,t}}{\Xi} \right) \alpha_i + \frac{\epsilon_{i,t}}{\Xi} d_i^0
$$

$$
= \alpha_i + \epsilon_{i,t}.
$$

$\pi^{\mathrm{mix}}$ may not a Markov policy. However, by Lemma 7, there exists a markov policy $\hat{\pi}^{\mathrm{mix}}$, which has the same performance as $\pi^{\mathrm{mix}}$, which is the feasible policy for the problem below

$$\max_{\pi \in \Pi} \ V_r^{\pi} + \tau_t \mathcal{H}(\pi) \quad \text{s.t.} \quad V_{d_i}^{\pi} \geq \alpha_i + \epsilon_{i,t} \quad (\forall i \in [m]). \tag{14}$$

This indicates that

$$V_r^{\pi_{\tau_t,\epsilon}^{\star}} + \tau_t \mathcal{H}(\pi_{\tau_t,\epsilon}^{\star}) \geq V_r^{\pi^{\mathrm{mix}}} + \tau_t \mathcal{H}(\pi^{\mathrm{mix}}).$$

We then obtain

$$V_r^{\pi_{\tau_t,\epsilon}^{\star}} + \tau_t \mathcal{H}(\pi_{\tau_t,\epsilon}^{\star}) \geq \left( \left(1 - \frac{\epsilon_{i,t}}{\Xi}\right) V_r^{\pi^{\star}} + \frac{\epsilon_{i,t}}{\Xi} V_r^{\pi^0} \right) + \tau_t \mathcal{H}(\pi^{\mathrm{mix}})$$

Putting the difference term on the right side, it thus holds that

$$V_r^{\pi^{\star}} - V_r^{\pi_{\tau_t,\epsilon}^{\star}} \leq \frac{\epsilon_{i,t}}{\Xi} \left( V_r^{\pi^{\star}} - V_r^{\pi^0} \right) + \tau_t \left( \mathcal{H}(\pi_{\tau_t,\epsilon}^{\star}) - \mathcal{H}(\pi^{\mathrm{mix}}) \right)$$

$$\leq \frac{\epsilon_{i,t}}{\Xi} \cdot H + \tau_t H \log(A). \tag{15}$$

In terms of Term (2), we have

$$(2) = \sum_{h=1}^{H} \mathbb{E} \left[ \sum_a (\pi_{\tau_t,h}^{\star}(a|s) - \pi_{t,h}(a|s)) Q_{r,h}^{\pi_t}(s, a) \mid s_0 \right]$$

$$= \sum_{h=1}^{H} \sum_{s \in \mathcal{S}} q_h^{\pi_{\tau_t,\epsilon}^{\star}}(s) \sum_a (\pi_{\tau_t,h}^{\star}(a|s) - \pi_{t,h}(a|s)) Q_{r,h}^{\pi_t}(s, a)$$

$$\leq H \sum_{h=1}^{H} \sum_{s \in \mathcal{S}} q_h^{\pi_{\tau_t,\epsilon}^{\star}}(s) \| \pi_{\tau_t}(\cdot|s) - \pi_t(\cdot|s) \|_1 \qquad (\text{since } |Q_{r,h}^{\pi_t}(s, a)| \leq H)$$

$$\leq H \sum_{h=1}^{H} \sum_{s \in \mathcal{S}} q_h^{\pi_{\tau_t,\epsilon}^{\star}}(s) \sqrt{2 \mathrm{KL}(\pi_{\tau_t,h}^{\star}(\cdot|s), \pi_{t,h}(\cdot|s))} \qquad (\text{by Pinsker's Inequality})$$

$$\leq \sqrt{2} H \sqrt{ \left( \sum_{s,h} q_h^{\pi_{\tau_t,\epsilon}^{\star}}(s) \right) \left( \sum_{s,h} q_h^{\pi_{\tau_t,\epsilon}^{\star}}(s) \mathrm{KL}(\pi_{\tau_t,h}^{\star}(\cdot|s), \pi_{t,h}(\cdot|s)) \right) }$$

$$\leq \sqrt{2} H \sqrt{ H \sum_{s,h} q_h^{\pi_{\tau_t,\epsilon}^{\star}}(s) \mathrm{KL}(\pi_{\tau_t,h}^{\star}(\cdot|s), \pi_{t,h}(\cdot|s)) } \qquad \left( \text{since } \sum_{s,h} q_h^{\pi_{\tau_t,\epsilon}^{\star}}(s) \leq H \right)$$

$$= H^{3/2} \sqrt{2 \mathrm{KL}_t}. \tag{16}$$

Combining equation 15 and equation 16 together, we obtain

$$\left[ V_r^{\pi^{\star}} - V_r^{\pi_t} \right]_+ \leq H^{3/2} \left( 2 \mathrm{KL}_t \right)^{1/2} + \tau_t H \log(A) + \frac{H}{\Xi} \epsilon_{i,t}$$

$$\leq H^{3/2} \left( 2 \Phi_t \right)^{1/2} + \tau_t H \log(A) + \frac{H}{\Xi} \epsilon_{i,t}.$$

Next, we bound the maximum constraint violation between the thresholds and the policy. For any constraint $i \in [m]$, we give the decomposition as

$$\alpha_i - V_{d_i}^{\pi_t} = \underbrace{\alpha_i - V_{d_i}^{\pi_{\tau_t,\epsilon}^{\star}}}_{(3)} + \underbrace{V_{d_i}^{\pi_{\tau_t,\epsilon}^{\star}} - V_{d_i}^{\pi_t}}_{(4)}.$$

We then bound terms (3) and (4) respectively. The bound for Term (3) is derived from the properties of the saddle point $(\pi^\star_{\tau_t,\epsilon}, \lambda^\star_{\tau_t,\epsilon})$. Specially, we use the second inequality from Lemma 10, which states that for any $\lambda \in \mathcal{C}$: $\lambda^{\star\top}_{\tau_t,\epsilon}(V_d^{\pi^\star_{\tau_t,\epsilon}} - \epsilon_t - \alpha) \leq \lambda^\top(V_d^{\pi^\star_{\tau_t,\epsilon}} - \epsilon_t - \alpha) + \frac{\tau_t}{2}(\|\lambda\|^2 - \|\lambda^\star_{\tau_t,\epsilon}\|^2)$. Rearranging this inequality, it holds that:

$$(\lambda - \lambda^\star_{\tau_t,\epsilon})^\top (V_d^{\pi^\star_{\tau_t,\epsilon}} - \epsilon_t - \alpha) + \frac{\tau_t}{2}(\|\lambda\|^2 - \|\lambda^\star_{\tau_t,\epsilon}\|^2) \geq 0. \tag{17}$$

For any constraint $j \in [m]$, we construct a specific $\lambda$ vector by choosing $\lambda_i = \lambda^\star_{\tau_t,\epsilon,i}$ for all $i \neq j$, and $\lambda_j = z$ for some $z \in [0, \left(\frac{4H}{\Xi}\right)]$. Substituting this into equation 17 reduces it to terms concerning only the $j$-th dimension

$$\left(z - \lambda^\star_{\tau_t,\epsilon,j}\right) \left(V_{d_j}^{\pi^\star_{\tau_t,\epsilon}} - \epsilon_t - \alpha_j\right) + \frac{\tau_t}{2}\left(z^2 - (\lambda^\star_{\tau_t,\epsilon,j})^2\right) \geq 0.$$

The dual solution does not lie on the boundary of the feasible set (i.e. $\lambda^\star_{\tau_t,\epsilon,j} < \frac{4H}{\Xi}$). Thus we can choose a value $z$ such that $\lambda^\star_{\tau_t,j} < z \leq \left(\frac{4H}{\Xi}\right)$, which means $z - \lambda^\star_{\tau_t,j} > 0$. After rearranging, we get

$$(3) = \alpha_j - V_{d_j}^{\pi^\star_{\tau_t,\epsilon}} \leq \frac{\tau_t}{2}(z + \lambda^\star_{\tau_t,\epsilon,j}) - \epsilon_{j,t}$$

$$\leq \tau_t \left(\frac{4H}{\Xi}\right) - \epsilon_{j,t}. \tag{18}$$

For term (4), a similar analysis to that of the reward gap (in equation 16), we have

$$(4) = V_{d_i}^{\pi^\star_{\tau_t,\epsilon}} - V_{d_i}^{\pi_t} \leq H^{3/2}(2\mathrm{KL}_t)^{1/2}. \tag{19}$$

Then we combine the bounds from equation 18 and equation 19 together to get the upper bound below:

$$\alpha_i - V_{d_i}^{\pi_t} \leq H^{3/2}\left(2\mathrm{KL}_t\right)^{1/2} + \tau_t\left(\frac{4H}{\Xi}\right) - \epsilon_{i,t}$$

$$\leq H^{3/2}\left(2\Phi_t\right)^{1/2} + \tau_t\left(\frac{4H}{\Xi}\right) - \epsilon_{i,t}.$$

Since this holds for any constraint $i$, we can take the positive part on both sides and then the maximum over $i$ to obtain the final result:

$$\max_{i \in [m]} \left[\alpha_i - V_{d_i}^{\pi_t}\right]_+ \leq \left[H^{3/2}\left(2\Phi_t\right)^{1/2} + \tau_t\left(\frac{4H}{\Xi}\right) - \epsilon_{i,t}\right]_+.$$

$\square$

### E.1 STRONG REGRET BOUNDS

We are now ready to establish the bounds for strong reward regret and strong constraint violation of Algorithm 1.

**Theorem 1** (Bounds for reward regret and constraint violation regret). *Let $\eta_t = t^{-3/4}$, $\tau_t = t^{-1/8}$ for $t \geq 1$, and $\epsilon_{i,t} = 6\sqrt{H^3 C_B}\left(t^{-1/8} \cdot \log(SAHt/\delta')^{1/4}\right)$ for all constraint $i$. For a confidence parameter $\delta \in (0,1)$, with probability at least $1 - \delta$, when $T$ is sufficiently large, Algorithm 1 achieves the following bounds:*

$$\mathcal{R}_T(r) \leq \tilde{O}(T^{7/8}) \quad and \quad \mathcal{R}_T(d) = \tilde{O}(1).$$

*where $T$ denotes the number of episodes, $C_B = \left(1 + \frac{8mH}{\Xi}\right)\left(4H\sqrt{2SA}\left(H\sqrt{S} + H + 1\right)\right) + \frac{4mH}{\Xi}\sqrt{2H}$ is a $T$-independent constant and $\tilde{O}$ hides polylogarithmic factors in $(S, A, H, m, \log(T), \log(\frac{1}{\delta}), \Xi)$.*

*Proof.* For episode $t \geq C''$, according to Lemmas 1 and 2, we can obtain that for any constraint $i$

$$[V_r^\star - V_r^{\pi_t}]_+ \leq H^{3/2}\sqrt{2\Phi_t} + \tau_t \log(A)H + \frac{H}{\Xi}\epsilon_{i,t}$$

$$\leq H^{3/2}\exp\left(-\sum_{j=1}^t \eta_j\tau_j/2\right)\sqrt{2\Phi_1} + H^{3/2}\sqrt{HC+D}\left(\sum_{j=1}^t \eta_j^2 \exp\left(-\sum_{k=j+1}^t \eta_k\tau_k\right)\right)^{1/2}$$

$$+ \sqrt{2H^3}\left(\sum_{j=1}^t \eta_j\delta_j \exp\left(-\sum_{k=j+1}^t \eta_k\tau_k\right)\right)^{1/2} + \tau_t\log(A)H + \frac{H}{\Xi}\epsilon_{i,t}.$$

Since the strong reward regret $\mathcal{R}_T(r) = \sum_{t\in[T]}[V_r^{\pi^\star} - V_r^{\pi_t}]_+$, it can obtain by summing all the terms over $T$ episodes. Thus, we have

$$\mathcal{R}_T(r) \leq \sum_{t=1}^{C''-1}[V_r^{\pi^\star} - V_r^{\pi_t}]_+ + \sum_{t=C''}^T \left(H^{3/2}\sqrt{2\Phi_t} + \tau_t\log(A)H + \frac{H}{\Xi}\epsilon_{i,t}\right)$$

$$\leq (C''-1)H$$

$$+ \sum_{t=C''}^T H^{3/2}\exp\left(-\sum_{j=1}^t \eta_j\tau_j/2\right)\sqrt{2\Phi_1} \tag{a}$$

$$+ \sum_{t=C''}^T H^{3/2}\sqrt{HC+D}\left(\sum_{j=1}^t \eta_j^2\exp\left(-\sum_{k=j+1}^t \eta_k\tau_k\right)\right)^{1/2} \tag{b}$$

$$+ \sum_{t=C''}^T \sqrt{2H^3}\left(\sum_{j=1}^t \eta_j\delta_j\exp\left(-\sum_{k=j+1}^t \eta_k\tau_k\right)\right)^{1/2} \tag{c}$$

$$+ \sum_{t=C''}^T \tau_t\log(A)H \tag{d}$$

$$+ \sum_{t=C''}^T \frac{H}{\Xi}\epsilon_{i,t}. \tag{e}$$

Since $(C''-1)H$ is a constant that is $T$-independent, we will proceed to analysis the bound for term (a)-term (f) separately.

**Bounding term (a)** According to the definition of $\Phi_1$, we have $\sqrt{2\Phi_1} \leq \sqrt{2H\log(A) + m\left(\frac{4H}{\Xi}\right)^2} := C'$. Thus, it holds

$$(a) \leq H^{3/2}C'\sum_{t=C''}^T \exp(-\sum_{j=1}^t \eta_j\tau_j/2)$$

$$= H^{3/2}C'\sum_{t=C''}^T \exp(-\sum_{j=1}^t j^{-7/8}/2). \tag{20}$$

For the exponent, it yields that $\sum_{j=1}^{t} j^{-7/8} \geq \int_{1}^{t+1} x^{-7/8} = 8(t+1)^{1/8} - 8$. Substituting this lower bound into the summand yields:

$$\exp\left(-\frac{1}{2}\sum_{j=1}^{t} j^{-7/8}\right) \leq e^4 \exp\left(-4(t+1)^{1/8}\right).$$

The sum over $t$ is therefore bounded by

$$\sum_{t=C''}^{T} e^4 \exp\left(-4(t+1)^{1/8}\right) \leq \sum_{t=C''}^{\infty} e^4 \exp\left(-4(t+1)^{1/8}\right)$$

$$\leq \int_{C''}^{\infty} e^4 \exp\left(-4(x+1)^{1/8}\right) dx$$

$$\leq 4e^4 (C'')^{7/8} \exp\left(-4(C'')^{1/8}\right).$$

Combining this result with the constant, it yields:

$$(a) \leq H^{3/2} C' \left(4e^4 (C'')^{7/8} \exp\left(-4(C'')^{1/8}\right)\right) = \tilde{O}(1).$$

**Bounding term (b)** We first calculate the bound of the term $HC + D$ as follows:

$$HC + D = H \exp\left(\eta_t H\left(1 + \frac{4mH}{\Xi} + \tau_t \log(A)\right)\right)\left(2A^{\eta_t \tau_t} H^2 \left(1 + \frac{4mH}{\Xi} + \tau_t \log(A)\right)^2 + \frac{128\tau_t^2 \sqrt{A}}{e^2}\right)$$

$$+ m\left(H + \tau_t\left(\frac{4H}{\Xi}\right)\right)^2$$

$$\leq H \exp\left(H\left(1 + \frac{4mH}{\Xi} + \log(A)\right)\right)\left(2AH^2 \left(1 + \frac{4mH}{\Xi} + \log(A)\right)^2 + \frac{128\sqrt{A}}{e^2}\right)$$

$$+ m\left(H + \frac{4H}{\Xi}\right)^2.$$

We can find that the right-hand side of the second inequality is the order of constant, i.e., $\tilde{O}(1)$. Moreover, by Lemma 17, it yields

$$(b) \leq K \cdot T^{11/16},$$

where $K = \frac{16}{11} H^{3/2} \sqrt{HC + D} \sqrt{\zeta(3/2) + 4\sqrt{2}}$.

**Bounding term (c)** For term (c), by Lemma 18 and Lemma 13, we obtain

$$(c) \leq \tilde{O}(T^{7/8}).$$

**Bounding term (d)** In terms of (d), it holds

$$(d) = \sum_{t=C''}^{T} \tau_t \log(A) H \leq \log(A) H \sum_{t=C''}^{T} t^{-1/8} = \tilde{O}(T^{7/8}).$$

**Bounding term (e)** For term (e), by our setting,

$$(e) = \frac{H}{\Xi} \sum_{t=C''}^{T} \epsilon_{i,t} = \tilde{O}(T^{7/8}).$$

We now calculate the regret bound for constraint violation. By Lemma 2, for episode $t \geq C''$, the per-episode constraint violation is bounded by:

$$\max_{i \in [m]} \left[ \alpha_i - V_{d_i}^{\pi_t} \right]_+ \leq [H^{3/2}\sqrt{2\Phi_t} + \tau_t \left( \frac{4H}{\Xi} \right) - \epsilon_{i,t}]_+.$$

Let $P_t := H^{3/2}\sqrt{2\Phi_t} + \tau_t \left( \frac{4H}{\Xi} \right)$. The cumulative constraint violation is bounded by

$$R_T(d) \leq \max_{i \in [m]} \sum_{t=1}^{C''-1} [\alpha_i - V_{d_i}^{\pi_t}]_+ + \sum_{t=C''}^{T} [P_t - \epsilon_{i,t}]_+$$

$$\leq (C'' - 1)H + \sum_{t=C''}^{T} [P_t - \epsilon_{i,t}]_+$$

$$\leq (C'' - 1)H + \underbrace{\sum_{t=C''}^{T} \left[ H^{3/2}\sqrt{2\Phi_1} \exp(-\sum_{j=1}^{t} \eta_j \tau_j/2) - \epsilon_{i,t}^{(1)} \right]_+}_{(a')}$$

$$+ \underbrace{\sum_{t=C''}^{T} \left[ H^{3/2}\sqrt{HC+D} \left( \sum_{j=1}^{t} \eta_j^2 \exp\left( -\sum_{k=j+1}^{t} \eta_k \tau_k \right) \right)^{1/2} - \epsilon_{i,t}^{(2)} \right]_+}_{(b')}$$

$$+ \underbrace{\sum_{t=C''}^{T} \left[ \sqrt{2H^3} \left( \sum_{j=1}^{t} \eta_j \delta_j \exp\left( -\sum_{k=j+1}^{t} \eta_k \tau_k \right) \right)^{1/2} - \epsilon_{i,t}^{(3)} \right]_+}_{(c')} + \underbrace{\sum_{t=C''}^{T} \left[ \left( \frac{4H}{\Xi} \right) \tau_t - \epsilon_{i,t}^{(4)} \right]_+}_{(d')},$$

where $\epsilon_{i,t} \geq \epsilon_{i,t}^{(1)} + \epsilon_{i,t}^{(2)} + \epsilon_{i,t}^{(3)} + \epsilon_{i,t}^{(4)}$. We will show that with the appropriate choice of $\epsilon_{i,t}^{(i)}$ for any $i \in [4]$, the bound for each term (a')-(d') is $\tilde{O}(1)$.

**Bounding term (d')** For term (d'),

$$(d') = \sum_{t=C''}^{T} \left[ \left( \frac{4H}{\Xi} \right) t^{-1/8} - \epsilon_{i,t}^{(4)} \right].$$

With the choice of $\epsilon_{i,t}^{(4)} = \left( \frac{4H}{\Xi} \right) t^{-1/8}$, we immediately get that $(d') = 0$.

**Bounding term (a')** For term (a'),

$$(a') = \sum_{t=C''}^{T} [H^{3/2}\sqrt{2\Phi_1} \exp(-\sum_{j=1}^{t} j^{-7/8}/2) - \epsilon_{i,t}^{(1)}]_+.$$

With the choice of $\epsilon_{i,t}^{(1)} = 0$ and the same analysis as term (a), it yields that $(a') = \tilde{O}(1)$.

**Bounding term (b')** For term (b') and any $t \geq C''$,

$$(b') = \sum_{t=C''}^{T} [H^{3/2}\sqrt{HC+D}\left(\sum_{j=1}^{t} j^{-3/2}\exp\left(-\sum_{k=j+1}^{t}k^{-7/8}\right)\right)^{1/2} - \epsilon_{i,t}^{(2)}]_{+}.$$

By Lemma 17, it yields that $\left(\sum_{j=1}^{t} j^{-3/2}\exp\left(-\sum_{k=j+1}^{t}k^{-7/8}\right)\right)^{1/2}$ is of the same asymptotic order $t^{-5/16}$. We can choose $\epsilon_{i,t}^{(2)} = H^{3/2}\sqrt{HC+D}\cdot t^{-1/8}$. Since the term $t^{-5/16}$ decays strictly faster than $t^{-1/8}$, by Lemma 8, we thus obtain $(b') = \tilde{O}(1)$.

**Bounding term (c')** For term (c'),

$$(c') \leq \sum_{t=C''}^{T} \left[\sqrt{2H^3}\left(\sum_{j=1}^{t}\eta_j\delta_j\exp\left(-\sum_{k=j+1}^{t}\eta_k\tau_k\right)\right)^{1/2} - \epsilon_{i,t}^{(3)}\right]_{+}.$$

By Lemma 18 and Lemma 8, we pick $\epsilon_{i,t}^{(3)} = 4\sqrt{H^3 C_B}\left(t^{-1/8}\cdot \log(SAHt/\delta')^{1/4}\right)$, where $C_B = \left(1+\frac{8mH}{\Xi}\right)\left(4H\sqrt{2SA}\left(H\sqrt{S}+H+1\right)\right) + \frac{4mH}{\Xi}\sqrt{2H}$. It holds that $\lim_{t\to\infty}\frac{A_t}{\epsilon_{i,t}^{(3)}} < 1$, where $A_t = \sqrt{2H^3}\left(\sum_{j=1}^{t}\eta_j\delta_j\exp\left(-\sum_{k=j+1}^{t}\eta_k\tau_k\right)\right)^{1/2}$ and thus it holds $(b') = \tilde{O}(1)$.

Sum the terms from $\epsilon_{i,t}^{(1)}$ to $\epsilon_{i,t}^{(4)}$, we find that $\epsilon_{i,t} \geq \epsilon_{i,t}^{(1)} + \epsilon_{i,t}^{(2)} + \epsilon_{i,t}^{(3)} + \epsilon_{i,t}^{(4)}$. Moreover, sum all of the bounds for reward regret and constraint violation together, we get the final result:

$$\mathcal{R}_T(r) = \tilde{O}(T^{7/8}) \quad \text{and} \quad \mathcal{R}_T(d) = \tilde{O}(1).$$

$\square$

# F LAST-ITERATE CONVERGENCE

In this section, we present the property of Algorithm 1, showing that the primal-dual iterates of FlexDOME converge in the last iterate. In contrast to Lemma 1 which accounts for estimation errors, we analyze a more fundamental scenario. Here, we assume the model is known, thereby allowing us to neglect the effects of estimation errors. This setting enables a more direct proof of its intrinsic convergence guarantee, as shown in the following lemma.

**Theorem 2** (Convergence for potential functions). *Let $\eta_t$, $\tau_t \leq 1$. The policy-dual divergence potential holds*

$$\Phi_{t+1} \leq (1-\eta_t\tau_t)\Phi_t + \frac{\eta_t^2}{2}(HC+D)$$

*where $C = \exp\left(\eta_t H\left(1+\frac{4mH}{\Xi}+\tau_t\log(A)\right)\right)\left(2A^{\eta_t\tau_t}H^2\left(1+\frac{4mH}{\Xi}+\tau_t\log(A)\right)^2 + \frac{128\tau_t^2\sqrt{A}}{e^2}\right)$ and $D = m\left(H+\tau_t\left(\frac{4H}{\Xi}\right)\right)^2$.*

*Proof.* We recall the definition of the potential function $\Phi_t = \sum_{s,h}\mathbb{P}_{\pi_{\tau_t,\epsilon}^{\star}}[s_h = s]\,\mathrm{KL}\left(\pi_{\tau_t,h}^{\star}(\cdot\,|\,s), \pi_{t,h}(\cdot\,|\,s)\right) + \frac{1}{2}\left\|\lambda_{\tau_t,\epsilon}^{\star} - \lambda_t\right\|^2$. We first decompose the primal-dual gap,

$$\mathcal{L}_{\tau_t,t}(\pi_{\tau_t,\epsilon}^{\star}, \lambda_t) - \mathcal{L}_{\tau_t,t}(\pi_t, \lambda_{\tau_t,\epsilon}^{\star}) = \underbrace{\mathcal{L}_{\tau_t,t}(\pi_{\tau_t,\epsilon}^{\star}, \lambda_t) - \mathcal{L}_{\tau_t,t}(\pi_t, \lambda_t)}_{(1)} + \underbrace{\mathcal{L}_{\tau_t,t}(\pi_t, \lambda_t) - \mathcal{L}_{\tau_t,t}(\pi_t, \lambda_{\tau_t,\epsilon}^{\star})}_{(2)}$$

and we next deal with (1) and (2), separately.

**Bounding term (1)**

$$(1) = \mathcal{L}_{\tau_t,t}\big(\pi^\star_{\tau_t,\epsilon}, \lambda_t\big) - \mathcal{L}_{\tau_t,t}\big(\pi_t, \lambda_t\big)$$

$$= V^{\pi^\star_{\tau_t,\epsilon}}_{r+\lambda_t^T g} - V^{\pi_t}_{r+\lambda_t^T g}$$

$$+ \tau_t \sum_{s,a,h} q_h^{\pi_t}(s)\pi_{t,h}(a\,|\,s) \log\big(\pi_{t,h}(a\,|\,s)\big) - \tau_t \sum_{s,a,h} q_h^{\pi^\star_{\tau_t,\epsilon}}(s)\pi^\star_{\tau_t,\epsilon,h}(a\,|\,s) \log\big(\pi^\star_{\tau_t,\epsilon,h}(a\,|\,s)\big)$$

$$= V^{\pi^\star_{\tau_t,\epsilon}}_{r+\lambda_t^T g+\tau_t\psi_t} - V^{\pi_t}_{r+\lambda_t^T g+\tau_t\psi_t}$$

$$+ \tau_t \sum_{s,a,h} q_h^{\pi^\star_{\tau_t,\epsilon}}(s)\pi^\star_{\tau_t,\epsilon,h}(a\,|\,s) \log\big(\pi_{t,h}(a\,|\,s)\big) - \tau_t \sum_{s,a,h} q_h^{\pi^\star_{\tau_t,\epsilon}}(s)\pi^\star_{\tau_t,\epsilon,h}(a\,|\,s) \log\big(\pi^\star_{\tau_t,\epsilon,h}(a\,|\,s)\big)$$

$$= V^{\pi^\star_{\tau_t,\epsilon}}_{r+\lambda_t^T g+\tau_t\psi_t} - V^{\pi_t}_{r+\lambda_t^T g+\tau_t\psi_t} - \tau_t \mathrm{KL}_t$$

$$= V^{\pi^\star_{\tau_t,\epsilon}}_{y_t} - V^{\pi_t}_{y_t} - \tau_t \mathrm{KL}_t.$$

By Lemma 5, we have

$$V^{\pi^\star_{\tau_t,\epsilon}}_{y_t} - V^{\pi_t}_{y_t} = \sum_{s,h} q_h^{\pi^\star_{\tau_t,\epsilon}} \big\langle Q^{\pi_t}_{y_t,h}(s,\cdot), \pi^\star_{\tau_t,h}(\cdot\mid s) - \pi_{t,h}(\cdot\mid s)\big\rangle$$

$$\leq \sum_{s,h} q_h^{\pi^\star_{\tau_t,\epsilon}} \Big(\frac{\mathrm{KL}_{t,h}(s) - \mathrm{KL}_{t+1,h}(s)}{\eta_t} + \frac{\eta_t}{2} \sum_a \pi_{t,h}(a\,|\,s)\exp\big(Q^{\pi_t}_{y_t,h}(s,a)\big) Q^{\pi_t}_{y_t,h}(s,a)^2\Big).$$

According to Lemma 15, it holds that $\sum_a \pi_{t,h}(a\mid s)\exp\big(Q^{\pi_t}_{y_t,h}(s,a)\big) Q^{\pi_t}_{y_t,h}(s,a)^2 \leq C$, where $C = \exp\big(\eta_t H\big(1 + \frac{4mH}{\Xi} + \tau_t\log(A)\big)\big)\Big(2A^{\eta_t\tau_t}H^2\big(1 + \frac{4mH}{\Xi} + \tau_t\log(A)\big)^2 + \frac{128\tau_t^2\sqrt{A}}{e^2}\Big)$. Then we obtain

$$V^{\pi^\star_{\tau_t,\epsilon}}_{y_t} - V^{\pi_t}_{y_t} \leq \sum_{s,h} q_h^{\pi^\star_{\tau_t,\epsilon}}\Big(\frac{\mathrm{KL}_{t,h}(s) - \mathrm{KL}_{t+1,h}(s)}{\eta_t} + \frac{\eta_t}{2}C\Big)$$

$$= \frac{\mathrm{KL}_t - \mathrm{KL}_{t+1}}{\eta_t} + \frac{\eta_t H}{2}C.$$

Therefore, we get

$$(1) = V^{\pi^\star_{\tau_t,\epsilon}}_{y_t} - V^{\pi_t}_{y_t} - \tau_t\mathrm{KL}_t \leq \frac{\mathrm{KL}_t - \mathrm{KL}_{t+1}}{\eta_t} + \frac{\eta_t H}{2}C - \tau_t\mathrm{KL}_t = \frac{1 - \eta_t\tau_t}{\eta_t}\mathrm{KL}_t - \frac{\mathrm{KL}_{t+1}}{\eta_t} + \frac{\eta_t H}{2}C. \quad (21)$$

**Bounding term (2)**

$$(2) = \mathcal{L}_{\tau_t,t}(\pi_t, \lambda_t) - \mathcal{L}_{\tau_t,t}(\pi_t, \lambda^\star_{\tau_t,\epsilon})$$

$$= \sum_{i\in[m]} \lambda_{t,i}(V^{\pi_t}_{d_i} - \epsilon_t - \alpha_i) - \sum_{i\in[m]} \lambda^\star_{\tau_t,\epsilon,i}(V^{\pi_t}_{d_i} - \epsilon_t - \alpha_i) + \frac{\tau_t}{2}\|\lambda_t\|^2 - \frac{\tau_t}{2}\|\lambda^\star_{\tau_t,\epsilon}\|^2$$

$$= \sum_{i\in[m]} (\lambda_{t,i} - \lambda^\star_{\tau_t,\epsilon,i})(V^{\pi_t}_{d_i} - \epsilon_t - \alpha_i + \tau_t\lambda_{t,i}) - \frac{\tau_t}{2}\|\lambda_t - \lambda^\star_{\tau_t,\epsilon}\|^2$$

$$\leq \frac{\|\lambda^\star_{\tau_t,\epsilon} - \lambda_t\|^2 - \|\lambda^\star_{\tau_t,\epsilon} - \lambda_{t+1}\|^2}{2\eta_t} + \frac{\eta_t}{2}\|V^{\pi_t}_d - \epsilon_t - \alpha + \tau_t\lambda_t\|^2 - \frac{\tau_t}{2}\|\lambda_t - \lambda^\star_{\tau_t,\epsilon}\|^2.$$

Since $\|V_d^{\pi_t} - \epsilon_t - \alpha\| \le \sqrt{m}H$ and $\|\lambda_t\| \le \sqrt{m}\left(\frac{4H}{\Xi}\right)$, we have $\|V_{d_i}^{\pi_t} - \epsilon_t - \alpha_i + \tau_t\lambda_{t,i}\|^2 \le D$, where $D = m\left(H + \tau_t\left(\frac{4H}{\Xi}\right)\right)^2$. Hence, it holds that

$$(2) \le \frac{\|\lambda_{\tau_t,\epsilon}^\star - \lambda_t\|^2 - \|\lambda_{\tau_t,\epsilon}^\star - \lambda_{t+1}\|^2}{2\eta_t} + \frac{\eta_t}{2}D - \frac{\tau_t}{2}\|\lambda_t - \lambda_{\tau_t,\epsilon}^\star\|^2$$

$$= \frac{1 - \eta_t\tau_t}{2\eta_t}\|\lambda_{\tau_t,\epsilon}^\star - \lambda_t\|^2 - \frac{1}{2\eta_t}\|\lambda_{\tau_t,\epsilon}^\star - \lambda_{t+1}\|^2 + \frac{\eta_t}{2}D. \tag{22}$$

By combining equation 21 and equation 22, we obtain

$$\Phi_{t+1} = \mathrm{KL}_{t+1} + \frac{1}{2}\|\lambda_{t+1} - \lambda_{\tau_t,\epsilon}^\star\|^2$$

$$\le (1 - \eta_t\tau_t)(\mathrm{KL}_t + \frac{1}{2}\|\lambda_t - \lambda_{\tau_t,\epsilon}^\star\|^2) + \frac{\eta_t^2}{2}(HC + D) - \eta_t((1) + (2))$$

$$\le (1 - \eta_t\tau_t)\Phi_t + \frac{\eta_t^2}{2}(HC + D), \qquad\qquad (\text{since } (1) + (2) \ge 0)$$

where $HC + D = H\exp\left(\eta_t H\left(1 + \frac{4mH}{\Xi} + \tau_t\log(A)\right)\right)\left(2A^{\eta_t\tau_t}H^2\left(1 + \frac{4mH}{\Xi} + \tau_t\log(A)\right)^2 + \frac{128\tau_t^2\sqrt{A}}{e^2}\right) + m\left(H + \tau_t\left(\frac{4H}{\Xi}\right)\right)^2$. $\qquad\square$

Building upon the linear convergence of the regularized scheme established in Lemma 2 and error bounds guaranteed by Lemma 2, we now demonstrate that the iterates converge to the optimal policy of the original and unregularised problem. We first prove the following lemma and then provide the last-iterate convergence guarantee.

**Lemma 19** (Asymptotic bound on the recursive error sum). *Let $\varepsilon \in (0, 1)$ be a sufficiently small positive number. Assume the step-size and regularization parameters satisfy $\eta_j = \Theta(\varepsilon^4)$ and $\tau_j = \Theta(\varepsilon^2)$ for all $j \in [t]$. Further, assume the number of steps $t = \Omega(\varepsilon^{-7})$. Then the following bound holds:*

$$\sum_{j=1}^{t} \eta_j^2 \exp\left(-\sum_{k=j+1}^{t} \eta_k\tau_k\right) = \Theta(\varepsilon^2).$$

*Proof.* Let the expression be denoted by $S_t$. By the definitions of asymptotic notation, there exist positive constants $c_{\eta,1}, c_{\eta,2}, c_{\tau,1}, c_{\tau,2}$ such that for all $j \in [t]$:

$$c_{\eta,1}\varepsilon^4 \le \eta_j \le c_{\eta,2}\varepsilon^4 \quad \text{and} \quad c_{\tau,1}\varepsilon^2 \le \tau_j \le c_{\tau,2}\varepsilon^2.$$

The proof proceeds by establishing a matching upper bound ($\mathcal{O}$) and lower bound ($\Omega$).

Let $C_1 := c_{\eta,1}c_{\tau,1}$. The sum in the exponent is bounded below by $\sum_{k=j+1}^{t} \eta_k\tau_k \ge (t - j)C_1\varepsilon^6$. We then have:

$$S_t \le \sum_{j=1}^{t} (c_{\eta,2}^2\varepsilon^8)\exp\left(-(t - j)C_1\varepsilon^6\right)$$

$$= c_{\eta,2}^2\varepsilon^8 \sum_{m=0}^{t-1} \left(e^{-C_1\varepsilon^6}\right)^m \qquad\qquad (m = t - j)$$

$$\le c_{\eta,2}^2\varepsilon^8 \left(\frac{1}{1 - e^{-C_1\varepsilon^6}}\right).$$

Since $1 - e^{-x} = \Theta(x)$ for small $x > 0$, the term in the parenthesis is $\mathcal{O}(\varepsilon^{-6})$. Thus, $S_t \leq c_{\eta,2}^2 \varepsilon^8 \cdot \mathcal{O}(\varepsilon^{-6}) = \mathcal{O}(\varepsilon^2)$. Let $C_2 := c_{\eta,2} c_{\tau,2}$. The sum in the exponent is bounded above by $\sum_{k=j+1}^{t} \eta_k \tau_k \leq (t-j) C_2 \varepsilon^6$. We then have:

$$S_t \geq \sum_{j=1}^{t} (c_{\eta,1}^2 \varepsilon^8) \exp\left(-(t-j) C_2 \varepsilon^6\right)$$

$$= c_{\eta,1}^2 \varepsilon^8 \sum_{m=0}^{t-1} \left(e^{-C_2 \varepsilon^6}\right)^m.$$

The finite geometric sum is $\frac{1-r^t}{1-r}$, where $r = e^{-C_2 \varepsilon^6}$. As $t = \Omega(\varepsilon^{-7})$, the term $t C_2 \varepsilon^6 = \Omega(\varepsilon^{-1})$ approaches infinity as $\varepsilon \to 0$. Therefore, $r^t = \exp(-t C_2 \varepsilon^6)$ approaches 0, which implies $1 - r^t$ is bounded below by a positive constant for sufficiently small $\varepsilon$. The denominator $1 - r$ is $\Theta(\varepsilon^6)$. Thus, the sum is $\Omega(\varepsilon^{-6})$. Combining these terms, we obtain the lower bound: $S_t \geq c_{\eta,1}^2 \varepsilon^8 \cdot \Omega(\varepsilon^{-6}) = \Omega(\varepsilon^2)$. This completes the proof. $\square$

Building upon the lemmas above, we now give the guarantee of last-iterate convergence.

**Theorem 3** (Last-iterate convergence). *Conditioned on Assumption 1, for small $\varepsilon > 0$ and $t = \Omega(\varepsilon^{-7})$, if $\eta_t = \Theta(\varepsilon^4)$, $\tau_t = \Theta(\varepsilon^2)$ and $\epsilon_{i,t} = \Theta(\varepsilon)$ for all constraint $i$, then we have*

$$\left[V_r^{\pi^\star} - V_r^{\pi_t}\right]_+ \leq \Theta(\varepsilon), \quad \left[\alpha_i - V_{d_i}^{\pi_t}\right]_+ = 0 \quad (\forall i \in [m]).$$

*Proof.* According to Lemma 2, we have $\Phi_{t+1} \leq (1 - \eta_t \tau_t) \Phi_t + \frac{\eta_t^2}{2}(HC + D)$, and thus it holds that

$$\Phi_{t+1} \leq \left(\prod_{j=1}^{t} (1 - \eta_j \tau_j)\right) \Phi_1 + \sum_{j=1}^{t} \left[\frac{\eta_j^2}{2}(HC + D)\left(\prod_{k=j+1}^{t} (1 - \eta_k \tau_k)\right)\right]$$

$$\leq \left(\prod_{j=1}^{t} (1 - \eta_j \tau_j)\right) \Phi_1 + \frac{HC + D}{2} \sum_{j=1}^{t} \left[\eta_j^2\left(\prod_{k=j+1}^{t} (1 - \eta_k \tau_k)\right)\right]$$

$$\leq \exp\left(-\sum_{j=1}^{t} \eta_j \tau_j\right) \Phi_1 + \frac{HC + D}{2} \sum_{j=1}^{t} \left[\eta_j^2 \exp\left(-\sum_{k=j+1}^{t} \eta_k \tau_k\right)\right].$$

Combining Lemmas 1 and 2, we obtain the reward distance between the optimal policy and the exact policy as

$$\left[V_r^{\pi^\star} - V_r^{\pi_t}\right]_+ \lesssim H^{3/2} \exp\left(-\sum_{j=1}^{t} \eta_j \tau_j / 2\right) \sqrt{\Phi_1} \tag{a}$$

$$+ H^{3/2} \sqrt{HC + D} \left(\sum_{j=1}^{t} \eta_j^2 \exp\left(-\sum_{k=j+1}^{t} \eta_k \tau_k\right)\right)^{1/2} \tag{b}$$

$$+ \tau_t H \log(A) \tag{c}$$

$$+ \frac{H}{\Xi} \epsilon_t. \tag{d}$$

We now analyze the order of each term with the chosen parameters $\tau_t = \Theta(\varepsilon^2)$, $\eta_t = \Theta(\varepsilon^4)$ and $t = \Omega(\varepsilon^{-7})$. we discuss each term individually.

For term (a), the product in the exponent scales as:

$$\sum_{j=1}^{t} \eta_j \tau_j = \Theta(\varepsilon^4) \cdot \Theta(\varepsilon^2) \cdot \Omega(\varepsilon^{-7}) = \Omega(\varepsilon^{-1}).$$

As $\varepsilon$ goes to zero, the exponent $\sum_{j=1}^{t} \eta_j \tau_j$ grows to infinity, causing $\exp\left(-\sum_{j=1}^{t} \eta_j \tau_j / 2\right)$ to decay to zero faster than any polynomial in $\varepsilon$. For potential term $\Phi_1$, we have $\Phi_1 \leq (H \log(A) + \frac{1}{2} m \left(\frac{4H}{\Xi}\right)^2)^{1/2}$. Thus, it shows

$$(a) = o(\varepsilon). \tag{23}$$

For term (b), by Lemma 19, we have

$$\left(\sum_{j=1}^{t} \eta_j^2 \exp\left(-\sum_{k=j+1}^{t} \eta_k \tau_k\right)\right)^{1/2} = \Theta(\varepsilon^{2/2}) = \Theta(\varepsilon).$$

We can find that $H^{3/2}\sqrt{HC + D}$ is a constant. Thus, we have

$$(b) = \Theta(\varepsilon). \tag{24}$$

In terms of (c), we immediately get

$$(c) = \tau_t H \log(A) = \Theta(\varepsilon^2). \tag{25}$$

For term (d), it holds

$$(d) = \frac{H}{\Xi} \epsilon_{i,t} = \Theta(\varepsilon). \tag{26}$$

By Lemma 1 and Lemma 2, the constraint violation between the thresholds and the policy satisfies

$$\left[\alpha_i - V_{d_i}^{\pi_t}\right]_+ \lesssim \left[H^{3/2} \exp\left(-\sum_{j=1}^{t} \eta_j \tau_j / 2\right)\sqrt{\Phi_1}\right. \tag{a'}$$

$$+ H^{3/2}\sqrt{HC + D}\left(\sum_{j=1}^{t} \eta_j^2 \exp\left(-\sum_{k=j+1}^{t} \eta_k \tau_k\right)\right)^{1/2} \tag{b'}$$

$$+ \left(\frac{4H}{\Xi}\right)\tau_t \tag{c'}$$

$$\left. - \epsilon_t\right]_+. \tag{d'}$$

The bounds of terms (a') and (b') are equal to that in the case of reward analysis. For term (c'), with the choice of regularized parameter $\tau_t$, we have

$$(c') = \tau_t \left(\frac{4H}{\Xi}\right) = \Theta(\varepsilon^2). \tag{27}$$

Let $E_t = (a') + (b') + (c')$. The dominant term in $E_t$ is $(b')$, which is of order $\Theta(\varepsilon)$. The terms $(a')$ and $(c')$ are of order $o(\varepsilon)$. The sum of a $\Theta(\varepsilon)$ term and $o(\varepsilon)$ terms remains $\Theta(\varepsilon)$. Thus, there exists a constant $C_{\text{err}} > 0$ such that for sufficiently small $\varepsilon$, the total error is bounded by $E_t \leq C_{\text{err}}\varepsilon$.

The safety margin is chosen as $\epsilon_{i,t} = C_\epsilon \varepsilon$ for a constant $C_\epsilon > 0$ that we control and any constraint $i$. The expression inside the operator is therefore bounded as:

$$E_t - \epsilon_{i,t} \leq (C_{\text{err}} - C_\epsilon)\varepsilon.$$

By choosing the constant for the safety margin to be sufficiently large, specifically $C_\epsilon > C_{\text{err}}$, the coefficient $(C_{\text{err}} - C_\epsilon) < 0$. Consequently, for all sufficiently small $\varepsilon$, the term $E_t - \epsilon_t$ is negative.

This implies that its positive part is zero, leading to the following:

$$\left[\alpha_i - V_{d_i}^{\pi_t}\right]_+ \lesssim [E_t - \epsilon_{i,t}]_+ = 0.$$

This means $\left[\alpha_i - V_{d_i}^{\pi_t}\right]_+ \leq C \cdot 0 = 0$. This completes the proof. $\qquad\square$

## G ADDITIONAL DETAILS OF EXPERIMENTS

All experiments are conducted in a randomly generated CMDP, with results averaged over five distinct random seeds. The environment is defined by a state space of $S = 20$ states, an action space of $A = 5$ actions, a finite horizon of $H = 5$ steps and $m = 1$ constraint. The environment's dynamics are stochastic; for each state-action pair $(s, a)$ and step $h$, the transition probabilities $\tilde{p}_h(\cdot|s, a)$ are sampled from a Dirichlet distribution with a low concentration parameter of 0.1 to foster sparse transitions.

The learning challenge is shaped by the conflicting design of the reward and constraint functions. The reward $\tilde{r}_h(s, a)$ is binary. At initialization, we independently draw $\tilde{r}_h(s, a) \sim \text{Bernoulli}(0.5)$ for each step $h$ and state-action pair $(s, a)$. Hence, each $\tilde{r}_h(s, a) \in \{0, 1\}$. The constraint value $\tilde{d}_h(s, a)$ is defined in opposition to the reward function: $\tilde{d}_h(s, a) = 1 - \tilde{r}_h(s, a)$. This design creates a challenging learning problem where the agent must balance the conflicting objectives of maximizing rewards while satisfying the constraint (Moskovitz et al., 2023). At the beginning of each run, the initial state $s_0$ is selected uniformly at random and remains fixed for all subsequent episodes.

We assess algorithm performance under two threshold scenarios. In the fixed-threshold setting, the threshold $\alpha$ is set to half of the maximum achievable expected constraint value: $\alpha = \frac{1}{2} \max_{\pi \in \Pi} V_d^\pi$. This ensures the constraint is both feasible and non-trivial. To model more dynamic conditions, the stochastic-threshold setting samples a constraint value $\alpha_t$ for each episode $t$ from a Normal distribution, $\alpha_t \sim \mathcal{N}(\alpha, (0.5\alpha)^2)$, centered at the fixed threshold value, with a standard deviation equal to half of its mean.

Each algorithm is executed for $T = 80000$ episodes, with the confidence parameter $\delta = 0.1$. To effectively translate theoretical guarantees into practical performance, we introduce empirically tuned scaling factors. The exploration bonus is scaled by a factor of $10^{-3}$, akin to that of Kitamura et al. (2024). Similarly, the safety margin is scaled by a factor of $10^{-5}$ to mitigate the over-conservatism of the theoretical bound and observe the algorithms' behavior in relatively smaller episodes.

All experiments were performed on a Lenovo ThinkBook 14 G5+ APO with an AMD Ryzen 7 7840H.

## H DECLARATION ON LARGE LANGUAGE MODELS

Large Language Models were used for (1) polishing the wording of the manuscript for clarity and readability, (2) brainstorming about algorithm names and their abbreviations, and (3) assisting in formalizing proof sketches into some lemma statements, which is later manually checked to ensure correctness.

