# OpenReview forum: "Achieving $\tilde{O}(1)$ Strong Constraint Violation and Sublinear Strong Regret in Online CMDPs"
_ICLR.cc/2026/Conference — Submitted to ICLR 2026_

### Official Review · Reviewer_6bYB · 2025-10-17

**Soundness:** 2
**Presentation:** 2
**Contribution:** 2
**Rating:** 2
**Confidence:** 4

**Summary:**

The authors propose a primal-dual method which employing optimism and regularisation of the dual update attains $\widetilde O(1) $ strong violation e $\widetilde{O}(T^{7/8})$ regret in stochastic CMDPs assuming Slater's condition and knowledge of the Slater parameter. Moreover, the algorithm attains last iterate convergence with respect to the optimal safe policy.

**Strengths:**

I find the use of the pessimistic $\epsilon_t$ factor in the estimated constraints definition interesting.

**Weaknesses:**

I have many concerns on the novelty and the significance of the work.

Specifically:

1. It is not clear to me which is the difference between standard episodic CMDP and CMDPs with stochastic thresholds. For instance, assume that standard constraints are of the form $g_t^\top q \leq \alpha_t$, which is exactly the setting studied in this work (I have expanded the expectation using the occupancy measure and I have changed changed $\geq$ to $\leq$ for simplicity, but it is the same). Now, if I want a fixed threshold such as, for example $1/2$, I can rewrite the constraints as $(g_t + \frac{1}{2H}-\frac{\alpha_t}{H})^\top q \leq 1/2$, since the occupancy sums to $H$. Now defining $g^\prime_t=g_t + \frac{1}{2H}-\frac{\alpha_t}{H}$ we have a constraints of there form $g^{\prime\top}_t q \leq \alpha_t$, where $\alpha=1/2$. Thus, we have a reduction from your setting to standard CMDPs.

2. The authors are missing the following reference [1]. In [1], the authors designs a CMDP algorithm (which works with adversarial losses) that achieves constant strong constraint violation and  sublinear $\sqrt{T}$ regret. The regret definition is trivially non-strong since the losses are adversarial. Nonetheless, notice that 1) the strong regret metric makes much less sense that the strong violation one 2) substituting the OMD update with a UCB one the same strong regret can be attained. Thus, I believe that the novelty this work is strongly worsen by [1].

3. The regret bound of the work is far from being optimal.

4. The algorithm strongly relies on Muller et al. (2024). Thus, the algorithmic novelty is limited.

Minor: I think there is something wrong in the template. The bottom margin is too large.

[1] Stradi, F. E., Castiglioni, M., Marchesi, A., & Gatti, N. (2025). Learning Adversarial MDPs with Stochastic Hard Constraints. In Forty-Second International Conference on Machine Learning

**Questions:**

See weaknesses.

---

> ### Author Response · Authors · 2025-11-20
>
> >It is not clear to me which is the difference between standard episodic CMDP and CMDPs with stochastic thresholds. For instance, assume that standard constraints are of the form $g_t^\top q \leq \alpha_t$, which is exactly the setting studied in this work (I have expanded the expectation using the occupancy measure and I have changed $\geq$ to $\leq$ for simplicity, but it is the same). Now, if I want a fixed threshold such as, for example 1/2, I can rewrite the constraints as $(g_t + \frac{1}{2H} - \frac{\alpha_t}{H})^\top q \leq 1/2$, since the occupancy sums to H. Now defining $g_t' = g_t + \frac{1}{2H} - \frac{\alpha_t}{H}$ we have a constraints of there form $g_t'^\top q \leq \alpha_t$, where $\alpha = 1/2$. Thus, we have a reduction from your setting to standard CMDPs.
>
> We believe there is a possible misunderstanding. In our formulation, the thresholds are external environmental variables **independent of state-action pairs** and are estimated via global empirical averaging. This is fundamentally different from the Bellman-based evaluation used for the state-action dependent constraint values. Our formulation ($g^{\top} q \le \alpha$ with unknown $\alpha$) combines Bellman estimation for the constraint functions and MC estimation for the thresholds. Due to this distinct statistical structure, our setting is not equivalent to the proposed reduction.
>
> >The authors are missing the following reference [1]. In [1], the authors designs a CMDP algorithm (which works with adversarial losses) that achieves constant strong constraint violation and sublinear $\sqrt{T}$ regret. The regret definition is trivially non-strong since the losses are adversarial. Nonetheless, notice that 1) the strong regret metric makes much less sense that the strong violation one 2) substituting the OMD update with a UCB one the same strong regret can be attained. Thus, I believe that the novelty this work is strongly worsen by [1].
>
> We sincerely appreciate you pointing out this relevant work. We have carefully read reference [1] and will incorporate it into our Introduction and Related Work sections. However, we respectfully disagree that it diminishes the novelty of our work. Even considering [1], our work remains of significant importance:
>
> 1. Beyond the results on strong regret and constraint violation, there is a critical property: **last-iterate convergence**. Last-iterate convergence implies that the final policy converges to the optimal solution, which is a **crucial issue for the deployment of algorithms in real-world problems**. Based on the work in [1], one might raise a valid concern: although it achieves constant strong constraint violation, due to its reliance on weak regret, we can reasonably suspect that its policy might oscillate around the optimal solution. The $\tilde{O}(\sqrt{T})$ regret holds only because positive and negative values cancel each other out. Consequently, whether it can be safely deployed in practical problems becomes a concern. However, our algorithm does not suffer from this issue. We explicitly show that our algorithm converges to the optimal solution while guaranteeing $\tilde{O}(1)$ strong constraint violation. We are the first work to **achieve $\tilde{O}(1)$ strong constraint violation while guaranteeing last-iterate convergence**.
>
> 2. Secondly, upon a careful reading of [1], its implementation requires two steps: first, using a PD (Primal-Dual) method to learn a safe policy; and second, using that safe policy within another PD algorithm to learn the optimal policy. In contrast, our work does not require pre-learning a safe policy. Instead, we adopt a relatively conservative strategy that gradually relaxes constraints, achieving $\tilde{O}(1)$ strong constraint violation with just a single PD algorithm. This makes our algorithm **more computationally efficient**.
>
> 3. Furthermore, we believe the achievement of **strong regret remains highly significant**. If this were not the case, given that [1] already achieved constant strong constraint violation and $\tilde{O}(\sqrt{T})$ weak regret, why would Stradi et al. publish [2] to achieve $\tilde{O}(\sqrt{T})$ strong regret and strong constraint violation? Does this imply that the significance of that work is ``strongly worsened" by [1]? We believe not. Strong constraint violation and strong regret are closely related; the trade-off between them is subtle, and the theoretical proofs required are distinct.
>
> [2] Stradi, F. E., Castiglioni, M., Marchesi, A., & Gatti, N. (2025). Optimal strong regret and violation in constrained MDPs via policy optimization. The Thirteenth International Conference on Learning Representations.

---

> ### Author Response · Authors · 2025-11-21
>
> >The regret bound of the work is far from being optimal.
>
> The regret bound is indeed not optimal. We will also update our Conclusion section to include this open question: Is it possible to simultaneously guarantee $\tilde{O}(1)$ strong violation and $\tilde{O}(\sqrt{T})$ strong regret under last-iterate convergence? Bridging the gap between $\tilde{O}(T^{7/8})$ bound and $\tilde{O}(\sqrt{T})$ presents a significant challenge, which serves as a key direction for our future research. However, we also emphasize that achieving last-iterate convergence makes a fundamental difference, both in the proof framework and in algorithm deployment.
>
> >The algorithm strongly relies on Muller et al. (2024). Thus, the algorithmic novelty is limited.
>
> We acknowledge that our algorithmic utilizes the ideas of entropy and $L_2$ regularization from Müller et al. (2024), but our contributions extend significantly beyond this foundation.
>
> 1. We have fundamentally **redesigned the optimization objective and the potential function analysis** by integrating time-decaying safety margins, learning rates, and regularization terms. We establish a new theoretical framework to characterize the interplay between the safety margin and per-episode strong regret/violation. This framework allows us to rigorously derive last-iterate convergence and improved regret bounds. As detailed in our proofs, this is a delicate and intricate construction. Notably, if one compares our work directly to Müller et al. (2024), we achieve strictly superior bounds for both strong regret ($\tilde{O}(T^{7/8})$ vs. $\tilde{O}(T^{0.93})$) and constraint violation ($\tilde{O}(1)$ vs. $\tilde{O}(T^{0.93})$).
>
> 2. The **``potential function + safety margin"** approach we introduce is not limited to this specific algorithm; it is **a generalizable theoretical framework** applicable to scenarios where per-episode errors can be expressed functionally.
> Our greatest innovation is that we did not attempt to construct a large safety margin to offset the preceding error terms. Instead, our strategy involves constructing **slower-decaying functions** to control the impact of the error terms term-by-term. This ensures that even if large errors occur early in training, they are brought under control within finite time. This mechanism allows us to achieve $\tilde{O}(1)$ strong violation.
>
> 3. We respectfully invite a discussion on what constitutes **``novelty."** Scientific progress is built by standing on the shoulders of giants. For example, Müller et al. (2024) built upon Efroni et al. (2020) by introducing regularization terms (which were also discussed in Ding et al., 2024). Similarly, Stradi et al. (2025) adapted the PO-DB algorithm by incorporating optimistic $\lambda_t$ and a corresponding loss function. These works are considered novel because they solved open problems or improved bounds. In the same vein, our contribution enables $\tilde{O}(1)$ strong constraint violation under last-iterate convergence by introducing decaying safety margin and   regularization terms. We believe this work represents a step forward for safer deployment, and we do not consider this a minor contribution.
>
> >Minor: I think there is something wrong in the template. The bottom margin is too large.
>
> We confirmed that we are using the official ICLR 2026 template without manual modifications. The large margin is likely an artifact of automatic typesetting.

---

> > ### Comment · Reviewer_6bYB · 2025-11-24
> >
> > I would like to thank the Authors for the prompt responses.
> >
> > > On the stochastic thresholds.
> >
> > I underline that, in my reduction, I did not use the fact that $\alpha_t$ is dependent on the state-action pairs. Indeed, it works in both cases, that is, in the harder scenario where $\alpha_t$ is state-action dependent and in the easier one pointed out by the Authors. I see that the algorithm is estimating it with a global empirical averaging, but I claim that it is not necessary. To see it, think to the simpler simplex case. The “stochastic” constraint is simply identifying an hyperplane that cut the simplex. This hyperplane is of course noisy with respect to the true one which is the expected value of the aforementioned noisy observation. Employing a stochastic threshold is simply adding additional noise to the hyperplane, not changing the fact that the hardness of the problem is identifying an hyperplane with a noisy observation. By simply apply the aforementioned transformation to the feedback observed, any algorithm tailored for stochastic CMDPs works with stochastic thresholds.
> >
> > > On the comparison with [1].
> >
> > 1. I agree that the result on last iterate convergence is interesting. Nonetheless, the following considerations are in order. First, [1] does not attain strong regret because the setting is adversarial, thus strong regret is not attainable. If you substitute an UCB update for the reward in the subroutine, the strong regret guarantees follows immediately. Finally, employing the meta algorithm, it allows to achieve constant violation. This algorithm guarantees convergence as any UCB approach when $T$ is known. Moreover, to do that your paper assume the knowledge of the Slater’s parameter, which is not the case of [1]. Second, the last-iterate convergence result of your paper follows from employing techniques from Muller et al. (2024).
> > 2. I agree that the algorithm presented in your work is more efficient than the one in [1]. This is not due the two steps. Indeed, it is due to the fact that that you avoid the convex projection at each episode. Nonetheless, as far as I see, computational aspects are not highlighted as selling points in the paper.
> > 3. On the comparison with [2], I think that the main focus on that work was to build a primal-dual method that achieve strong regret and violation, since it was the problem left open from Muller et al. (2024). As far as I see, the focus of this work is to attain strong regret and strong constant violation. As pointed out before, these results can be easily obtain by state-of-the-art.
> >
> > To conclude on this aspect, the question which the paper wants to answer clearly states:
> >
> > “Can we design a CMDP algorithm that achieves (i) $O(1)$ strong constraint violation, and (ii) sublinear strong regret, while guaranteeing (iii) last-iterate convergence?” The algorithm presented before attains these results, with better bound on the regret.
> >
> > > On the novelty.
> >
> > I believe that in a top conference such as ICLR, the novelty must be sufficiently high for publication. As pointed out before, a significantly large part of results attained in this work can be replicated by state-of-the-arts algorithm. I still believe that the main contribution of this work is attaining the constant strong violation (and last iterate) employing an efficient primal-dual method. Nonetheless, first this is not the selling point stated in the work. Second, the “efficiency” part of the algorithm comes directly from Müller et al 2024. Thus, to me, both the results and the algorithmic novelty are not sufficient for publication.

---

> > > ### Author Response · Authors · 2025-11-27
> > >
> > > We thank the reviewer again for the detailed follow-up. We respond point-by-point below and clarify several technical misunderstandings regarding what can or cannot be obtained from [1], [2], or from Müller et al. (2024).
> > >
> > > > On the stochastic thresholds.
> > >
> > > We appreciate the reviewer’s clarification. The stochastic-threshold component is a practical extension rather than a core novelty, and we include it mainly to show that FlexDOME naturally accommodates such settings. Handling stochastic thresholds is only a minor aspect of our work.
> > >
> > > >On the novelty and Compare with other works
> > >
> > > 1. **Core novelty of our work**. We respectfully clarify that the core novelty of our work lies in the theoretical framework that enables the simultaneous guarantees of: (i) $\tilde{O}(1)$ strong violation; (2) last-iterate convergence and (iii) sublinear strong regret. To the best of our knowledge, no prior method, including SoTA algorithms cited by the reviewer, offers a complete analysis that achieves all these three properties together. **It is also not the case that these guarantees can be trivially attained by plugging in a known SoTA method.**
> > > 2. **On the claim that `` If you substitute an UCB update for the reward in the subroutine, the strong regret guarantees follows immediately.''** Strong regret does **not** automatically follow from replacing the adversarial loss oracle with a UCB update. The update dynamics in [1] were analyzed under adversarial feedback and do not control how stochastic estimation errors propagate through the dual variables in the CMDP setting. To our knowledge, no existing analysis shows that strong regret ``follows immediately'' from such a  substitution; establishing it would require nontrivial new arguments rather than being a straightforward extension. Whether this modification can yield strong regret remains theoretically unresolved.
> > > 3. **On last-iterate convergence**. Across all modifications proposed by the reviewer, none provides a mechanism for achieving last-iterate convergence in CMDPs. Moreover, our analysis is **not a trivial extension** of Müller et al. (2024). By incorporating a time-decaying safety margin and dynamic regularization, we obtain a strictly stronger form of last-iterate convergence: after a finite number of episodes, the constraint violation becomes **exactly zero**, rather than merely asymptotically vanishing. This finite-time zero-violation behavior does not appear in [1], nor does Müller et al. (2024) provide such a guarantee. We would also like to clarify that the reviewer’s statement that these results ((i) $O(1)$ strong constraint violation, (ii) sublinear strong regret, and (iii) last-iterate convergence) can already be achieved with even better regret bounds **is not supported by any existing analysis**. Neither [1] nor Müller et al. (2024) establishes these three guarantees jointly, and no proof has been provided in the reviewer’s comments showing that such results follow from their algorithms or suggested modifications (such as substituting UCB updates).
> > > 4. **About efficiency.** We thank the reviewer for recognizing the efficiency of our algorithm, and we will emphasize this point more explicitly in the revision.
> > > 5. **Missing our proof contributions**. The reviewer’s assessment overlooks some key difficulties: achieving both constant strong violation and last-iterate convergence is **not attainable by simply combining safety margins and regularization**. Prior weak-regret analyses (Liu et al., 2021; Kalagarla et al., 2025) using safety margin rely on cumulative offsets, which fail in the strong-regret setting because positive error terms cannot be canceled once incurred. Our approach instead involves constructing slower-decaying control functions and dynamic regularization terms that bound each error term per episode, ensuring that early errors remain controlled within finite time. This mechanism requires a proof strategy fundamentally different from Müller et al. (2024).

---

### Official Review · Reviewer_GW2V · 2025-10-30

**Soundness:** 2
**Presentation:** 3
**Contribution:** 2
**Rating:** 4
**Confidence:** 3

**Summary:**

This paper proposes FlexDOME, a regularized primal–dual algorithm for online constrained MDPs under strong regret and constraint violation metrics. By introducing a decaying safety margin that tightens and gradually relaxes the constraint thresholds, FlexDOME achieves a near-constant $\tilde{\mathcal O}(1)$ strong constraint violation and sublinear $\tilde{\mathcal O}(T^{7/8})$ strong reward regret, while ensuring last-iterate convergence. Theoretical analyses establish these guarantees, and experiments on tabular CMDPs confirm the method’s superior safety performance compared to baselines.

**Strengths:**

+ The paper is well written and easy to follow. The technical contribution is clearly presented. To achieve tighter constraint satisfaction, the algorithm introduces a time-decaying safety margin that reduces strong constraint violation from the state-of-the-art $\tilde{\mathcal O}(\sqrt{T})$ to $\tilde{\mathcal O}(1)$, at the expense of higher regret.

**Weaknesses:**

- The paper is largely an extension of Müller et al. (2024); most algorithmic ideas and analytical techniques are inherited there.

- The method assumes prior knowledge of the Slater constant ($\gamma$), and the resulting regret bound scales inversely with $\gamma$. In some critical scenarios,  where $\gamma$ is near to zero, the regret may become large.

- If I understand correctly, the algorithm used TPE for policy evaluation. For sample complexity, it would be more reasonable to take these interactions into account.

- To achieve constant constraint violation, the algorithm sets the decaying safety margin $\epsilon$ as $\tilde{\mathcal O}(t^{-1/8})$, which leads to higher regret than the state-of-the-art  $\tilde{\mathcal O}(\sqrt{T})$. So could the algorithm be made more adaptive—for example, by adjusting the decay rate dynamically to balance regret and violation?

- The experiments compare only two relatively weak baselines. Given that Table 1 lists stronger alternatives such as (Stradi et al., 2025) and (Zhu et al., 2025), these should be included for a fair and comprehensive evaluation.

**Questions:**

Please see weakness above.

---

> ### Author Response · Authors · 2025-11-26
>
> >Q1: The paper is largely an extension of Müller et al. (2024); most algorithmic ideas and analytical techniques are inherited there.
>
> We thank the reviewer for the comment. While our method incorporates entropy and $L_2$ regularization components that also appear in Müller et al. (2024), the core of our algorithm and analysis is substantially different. In particular, the guarantees we obtain cannot be derived by directly applying or extending the techniques in Müller et al. (2024), as explained below.
>
> 1. **Redesigned optimization objective and potential-function framework.** We fundamentally reformulate the objective and the potential function analysis by integrating time-decaying safety margins, learning rates, and regularization in a unified framework. This allows us to characterize, for the first time, how the safety margin interacts with per-episode strong regret and violation. As a result, we obtain strictly stronger guarantees than Müller et al. (2024): $\tilde{O}(T^{7/8})$ vs. $\tilde{O}(T^{0.93})$ regret and $\tilde{O}(1)$ vs. $\tilde{O}(T^{0.93})$ violation.
>
> 2. **New analysis for strong violation.** Achieving both constant strong violation and last-iterate convergence is not attainable by simply combining safety margins and regularization. Prior weak-regret analyses (Liu et al., 2021; Kalagarla et al., 2025) using safety margin rely on cumulative offsets, which **fail in the strong-regret setting** because positive error terms cannot be canceled. Our greatest innovation is that we did not attempt to construct a large safety margin to offset the preceding error terms. Instead, our strategy involves constructing **slower-decaying control functions** that bound each error term per episode, ensuring that early errors **remain controlled within finite time**. This mechanism requires a proof strategy fundamentally different from Müller et al. (2024).
>
> 3. **General applicability of the potential-margin mechanism.** The **``potential function + safety margin"** methodology is not specific to our update rule. It applies broadly whenever per-episode errors can be expressed functionally, making it a reusable analytical tool that extends beyond the scope of Müller et al. (2024).
>
> 4. **Experimental improvement.** The experiments further corroborate the theory: the added safety-margin mechanism yields smaller constraint-violation values and a much slower growth trend compared to existing last-iterate methods.
>
> >Q2: The method assumes prior knowledge of the Slater constant ($\gamma$), and the resulting regret bound scales inversely with $\gamma$. In some critical scenarios, where $\gamma$ is near to zero, the regret may become large.
>
> We thank the reviewer for the helpful suggestions for improving the paper. we agree that requiring a known Slater constant is a limitation, which is a common and relatively mild assumption (Efroni et al., 2020; Ying et al., 2022; Ding et al., 2023; Kitamura et al., 2024; Ding et al., 2022c; Paternain et al., 2022; Ding et al., 2023). We appreciate the reviewer noting this point and view this as an exciting direction for future work, as no existing method achieves last-iterate convergence without the Slater condition.
>
> >Q3: If I understand correctly, the algorithm used TPE for policy evaluation. For sample complexity, it would be more reasonable to take these interactions into account.
>
> We have indeed discussed the TPE in our analysis, and the resulting impact is captured in **Lemma 16** and the preceding discussion.

---

> ### Author Response · Authors · 2025-11-26
>
> >Q4: To achieve constant constraint violation, the algorithm sets the decaying safety margin $\epsilon$ as $\tilde{\mathcal O}(t^{-1/8})$, which leads to higher regret than the state-of-the-art $\tilde{\mathcal O}(\sqrt{T})$. So could the algorithm be made more adaptive—for example, by adjusting the decay rate dynamically to balance regret and violation?
>
> We thank the reviewer for raising this important point. The choice $\epsilon_t=\tilde{\mathcal O}(t^{-1/8})$ is not a manual design choice but **the outcome of balancing regret and violation** in our analysis. The key reason is that the regret rate arises from the combined effect of the potential-function framework, the decaying safety margin, the learning rate, and the regularization needed to guarantee last-iterate convergence under strong constraints. Our per-step bounds take the form $$\text{Per-step Strong Regret} \le A \cdot (\Phi_t)^{1/2} + B \cdot \tau_t + C \cdot \epsilon_t, \quad\text{Per-step Strong Violation} \le [A \cdot (\Phi_t)^{1/2} + D \cdot \tau_t - \epsilon_t]_+.$$ To ensuring $\tilde{O}(1)$ cumulative violation, $\epsilon_t$ need decay slowly enough to dominate the positive terms in the violation bound, while a slower decay enlarges its contribution to regret.
>
> Under this trade-off, the exponent $1/8$ is the explicit balancing point that **minimizes the largest $T$-order** among all regret terms subject to constant strong violation. Thus, simply making $\epsilon_t$ adaptive or data-dependent cannot improve the regret while keeping both last-iterate convergence and constant violation.
>
> We believe that improving beyond the current $\tilde{O}(T^{7/8})$ will likely require (i) **a new potential function**, (ii) **alternative regularization**, or (iii) **an update rule with smaller optimization error**, rather than adjusting the decay schedule alone. We leave this direction as future work.
>
> >Q5: The experiments compare only two relatively weak baselines. Given that Table 1 lists stronger alternatives such as (Stradi et al., 2025) and (Zhu et al., 2025), these should be included for a fair and comprehensive evaluation.
>
> We would like to clarify that the two baselines included are not ``weak baselines’’. Vanilla-PD is used to isolate the role of regularization and safety margin in stabilizing last-iterate convergence, and UOpt-RPGPD is the state-of-the-art existing method with proven last-iterate guarantees. Our experimental comparison therefore focuses on algorithms operating in the same
> **last-iterate convergence** regime, which is the setting most relevant to our contributions.
>
> Regarding the stronger baselines listed in Table 1, we note that Stradi et al. (2025) is purely theoretical and lack of experiments for their work, and therefore we only compare our results with OMDPD (Zhu et al. (2025)) under the same setup. We present the numerical comparison of strong constraint violation below.
>
> **Table: Comparison of strong constraint violation between OMDPD and FlexDOME**
>
> | Algorithm / Episode | 100   | 200   | 400   | 600   | 800   | 1000  | 1500  | 2000  | 2500  | 3000  |
> |---------------------|--------|--------|--------|--------|--------|--------|--------|--------|--------|--------|
> | OMDPD               | 188.54 | 338.26 | 665.73 | 759.52 | 830.76 | 878.21 | 958.12 | 1012.57 | 1039.43 | 1079.21 |
> | FlexDOME            | 10.89  | 15.34  | 23.28  | 30.83  | 38.18  | 45.19  | 59.05  | 68.35  | 76.20  | 83.68  |
>
> We notice that our algorithm yields smaller constraint-violation values and exhibits a much slower growth rate compared to the baseline. Our primary contribution is the theoretical development, and we plan to include more comprehensive comparisons in future work.

---

> ### Author Response · Authors · 2025-11-28
>
> Dear Reviewer GW2V,
>
> We sincerely appreciate the time and effort you dedicated to reviewing our submission. We will incorporate your constructive suggestions to enhance the presentation and highlight the contributions of our work.
>
> As the discussion period is drawing to a close, we hope that our responses have addressed your concerns. Should any points remain unclear or require further elaboration, we would be happy to provide additional clarification.
>
> Thank you again for your evaluation and consideration.
>
> Best,
>
> Authors of Submission 11226

---

### Official Review · Reviewer_WsMi · 2025-11-01

**Soundness:** 3
**Presentation:** 3
**Contribution:** 3
**Rating:** 6
**Confidence:** 4

**Summary:**

The paper studies online safe reinforcement learning in finite-horizon CMDPs under strong metrics of both regret and constraint violation (i.e., without cross-episode cancellation). It proposes a regularized primal–dual algorithm, FlexDOME, which introduces a decaying safety margin to each constraint threshold and employs entropy and $\ell_2$ regularization to stabilize last-iterate dynamics. With a carefully calibrated schedule for the learning rate, regularization, and margin parameters, the algorithm achieves $\tilde{O}(1)$ cumulative strong constraint violation and sublinear strong regret ($\tilde{O}(T^{7/8})$), along with a non-asymptotic last-iterate convergence guarantee. The analysis follows and extends the recent work of Müller et al., incorporating a vanishing margin mechanism to attain the $\tilde{O}(1)$ strong-violation bound.

**Strengths:**

- FlexDOME achieves $\tilde O(1)$ cumulative strong violation and $\tilde O(T^{7/8})$ strong regret, with last-iterate convergence---a meaningful milestone for strong safety during learning.
- The margin-regularized Lagrangian plus entropy/dual regularization induces a well-behaved concave--convex landscape; the potential-function analysis is clean and reusable.
- The paper is generally well written, and its overall theoretical contribution appears sound. Although I have not verified every technical detail, the employed approaches are standard, and the derivations seem largely correct.

**Weaknesses:**

- The paper motivates safety in broad terms but does not present a concrete, realistic scenario where strong-metric guarantees, and specifically $\tilde O(1)$ violation during learning, are operationally indispensable, with explicit costs for any violation.

- The paper overlooks several recent and closely related works that have appeared on adjacent topics.

- The $\tilde O(T^{7/8})$ strong regret is traded for $\tilde O(1)$ strong violation and last-iterate stability. A brief discussion on whether the $7/8$ exponent is tight or proof-artifact (and whether alternative decay could improve it) would help.

- Treating a stochastic threshold effectively adds an estimated scalar with standard concentration; the contribution feels more like a neat extension than a central novelty.

- Evaluations are on synthetic tabular CMDPs with one constraint. Including structured environments and sensitivity to schedule constants would strengthen practical relevance.

**Questions:**

- You regularize with policy entropy $H(\pi)$ and then rewrite over occupancy measures with an entropy $H(q)$. Please justify the exact equivalence.

- Is $t^{-1/8}$ required for $\tilde O(1)$ strong violation with your potential bounds, or could adaptive/data-dependent schedules preserve $\tilde O(1)$ violation while improving the $T^{7/8}$ regret rate?

- Please expand on realistic application domains where \emph{strong} safety with vanishing cumulative violation is the right operational objective (and why), beyond general safety narratives.

- Please provide the full MDP specifications used in the appendix (transition kernels, reward/cost functions, threshold distributions, and code). ``Randomly generated'' is insufficient for replication or interpreting oscillations.

---

> ### Author Response · Authors · 2025-11-26
>
> >Q1: You regularize with policy entropy $H(\pi)$ and then rewrite over occupancy measures with an entropy $H(q)$. Please justify the exact equivalence.
>
> We thank the reviewer for the helpful suggestions for improving the paper. We provide the explicit derivation below and will include it in the Appendix.
>
> By definition, the policy entropy is $H(\pi) = -E_{\pi}[\sum_h \log \pi_h(a_h|s_h)]$. Since the occupancy measure $q_h(s,a)$ represents the visitation probability, this expectation can be rewritten as a sum over the state-action space:$H(\pi) = -\sum_{h,s,a} q_h(s,a) \log \pi_h(a|s)$. Using the relationship between valid occupancy measures and policies $\pi_h(a|s) = \frac{q_h(s,a)}{\sum_{a'} q_h(s,a')}$, and substituting into it obtains: $$H(\pi) = -\sum_{h,s,a} q_h(s,a) \log \left( \frac{q_h(s,a)}{\sum_{a'} q_h(s,a')} \right) \equiv H(q).$$ Thus, the regularization term $H(q)$ used in the convex formulation (Lemma 9) is mathematically identical to the original policy entropy $H(\pi)$.
>
> >Q2 and W2:
> Is $t^{-1/8}$ required for $\tilde{O}(1)$ strong violation with your potential bounds, or could adaptive/data-dependent schedules preserve $\tilde{O}(1)$ violation while improving the $T^{7/8}$ regret rate? The $\tilde{O}(T^{7/8})$ strong regret is traded for $\tilde{O}(1)$ strong violation and last-iterate stability. A brief discussion on whether the 7/8 exponent is tight or proof-artifact (and whether alternative decay could improve it) would help.
>
> We thank the reviewer for raising this important point. The $t^{-1/8}$ decay is not itself the fundamental bottleneck, and replacing it with an adaptive or data-dependent schedule would not preserve $\tilde{O}(1)$ violation while simultaneously improving the regret rate.
>
> Importantly, the $\tilde{O}(T^{7/8})$ strong regret does not arise from a particular choice of decay, but from **the interplay** between (i) the potential-function framework, (ii) the time-decaying safety margin, (iii) the learning rate, and (iv) the regularization terms required for last-iterate stability under strong constraints. When these components are jointly optimized to guarantee both last-iterate convergence and strong constant violation, the resulting optimization and stability errors dominate and lead to the current $T^{7/8}$ rate. The exponent $1/8$ of $t$ is the **explicit balancing point** that minimizes the largest $T$-order among all regret term under strong constant violation. Thus, simply making $\epsilon_t$ adaptive or data-dependent cannot improve the regret while keeping both last-iterate convergence and constant violation.
>
> We therefore believe that achieving both $\tilde{O}(1)$ strong violation and improving the regret toward $\tilde{O}(\sqrt{T})$ likely requires designing (i) **a new potential function**, (ii) **alternative regularization**, or (iii) **an algorithm with smaller optimization error**, rather than more refined decay schedules.

---

> ### Author Response · Authors · 2025-11-26
>
> > Q3 and W1: The paper motivates safety in broad terms but does not present a concrete, realistic scenario where strong-metric guarantees, and specifically $\tilde O(1)$ violation during learning, are operationally indispensable, with explicit costs for any violation. Please expand on realistic application domains where strong safety with vanishing cumulative violation is the right operational objective (and why), beyond general safety narratives.
>
> We thank the reviewer for the helpful suggestions for improving the paper and highlight **two representative real-world settings**. First, **in critical infrastructure control**, such as power-grid frequency or voltage regulation, constraint violations impose cumulative mechanical fatigue and thermal stress on turbines, generators, and actuators [1]. A guarantee of $\tilde{O}(\sqrt{T})$ cumulative violation implies that the total stress grows unbounded over long operational horizons, which is incompatible with equipment-lifetime requirements and reliability standards. In contrast, strong constraint violation ensures that the total number of off-limit excursions remains uniformly bounded [2].
>
> Second, It is also essential **in clinical therapeutics**, including automated anesthesia and sepsis management [3]. Here, constraints correspond to hard physiological thresholds, such as maintaining adequate mean arterial pressure, where even a few severe breaches can trigger organ dysfunction or drastically increase mortality risk [4]. Crucially, these harms cannot be “offset” by subsequently safe behavior, making horizon-independent safety guarantees more appropriate than average-case criteria. Recent RL approaches for critical-care dosing similarly emphasize maintaining hemodynamic or toxicity variables within strict safety bounds.
>
> We will integrate these motivations and references into the revised manuscript.
>
> >Q4 and W5: Evaluations are on synthetic tabular CMDPs with one constraint. Including structured environments and sensitivity to schedule constants would strengthen practical relevance. Please provide the full MDP specifications used in the appendix (transition kernels, reward/cost functions, threshold distributions, and code). "Randomly generated" is insufficient for replication or interpreting oscillations.
>
> We kindly remind the reviewer that we indeed have added the full MDP specifications into **Appendix G**, including transition kernels, reward and cost functions and threshold distributions. And we have submitted the code into the supplementary material.
>
> >W2: The paper overlooks several recent and closely related works that have appeared on adjacent topics.
>
> We appreciate the reviewer for pointing this out. In the revision, we will incorporate closely related recent works as follows ([1]-[5]) and will clarify how these works relate to our work. We would be glad to include any further relevant references the reviewer recommends.
>
> >W4: Treating a stochastic threshold effectively adds an estimated scalar with standard concentration; the contribution feels more like a neat extension than a central novelty.
>
> We agree that the stochastic-threshold treatment is a practical extension rather than a core novelty, and we include it mainly to show that FlexDOME naturally accommodates such cases. Dealing with stochastic thresholds is just a minor part of our contribution.
>
> [1] Su, T., Wu, T., Zhao, J., Scaglione, A., & Xie, L. (2025). A review of safe reinforcement learning methods for modern power systems. Proceedings of the IEEE.
>
> [2] Tabas, D., & Zhang, B. (2022, June). Computationally efficient safe reinforcement learning for power systems. In 2022 American Control Conference (ACC) (pp. 3303-3310). IEEE.
>
> [3] Cai, X., Chen, J., Zhu, Y., Wang, B., & Yao, Y. (2023). Towards real-world applications of personalized anesthesia using policy constraint Q learning for propofol infusion control. IEEE Journal of Biomedical and Health Informatics, 28(1), 459-469.
>
> [4] Jia, Y., Lawton, T., Burden, J., McDermid, J., & Habli, I. (2021). Safety-driven design of machine learning for sepsis treatment. Journal of Biomedical Informatics, 117, 103762.
>
> [5] Stradi, F. E., Castiglioni, M., Marchesi, A., & Gatti, N. (2025). Learning Adversarial MDPs with Stochastic Hard Constraints. In Forty-Second International Conference on Machine Learning

---

> ### Author Response · Authors · 2025-11-28
>
> Dear Reviewer WsMi,
>
> We sincerely appreciate the time and effort you dedicated to reviewing our submission. And we will incorporate your constructive suggestions to enhance the presentation and highlight the contributions of our work.
>
> We have carefully addressed all your previous questions and have provided detailed explanations in our rebuttal. As the discussion period is drawing to a close, we hope that our responses have resolved these concerns. Should any points remain unclear or require further elaboration, we would be happy to provide additional clarification.
>
> Thank you again for your evaluation and consideration.
>
> Best,
>
> Authors of Submission 11226

---

### Official Review · Reviewer_Grfz · 2025-11-01

**Soundness:** 3
**Presentation:** 2
**Contribution:** 3
**Rating:** 6
**Confidence:** 3

**Summary:**

The paper studies online CMDPs under strong metrics with no cancellation for regret and constraint violations and under stochastic thresholds, which constitutes a more general setting than standard CMDPs. The authors propose a regularized primal-dual algorithm with entropy and l2 norm terms, and provide the corresponding strong regret, strong constraint violation, and last-iteration convergence analysis. Experiments on tabular CMDPs show the algorithm keeps lower violation than existing methods.

**Strengths:**

1. This paper studies a more general and stronger CMDP setting than standard CMDPs, with stochastic thresholds and strong regret/violations.
2. This paper proposes a regularized primal-dual algorithm that can simultaneously achieve constant strong violations, sub-linear regret, and last-iterate convergence.
3. The experiments demonstrate to some extent that the algorithm, with regularizations, is able to tame oscillation while keeping the regret and violations in control.

**Weaknesses:**

1. The regret $\tilde{O}(T^{7/8})$, though sub-linear, seems still too high, despite the other strong guarantees. If I am understanding this correctly, this strong regret is the consequence of adopting a decay rate of $t^{-1/8}$, which is essential to achieve the theoretical guarantees. If so, what would the lower bound of strong regret be in this case? I would really like to see some discussions and analysis on where the potential sub-optimality (if any) in regret comes from, the regret analysis being too relaxed, or the fundamental problem of this algorithm design?
2. From the last-iterate convergence results, it can be translated that the sample complexity to achieve a $\varepsilon$-optimal policy is $\tilde{O}(\varepsilon^{-7})$, which is too high. Similar arguments and analysis as in Weakness 1 are expected here.

**Questions:**

Please see weaknesses for my questions. Additionally, I am curious if the authors think these results are improvable, and if so, how would the authors further improve the regret and/or convergence rate?

---

> ### Author Response · Authors · 2025-11-20
>
> >If I am understanding this correctly, this strong regret is the consequence of adopting a decay rate of
> , which is essential to achieve the theoretical guarantees. If so, what would the lower bound of strong regret be in this case? I would really like to see some discussions and analysis on where the potential sub-optimality (if any) in regret comes from, the regret analysis being too relaxed, or the fundamental problem of this algorithm design? Similar concerns apply to the sample complexity.
>
> We sincerely thank you for this insightful comments that have touched upon the core challenge of our algorithm design and theoretical proof. We agree with your observation. Our current regret of $\tilde{O}(T^{7/8})$ is still high. In settings without last-iterate convergence, the optimal strong regret is $\tilde{O}(\sqrt{T})$. It remains unknown whether this optimal solution changes under the property of last-iterate convergence. In terms of practical algorithm deployment, whether last-iterate convergence can be guaranteed makes a significant difference in the results. During our theoretical proof process, we also asked ourselves the same question: How does a strong regret higher than $\tilde{O}(\sqrt{T})$ arise?
>
> First, to state **our conclusion**: The resulting $\tilde{O}(T^{7/8})$ is not solely due to the choice of the safety margin. Rather, it is a fundamental problem stemming from the algorithm design. Specifically, It is due to the interplay between the potential function framework, the time-decaying safety margin, the learning rate, and the regularization terms required to ensure last-iterate convergence with constant strong violations.
>
> Below, we provide a detailed analysis of where this sub-optimality comes from:
>
> Our proof relies on a specific potential function, $\Phi_t$. The analysis proceeds in three steps:
>
> Step 1: Convergence of the potential function.
> $$\Phi_t \le (\text{a decaying term dependent on the initial potential})\Phi_1 + (\text{optimization error from updates}) + (\text{estimation error})$$
> Step 2: Linking convergence to performance (core contribution), which is essential for proving last-iterate convergence:
> $$\text{Per-step Strong Regret} \le A \cdot (\Phi_t)^{1/2} + B \cdot \tau_t + C \cdot \epsilon_t$$$$\text{Per-step Strong Violation} \le [A \cdot (\Phi_t)^{1/2} + D \cdot \tau_t - \epsilon_t]_+$$
>
> Step 3: Balancing the terms (we omit constant terms here).
> $$\text{Strong Regret} \le \sum \left( \sqrt{\text{the first decaying term}} + \sqrt{\text{optimization error}} + \sqrt{\text{estimation error}} + \tau_t + \epsilon_t \right)$$
> $$\text{Strong violation} \le \sum [\sqrt{\text{the first decaying term}} + \sqrt{\text{optimization error}} + \sqrt{\text{estimation error}} + \tau_t - \epsilon_t]_+$$
> We focus only on the order of $T$ here. Therefore, we must **minimize the maximum order of $T$** among the above five terms in strong regret. This leads to our solutions for $\eta_t, \tau_t$, and $\epsilon_t$. Consequently, the terms introducing an order higher than $\tilde{O}(\sqrt{T})$ are the latter four terms.
>
> >I am curious if the authors think these results are improvable, and if so, how would the authors further improve the regret and/or convergence rate?
>
> We found that this issue cannot be resolved simply by adjusting the time-decaying safety margin, learning rate, or regularization terms. Even if we assume no estimation error and solve for the minimum maximum order of $T$ using the remaining four terms, the lowest achievable order is $\tilde{O}(T^{3/4})$ (which is still high). Therefore, we believe the key to further improving the bound lies not in designing more dynamically adjustable safety margins, learning rates, or regularization, but potentially in **designing a new potential function**, **new regularization** and an algorithm with **smaller optimization error**. This limitation is inherent to the nature of our current algorithm.
>
> However, we would like to highlight **some interesting contributions in our theory**  that might accelerate the achievement of $\tilde{O}(\sqrt{T})$ strong regret and $\tilde{O}(1)$ violation under last-iterate convergence. Our approach of introducing a safety margin should not be limited to our current algorithm; this concept has broad applicability. It is suitable for scenarios where accumulated errors can be expressed in functional form, though it is less applicable to implicit cases.
>
> Our greatest innovation is that we did not attempt to construct a large safety margin to offset the preceding error terms. Instead, our strategy involves constructing functions that decay slower to control the impact of the error terms term-by-term. This ensures that even if large errors occur in the early stages of training, they remain controllable within a finite time (see the proof of the main theorem for details). This idea eliminates the need for a very large safety margin term; instead, a slower-decaying function is sufficient to achieve $\tilde{O}(1)$.

---

> ### Author Response · Authors · 2025-11-28
>
> Dear Reviewer Grfz,
>
> We sincerely appreciate the time and effort you dedicated to reviewing our submission. We will incorporate your constructive suggestions to enhance the presentation and highlight the contributions of our work.
>
> As the discussion period is drawing to a close, we hope that our responses have addressed your concerns. Should any points remain unclear or require further elaboration, we would be happy to provide additional clarification.
>
> Thank you again for your evaluation and consideration.
>
> Best,
>
> Authors of Submission 11226

---

### Meta-Review · Area_Chair_P5HA · 2025-12-30

**Summary:**

The Reviewers are mixed on this paper, with two of them supporting acceptance (scores 6 and 6) and two of them rejection (score 4 and 2).  Reviewer 6bYB (score 2) is particularly negative about this work, motivating their assessment as the lack of technical novelty. Overall, I think that, with the paper in its current state, the reasons to reject the paper outweigh those to accept it. Thus, I propose rejection.

**Reviewer Concerns:**

The Authors tried to overcome the issues raised by the Reviewers in their reviews. In particular, Reviewer 6bYB (score 2) was the most critical, but I think the Authors only partially addressed the issues raided by them. The most concerning aspect that remains unsolved is the actual relationship between the united work mentioned by Reviewer 6bYB [1] and the work by Muller et al. (2024). This substantially weakens the technical novelty of the paper. Moreover, the obtained results are far from being optimal, which further questions the significance of this paper.

[1] Stradi, F. E., Castiglioni, M., Marchesi, A., & Gatti, N. (2025). Learning Adversarial MDPs with Stochastic Hard Constraints. In Forty-Second International Conference on Machine Learning

**Reviewer Scores:**

Reviewer Grfz, Score: 6 - I believe that the rebuttal would not have changed the reviewer’s opinion, especially given the other reviews.

Reviewer WsMi, Score: 6 - I believe that the rebuttal would not have changed the reviewer’s opinion, especially given the other reviews.

Reviewer GW2V, Score: 4 - I believe that the rebuttal would not have changed the reviewer’s opinion, especially given the other reviews.

Reviewer 6bYB, Score: 2 - I believe that the rebuttal would not have changed the reviewer’s opinion, especially given the other reviews.

---

### Decision · Program_Chairs · 2026-01-26

Reject